# Scalable Back-Propagation-Free Training of Optical Physics-Informed Neural Networks

## Abstract

Physics-informed neural networks (PINNs) have shown promise in solving partial differential equations (PDEs), with growing interest in their energy-efficient, real-time training on edge devices. Photonic computing offers a potential solution due to its high operation speed. However, the lack of photonic memory and the large footprint of current photonic devices prevent training realistic-size PINNs on photonic chips. This paper proposes a completely back-propagation-free (BP-free) and highly salable framework to enable training real-size PINNs on silicon photonics platforms. Our approach involves three key innovations: (1) a sparse-grid Stein derivative estimator to avoid the BP in the loss evaluation of a PINN, (2) a dimension-reduced zeroth-order optimization via tensor-train decomposition to achieve better scalability and convergence in BP-free training, and (3) a scalable on-chip photonic PINN training accelerator design using photonic tensor cores. We validate the performance of our numerical methods in both low- and high-dimensional PDE benchmarks. Through circuit simulation based on real device parameters, we further demonstrate the significant performance benefit (e.g., real-time training, huge chip area reduction) of our photonic accelerator. Our framework addresses the fundamental challenges of photonic AI and will enable real-time training of real-size PINNs on photonic chips.

## 1 Introduction

Partial differential equations (PDEs) are used to describe numerous engineering problems, such as electromagnetic and thermal analysis of IC chips (Kamon et al., 1993; Li et al., 2004), medical imaging (Villena et al., 2015), and safety verification of autonomous systems (Bansal & Tomlin, 2021). Traditional numerical solvers (*e.g.,* finite-difference, finite-element methods) have been well studied and commercialized, but they become prohibitively expensive for high-dimensional PDEs due to the exponential increase of the unknown variables with respect to spatial/temporal/parameter dimensions. This bottleneck becomes more significant in PDE-constrained inverse and control problems, since the forward problem needs to be solved many times in an outer iteration loop.

Physics-informed neural networks (PINNs) (Lagaris et al., 1998; Dissanayake & Phan-Thien, 1994; Raissi et al., 2019) have emerged as a promising approach to solve both forward and inverse problems. Due to the discretization-free nature, PINN is more suitable for solving high-dimensional or parametric PDEs, but current PINN training is still very expensive. For instance, training a PINN for robotic safety analysis (Bansal & Tomlin, 2021) can easily take $> 10$ hours on a powerful GPU. Despite the development of operator learning (Lu et al., 2021), a PINN often needs to be trained *from scratch* again to obtain high-quality solution once the PDE initial conditions, boundary conditions, or measurement data changes. There has been increasing interest in training PINN on edge devices and in a real-time manner, including but not limited to PDE-based safety verification (Bansal & Tomlin, 2021), control (Onken et al., 2021) of autonomous systems, fast and private EPT (Yu et al., 2023). The design of real-time edge training accelerator for PINNs remains a sparse research field.

Photonic computing provides a promising low-energy and high-speed solution for various AI tasks due to the ultra-high operation speed of light. Many optical neural network (ONN) inference accelerators have been proposed (Shen et al., 2017; Tait et al., 2016; Zhu et al., 2022). However, designing a photonic training accelerator for real-size PINNs (e.g., a network with hundreds of neurons per layer) remains an open quesiton due to two fundamental challenges:

- **Large device footprints and low integration density.** Photonic multiply-accumulate (MAC) units such as Mach-Zhnder interferometers (MZIs) are much larger ($\sim$10s of microns) than CMOS transistors. A real-size PINN with $> 10^5$ parameters can easily exceed the available chip size with the square scaling rule where an $N \times N$ weight matrix requires $O(N^2)$ MZIs (Reck et al., 1994; Clements et al., 2016). Actually even the state-of-the-art photonic AI *inference* accelerator (Ramey, 2020) can only handle $64 \times 64$ weight matrices. Training a PINN on an photonic chip will face more significant scalability issue.

- **Difficulty of on-chip back propagation (BP).** It is hard to realize BP on photonic chips due to the lack of memory to store the computational graphs and intermediate results. Several BP-free and *in-situ* BP methods (Gu et al., 2020; 2021a; Filipovich et al., 2022; Buckley & McCaughan, 2022; Oguz et al., 2023; Hughes et al., 2018; Pai et al., 2023) are proposed, but their scalability remains a major bottleneck. This becomes more severe in PINN, since its loss function also includes (high-order) derivative terms. Subspace learning (Gu et al., 2021b) may scale up BP-based training, but still needs to save intermediate states. Due to lack of photonic memory, additional optical-electronic-optical conversion is needed, leading to dramatic energy and latency overhead.

BP-free training methods, especially stochastic zeroth-order (ZO) optimization (Nesterov & Spokoiny, 2017; Liu et al., 2020) or forward-forward method (Hinton, 2022), are easier to implement on edge hardware, since they do not need to detect or save any intermediate states (Gu et al., 2020; 2021a; Momeni et al., 2023; Oguz et al., 2023). However, the scalability issue remains in *end-to-end* training, as these methods typically have a dimension-dependent gradient estimation error, thus suffer from slow or even no convergence on realistic-size PINNs with hundreds of neurons per layer. ZO training shows great success in fine-tuning large language models (LLMs) (Malladi et al., 2023; Yang et al., 2024a; Zhang et al., 2024; Gautam et al., 2024), since the gradient of a well pre-trained LLM has a low intrinsic dimension on fine-tuning tasks. Unfortunately, such a low-dimensional structure does not exist in end-to-end training, preventing the convergence of ZO optimization in training realistic PINNs. Gu et al. (2020; 2021a) utilized ZO training on a photonic chip, but it only fine-tuned a small portion of model parameters based on an offline pre-trained model.

Different from the recent work of fine-tuning (Gu et al., 2020; 2021a; Malladi et al., 2023; Yang et al., 2024a; Zhang et al., 2024; Gautam et al., 2024), we investigate **end-to-end BP-free training of real-size PINNs on photonic chips from scratch**. This is a more challenging task because of (1) the differential operators in the PINN loss evaluation, and (2) the large number of optimization variables that cause divergence in end-to-end ZO training, (3) the scalability issue and lack of photonic memory on current photonic chips. This paper presents, for the first time, a **real-size** and **real-time** photonic PINN training accelerator, which can train a PINN with hundreds of neurons per layer on an integrated photonic platform. **Our novel contributions are summarized as follows:**

- **Two-Level BP-free PINN Training.** We present novel BP-free approaches in two implementation levels of PINN training. Firstly, we propose a sparse-grid Stein estimator to calculate the (high-order) derivative terms in PINN loss evaluations. Secondly, we propose a tensor-compressed variance reduction approach to improve the convergence of ZO-SGD for PINN model parameter updates. These innovations can completely by-pass the need of photonic memory, and greatly improve the convergence of on-chip BP-free training.

- **A Scalable Photonic Design.** We present a highly scalable and easy-to-implement photonic accelerator design. Our design reuses a tensorized ONN inference accelerator, and just add a digital control system to implement on-chip BP-free training. We present two designs: one implements the whole model on a single chip, and another uses a single photonic tensor core with time multiplexing. Our design can scale up to train real-size PINNs with hundreds of neurons per layer.

- **Numerical Experiments and Hardware Emulation.** We validate our method in solving a low-dimensional Black-Scholes equation and a high-dimensional Hamilton-Jacobi-Bellman (HJB) equation. Our two-level BP-free PINN training achieves a competitive error compared to standard PINN training with BP, and achieves the lowest error compared with previous photonic on-chip training methods. We further evaluate the performance of our photonic training accelerator on solving Black-Scholes. The simulation results show that our design can reduce the number of MZIs by $42.7\times$, with only 1.64 seconds to solve this equation.

To our best knowledge, this is the first real-size optical PINN training framework that can be applied to solve realistic PDEs. Our approach shows the great promise of photonic computing in solving

AI-based scientific computing problems. Our results can also be easily extended to solve image and speech problems on photonic and other types of edge platforms.

## 2 BACKGROUND

This section introduces the necessary background of Physics-Informed Neural Networks (PINN) as well as Optical Neural Networks (ONN).

**Physics-Informed Neural Networks (PINNs).** Consider a generic PDE:

$$
\begin{aligned}
\mathcal{N}[\boldsymbol{u}(\boldsymbol{x},t)] &= l(\boldsymbol{x},t), & \boldsymbol{x} \in \Omega,\ t \in [0,T], \\
\mathcal{I}[\boldsymbol{u}(\boldsymbol{x},0)] &= g(\boldsymbol{x}), & \boldsymbol{x} \in \Omega, \\
\mathcal{B}[\boldsymbol{u}(\boldsymbol{x},t)] &= h(\boldsymbol{x},t), & \boldsymbol{x} \in \partial\Omega,\ t \in [0,T],
\end{aligned}
\tag{1}
$$

where $\boldsymbol{x}$ and $t$ are the spatial and temporal coordinates; $\Omega \subset \mathbb{R}^D$, $\partial\Omega$ and $T$ denote the spatial domain, domain boundary and time horizon, respectively; $\mathcal{N}$ is a general nonlinear differential operator; $\mathcal{I}$ and $\mathcal{B}$ represent the initial (or terminal) and boundary condition; $\boldsymbol{u} \in \mathbb{R}^n$ is the solution for the PDE described above. In the contexts of PINNs (Raissi et al., 2019), a solution network $\boldsymbol{u}_{\boldsymbol{\theta}}(\boldsymbol{x},t)$, parameterized by $\boldsymbol{\theta}$, is substituted into PDE equation 1, resulting in a residual defined as:

$$
r_{\boldsymbol{\theta}}(\boldsymbol{x},t) := \mathcal{N}[\boldsymbol{u}_{\boldsymbol{\theta}}(\boldsymbol{x},t)] - l(\boldsymbol{x},t).
\tag{2}
$$

The parameters $\boldsymbol{\theta}$ can be trained by minimizing the loss:

$$
\mathcal{L}(\boldsymbol{\theta}) = \mathcal{L}_r(\boldsymbol{\theta}) + \lambda_0 \mathcal{L}_0(\boldsymbol{\theta}) + \lambda_b \mathcal{L}_b(\boldsymbol{\theta}).
\tag{3}
$$

Here

$$
\begin{aligned}
&\mathcal{L}_r(\boldsymbol{\theta}) = \frac{1}{N_r} \sum_{i=1}^{N_r} \left\| r_{\boldsymbol{\theta}}(\boldsymbol{x}_{\boldsymbol{r}}^i, t_r^i) \right\|_2^2, \quad
\mathcal{L}_0(\boldsymbol{\theta}) = \frac{1}{N_0} \sum_{i=1}^{N_0} \left\| \mathcal{I}[\boldsymbol{u}_{\boldsymbol{\theta}}(\boldsymbol{x}_{\boldsymbol{0}}^i, 0)] - g(\boldsymbol{x}_{\boldsymbol{0}}^i) \right\|_2^2, \\
&\mathcal{L}_b(\boldsymbol{\theta}) = \frac{1}{N_b} \sum_{i=1}^{N_b} \left\| \mathcal{B}[\boldsymbol{u}_{\boldsymbol{\theta}}(\boldsymbol{x}_{\boldsymbol{b}}^i, t_b^i)] - h(\boldsymbol{x}_{\boldsymbol{b}}^i, t_b^i) \right\|_2^2
\end{aligned}
\tag{4}
$$

are the residuals of the PDE, the initial (or terminal) condition and boundary condition, respectively.

**Zeroth-Order (ZO) Optimization.** We consider the minimization of a loss function $\mathcal{L}(\boldsymbol{\theta})$ by updating the model parameters $\boldsymbol{\theta} \in \mathbb{R}^d$ iteratively using a (stochastic) gradient descent method:

$$
\boldsymbol{\theta}_t \leftarrow \boldsymbol{\theta}_{t-1} - \alpha \boldsymbol{g}
\tag{5}
$$

where $\boldsymbol{g}$ denotes the (stochastic) gradient of the loss $\mathcal{L}$ w.r.t. model parameters $\boldsymbol{\theta}$. ZO optimization uses a few forward function queries to approximate the gradient $\boldsymbol{g}$:

$$
\boldsymbol{g} \approx \hat{\nabla}_{\boldsymbol{\theta}} \mathcal{L}(\boldsymbol{\theta}) = \sum_{i=1}^{N} \frac{1}{N\mu} \left[ \mathcal{L}(\boldsymbol{\theta} + \mu \boldsymbol{\xi}_i) - \mathcal{L}(\boldsymbol{\theta} - \mu \boldsymbol{\xi}_i) \right] \boldsymbol{\xi}_i.
\tag{6}
$$

Here $\{\boldsymbol{\xi}_i\}_{i=1}^N$ are some perturbation vectors and $\mu$ is the sampling radius, which is typically small. We consider the random gradient estimator (RGE), in which $\{\boldsymbol{\xi}_i\}_{i=1}^N$ are $N$ i.i.d. samples drawn from a distribution $\rho(\boldsymbol{\xi})$ with zero mean and unit variance (e.g., a multivariate Gaussian distribution or Rademacher distribution). The expectation of $\hat{\nabla}_{\boldsymbol{\theta}} \mathcal{L}$ is unbiased w.r.t. the gradient of the smoothed function $f_\mu(\boldsymbol{x}) := \mathbb{E}_{\boldsymbol{\xi} \sim \rho(\boldsymbol{\xi})}[f(\boldsymbol{x} + \mu\boldsymbol{\xi})]$, however biased w.r.t. the true gradient $\nabla_{\boldsymbol{\theta}} \mathcal{L}$ (Berahas et al., 2022). The variance of RGE involves a dimension-dependent factor $O(d/N)$ given $\mu = O(1/\sqrt{N})$ (Liu et al., 2020). ZO optimization has been used extensively in signal processing and adversarial machine learning (Ghadimi & Lan, 2013; Duchi et al., 2015; Lian et al., 2016; Chen et al., 2019; Shamir, 2017; Cai et al., 2021). A detailed survey was provided in Liu et al. (2020). Recently ZO optimizaiton has achieved great success in fine-tuning LLMs (Malladi et al., 2023; Yang et al., 2024a; Zhang et al., 2024; Gautam et al., 2024), due to low intrinsic dimensionality (e.g., around 300) of the gradient information. Without the low-dimensional structures, ZO optimization scales poorly in end-to-end training of real-size neural networks due to the large dimension-dependent gradient variance. Recently, Chen et al. (2023) improved the scalability of ZO end-to-end training by exploiting model sparsity, but its coordinate-wise gradient estimation is prohibitively expensive for edge devices or real-time applications.

**Optical Neural Networks (ONN) and On-chip ONN Training.** Photonic AI accelerators are expected to outperform their electronic counterparts due to the low latency, ultra-high throughput, high energy efficiency, and high parallelism (McMahon, 2023). Many optical inference accelerators have been reported, such as the MZI meshes (Shen et al., 2017; Clements et al., 2016), microring resonator (MRR) weight banks (Tait et al., 2016), MRR crossbar (Ohno et al., 2022), directional coupler crossbar (Feldmann et al., 2021), balanced homodyne detection (Hamerly et al., 2019), and integrated chip diffractive neural network (Zhu et al., 2022). Due to the limited scalability, the state-of-the-art photonic AI accelerator can only handle weight matrices of size 64×64 (Ramey, 2020). As a result, large-scale optical matrices are computed by tiles or blocks with time multiplexing, demanding intensive memory access to store the intermediate data. That means E/O and O/E conversions and DAC/ADCs are involved during memory access. Demirkiran et al. (2023) shows that only ∼10% of the overall power is consumed in the optical devices. On-chip training is essential to mitigate the significant performance degradation of applying a pre-trained model on non-ideal phototonic chips. Existing on-chip training algorithms include brute-force phase tuning (Shen et al., 2017), neuroevolution (Zhang et al., 2019), and an adjoint variable method which requires optical power monitoring inside each device. The primary issue with the above methods is that there is no access to intermediate states or full gradients on the photonic chip. Several BP-free methods are proposed to circumvent the "hardware-unfriendly" nature of error feedback in BP (Gu et al., 2020; 2021a; Filipovich et al., 2022; Buckley & McCaughan, 2022; Oguz et al., 2023). However, these methods are limited by the small number of training parameters they can handle.

# 3 TWO-LEVEL BP-FREE TRAINING FOR PINNs

Current PINN training methodologies heavily rely on BP for both loss evaluations (Eq. (3)) and gradient-descent model parameter updates (Eq. (5)). These BP computations are hard to implement on photonic chips. This section proposes a two-level BP-free PINN training framework to avoid such a challenge. We first propose a sparse-grid Stein estimator for BP-free loss evaluation. Then we propose a tensor-compressed ZO optimization for gradient-descent PINN model parameter update. This approach improves the convergence of the training framework as well as the scalability on photonic chips, enabling end-to-end training of real-size PINN with hundreds of neurons per layer.

## 3.1 LEVEL 1: BP-FREE PINN LOSS EVALUATION

### 3.1.1 STEIN DERIVATIVE ESTIMATION

Without loss of generality, for an input $\boldsymbol{x} \in \mathbb{R}^D$ and an approximated PDE solution $\boldsymbol{u_\theta}(\boldsymbol{x}) \in \mathbb{R}^n$ parameterized by $\boldsymbol{\theta}$, we consider the first-order derivative $\nabla_{\boldsymbol{x}} \boldsymbol{u_\theta}$ and Laplacian $\Delta \boldsymbol{u_\theta}$ involved in the loss function of a PINN training. Our implementation leverages the Stein estimator (Stein, 1981). Specifically, we represent the PDE solution $\boldsymbol{u_\theta}(\boldsymbol{x})$ via a Gaussian smoothed model:

$$\boldsymbol{u_\theta}(\boldsymbol{x}) = \mathbb{E}_{\boldsymbol{\delta} \sim \mathcal{N}(\boldsymbol{0}, \sigma^2 \boldsymbol{I})} f_{\boldsymbol{\theta}}(\boldsymbol{x} + \boldsymbol{\delta}), \tag{7}$$

where $f_{\boldsymbol{\theta}}$ is a neural network with parameters $\boldsymbol{\theta}$; $\boldsymbol{\delta} \in \mathbb{R}^D$ is the random noise sampled from a multivariate Gaussian distribution $\mathcal{N}(\boldsymbol{0}, \sigma^2 \boldsymbol{I})$. With this special formulation, the first-order derivative and Laplacian of $\boldsymbol{u_\theta}(\boldsymbol{x})$ can be reformulated as the expectation terms:

$$\nabla_{\boldsymbol{x}} \boldsymbol{u_\theta} = \mathbb{E}_{\boldsymbol{\delta} \sim \mathcal{N}(\boldsymbol{0}, \sigma^2 \boldsymbol{I})} \left[ \frac{\boldsymbol{\delta}}{2\sigma^2} (f_{\boldsymbol{\theta}}(\boldsymbol{x} + \boldsymbol{\delta}) - f_{\boldsymbol{\theta}}(\boldsymbol{x} - \boldsymbol{\delta})) \right],$$

$$\Delta \boldsymbol{u_\theta} = \mathbb{E}_{\boldsymbol{\delta} \sim \mathcal{N}[\boldsymbol{0}, \sigma^2 \boldsymbol{I}]} \left[ f_{\boldsymbol{\theta}}(\boldsymbol{x} + \boldsymbol{\delta}) + f_{\boldsymbol{\theta}}(\boldsymbol{x} - \boldsymbol{\delta}) - 2f_{\boldsymbol{\theta}}(\boldsymbol{x}) \right] \times \frac{\|\boldsymbol{\delta}\|^2 - \sigma^2 D}{2\sigma^4}. \tag{8}$$

In He et al. (2023), the above expectation is computed by evaluating $f_{\boldsymbol{\theta}}(\boldsymbol{x} + \boldsymbol{\delta})$ and $f_{\boldsymbol{\theta}}(\boldsymbol{x} - \boldsymbol{\delta})$ at a set of i.i.d. Monte Carlo samples of $\boldsymbol{\delta}$. However, the Monte-Carlo Stein derivative estimator needs a huge number of (e.g., $> 10^3$) function queries. Therefore, it is highly desirable to develop a more efficient BP-free method for evaluating the derivative terms in the loss function.

### 3.1.2 SPARSE-GRID STEIN DERIVATIVE ESTIMATOR

Now we leverage the sparse grid techniques (Garcke et al., 2006; Gerstner & Griebel, 1998) to significantly reduce the number of function queries in the Stein derivative estimator, while maintaining high accuracy in numerical integration.

To begin, we define a sequence of univariate quadrature rules $V = \{V_l : l \in \mathbb{N}\}$. Here $l$ denotes an accuracy level so that any polynomial function of order $\leq l$ can be exactly integrated with $V_l$. Each rule $V_l$ specifies $n_l$ nodes $N_l = \{\delta_1, \ldots, \delta_{n_l}\}$ and the corresponding weight function $w_l : N_l \to \mathbb{R}$. A univariate quadrature rule $V_k$ for a function $f$ of a random variable $\delta$, can be written as:

$$\int_{\mathbb{R}} f(\delta)p(\delta)\, d\delta \approx V_k[f] = \sum_{\delta_j \in N_k} w_k(\delta_j)f(\delta_j). \tag{9}$$

Here $p(\delta)$ is the probability density function (PDF) of $\delta$.

Next, we consider the multivariate integration of a function $f$ over a random vector $\boldsymbol{\delta} = (\delta^1, \ldots, \delta^D)$. We denote the joint PDF of $\boldsymbol{\delta}$ as $p(\boldsymbol{\delta}) = \prod_{m=1}^{D} p(\delta^m)$ and define the $D$-variate quadrature rule with potentially different accuracy levels in each dimension indicated by the multi-index $\boldsymbol{l} = (l_1, l_2, ..., l_D) \in \mathbb{N}^D$. We use Smolyak algorithm (Gerstner & Griebel, 1998) to construct sparse grids. This combines full tensor-product grids of different accuracy levels, removing redundant points. Specifically, for any non-negative integer $q$, define $\mathbb{N}_q^D = \left\{ \boldsymbol{l} \in \mathbb{N}^D : \sum_{m=1}^{D} l_m = D + q \right\}$ and $\mathbb{N}_q^D = \emptyset$ for $q < 0$. The level-$k$ Smolyak rule $A_{D,k}$ for $D$-dim integration can be written as (Wasilkowski & Wozniakowski, 1995):

$$A_{D,k}[f] = \sum_{q=k-D}^{k-1} (-1)^{k-1-q} \binom{D-1}{k-1-q} \times \sum_{\boldsymbol{l} \in \mathbb{N}_q^D} (V_{l_1} \otimes \cdots \otimes V_{l_D})\,[f]. \tag{10}$$

It follows that:

$$A_{D,k}[f] = \sum_{q=k-D}^{k-1} \sum_{\boldsymbol{l} \in \mathbb{N}_q^D} \sum_{\delta^1 \in N_{l_1}} \cdots \sum_{\delta^D \in N_{l_D}} (-1)^{k-1-q} \binom{D-1}{k-1-q} \prod_{m=1}^{D} w_{l_m}(\delta^m)f(\delta^1, \ldots, \delta^D),$$

which is a weighted sum of function evaluations $f(\boldsymbol{\delta})$ for $\boldsymbol{\delta} \in \bigcup_{q=k-D}^{k-1} \bigcup_{\boldsymbol{l} \in \mathbb{N}_q^D} (N_{l_1} \times \cdots \times N_{l_D})$.

The corresponding weight is $(-1)^{k-1-q} \binom{D-1}{k-1-q} \prod_{m=1}^{D} w_{l_m}(\delta^m)$. For the same $\boldsymbol{\delta}$ that appears multiple times for different combinations of values of $\boldsymbol{l}$, we only need to evaluate $f$ once and sum up the respective weights beforehand. The resulting level-$k$ sparse quadrature rule defines a set of $n_L$ nodes $S_L = \{\boldsymbol{\delta}_1, \ldots, \boldsymbol{\delta}_{n_L}\}$ and the corresponding weights $\{w_1, \ldots, w_{n_L}\}$. The $D$-dim integration can then be efficiently computed with the sparse grids as:

$$\int_{\mathbb{R}^D} f(\boldsymbol{\delta})p(\boldsymbol{\delta})d\boldsymbol{\delta} \approx A_{D,k}[f] = \sum_{j=1}^{n_L} w_j f(\boldsymbol{\delta}_j). \tag{11}$$

In practice, since the sparse grids and the weights do not depend on $f$, they can be pre-computed for the specific quadrature rule, dimension $D$, and accuracy level $k$.

Finally, we implement the Stein derivative estimator in Eq. (8) via the sparse-grid integration. Noting that $\boldsymbol{\delta} \sim \mathcal{N}\left(\mathbf{0}, \sigma^2 \boldsymbol{I}\right)$, we can use univariate Gaussian quadrature rules as basis to construct a level-$k$ sparse Gaussian quadrature rule $A_{D,k}^*$ for $D$-variate integration. Then the first-order derivative and Laplacian in Eq. (8) is approximated as:

$$\nabla_{\boldsymbol{x}} \boldsymbol{u_\theta} \approx \sum_{j=1}^{n_L^*} w_j^* \left[ \frac{\boldsymbol{\delta}_j^*}{2\sigma^2} (f_{\boldsymbol{\theta}}(\boldsymbol{x} + \boldsymbol{\delta}_j^*) - f_{\boldsymbol{\theta}}(\boldsymbol{x} - \boldsymbol{\delta}_j^*)) \right],$$

$$\Delta \boldsymbol{u_\theta} \approx \sum_{j=1}^{n_L^*} w_j^* \left( \frac{\|\boldsymbol{\delta}_j^*\|^2 - \sigma^2 D}{2\sigma^4} \right) \times \left( f_{\boldsymbol{\theta}}(\boldsymbol{x} + \boldsymbol{\delta}_j^*) + f_{\boldsymbol{\theta}}(x - \boldsymbol{\delta}_j^*) - 2f_{\boldsymbol{\theta}}(\boldsymbol{x}) \right), \tag{12}$$

where the node $\boldsymbol{\delta}_j^*$ and weight $w_j^*$ are defined by the sparse grid $A_{D,k}^*$.

**Remark:** With the sparse-grid Stein estimator in Eq. (12), we can compute the derivatives in Eq. (2) and the overall PINN loss in Eq. (3) without using any BP computation. Note that $n_L^*$ is usually significantly smaller than the number of Monte Carlo samples required to evaluate Eq. (8). For instance, a level-3 sparse-grid Gaussian quadrature for a 3-dim PDE requires only 25 function evaluations, compared to thousands in Monte Carlo estimation, offering substantial computational savings while maintaining accuracy.

Figure 1: Illustration of tensor-train decomposition. A large weight matrix $\mathbf{W}$ is first folded to a multi-way tensor $\mathcal{W}$, then decomposed into $L$ smaller tensor-train cores $\{\mathcal{G}_k\}_{k=1}^L$.

## 3.2 LEVEL 2: TENSOR-COMPRESSED ZO TRAINING

To avoid BP in the PINN model parametr update, now we leverage the ZO gradient estimator in Eq (6) to perform gradient-descent iteration. Considering the inquiry complexity, we consider randomized gradient estimation only to implement (6). In this case, the gradient mean squared approximation error scales with the perturbation dimension $d$ (Berahas et al., 2022): $\mathbb{E}\left[\|\hat{\nabla}_\theta \mathcal{L}(\theta) - \nabla_\theta \mathcal{L}(\theta)\|_2^2\right] = O\left(\frac{d}{N}\right) \|\nabla_\theta \mathcal{L}(\theta)\|_2^2 + O\left(\frac{\mu^2 d^3}{N}\right) + O\left(\mu^2 d\right)$. Consequently, the convergence rate also scales with $d$ as $O(\sqrt{d}/\sqrt{T})$ in non-convex unconstrained optimization (Berahas et al., 2022). Real-size PINNs typically have hundreds of neurons per hidden layer, and the total number of model parameters can easily exceed $10^5$ or $10^6$. As a result, ZO optimization converges slowly or even fail to converge in end-to-end PINN training.

### 3.2.1 TENSOR-COMPRESSED ZO OPTIMIZATION.

To improve the scalability of ZO training, we propose to significantly reduce the dimensionality and thus gradient variance via a *low-rank* tensor-compressed training as shown in Fig. 1. Let $\mathbf{W} \in \mathbb{R}^{M \times N}$ be a generic weight matrix in a PINN. We factorize its dimension sizes as $M = \prod_{i=1}^L m_i$ and $N = \prod_{j=1}^L n_j$, fold $\mathbf{W}$ into a 2L-way tensor $\mathcal{W} \in \mathbb{R}^{m_1 \times m_2 \times \cdots \times m_L \times n_1 \times n_2 \times \cdots \times n_L}$, and parameterize $\mathcal{W}$ with the tensor-train (TT) decomposition (Oseledets, 2011):

$$\mathcal{W}(i_1, i_2, \ldots, i_L, j_1, j_2, \ldots, j_L) \approx \prod_{k=1}^L \mathbf{G}_k(i_k, j_k) \tag{13}$$

Here $\mathbf{G}_k(i_k, j_k) \in \mathbb{R}^{r_{k-1} \times r_k}$ is the $(i_k, j_k)$-th slice of the TT-core $\mathcal{G}_k \in \mathbb{R}^{r_{k-1} \times m_k \times n_k \times r_k}$ by fixing its 2nd index as $i_k$ and 3rd index as $j_k$. The vector $(r_0, r_1, \ldots, r_L)$ is called TT-ranks with the constraint $r_0 = r_L = 1$. This TT representation reduces the number of unknown variables from $\prod_{k=1}^L m_k n_k$ to $\sum_{k=1}^L r_{k-1} m_k n_k r_k$. The compression ratio can be controlled by the TT-ranks, which can be learnt automatically (Hawkins & Zhang, 2021; Hawkins et al., 2022).

In the ZO training process, we change the trainable variables of each layer from $\mathbf{W}$ to the TT factors $\{\mathcal{G}_k\}_{k=1}^L$. Take a weight matrix with size $512 \times 512$ for example, the original dimension $d = 2.62 \times 10^5$, while the reduced number of variables in TT factors is $d' = 256$ (fold $512 \times 512$ into $8 \times 4 \times 4 \times 4 \times 4 \times 4 \times 4 \times 8$, and set TT-rank as (1,2,2,2,1)). This reduces the problem dimensionality $d$ by $1023\times$, leading to dramatic reduction of the variance in the RGE ZO gradient estimation in Eq. (6). In Table 2, we show that such dimension reduction does little harm to the model learning capacity, but greatly improves the ZO training convergence. In addition, the original matrix-vector product is replaced with low-cost tensor-network contraction in the forward evaluations (Yang et al., 2024b). This offers both memory and computing cost reduction in the ZO training process.

**Comparison and Compatability with other ZO Training.** Some other techniques have also been reported to improve the convergence of ZO training. For instance, model sparsity has exploited (Chen et al., 2023; Liu et al., 2024) to reduce the problem dimensionalty and thus improve the convergence of ZO optimization. Stochastic variance-reduced gradient descent (SVRG) (Johnson & Zhang, 2013) has been extended to ZO optimization (Liu et al., 2018). While these techniques can improve the convergence of ZO training, they cannot reduce the hardware complexity (i.e., the number of photonic devices needed to implement a training accelerator). The ZO SVRG method needs storing previous gradient information for variate control, thus can cause huge memory overhead and is not suitable for photonic implementation. Our method, as will be shown in Section 4, can improve both the convergence and scalability of photonic training accelerators. Our method may also be combined with existing approaches to achieve further better performance. For instance,

model sparsity may be explored at the TT factor level to get further convergence improvement in ZO training. We leave this to our future work.

## 4 DESIGN AND IMPLEMENTATION WITH INTEGRATED PHOTONICS

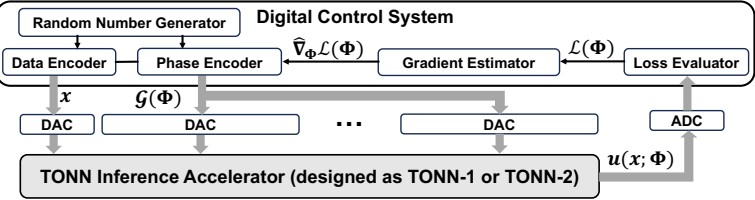

Figure 2: The overall architecture of the BP-free optical training accelerator.

This section presents a design scheme to implement our proposed photonic PINN training accelerator. Due to the BP-free nature, we can reuse a photonic inference accelerator to easily finish the training hardware design. The tensor-compressed ZO training can greatly reduce the number of required photonic devices, providing much better scalability than existing work.

**Overall Architecture.** Figure 2 illustrates the architecture of our optical PINN training accelerator. The training accelerator consists of an optical neural network (ONN) inference accelerator, and an additional digital control system to implement BP-free PINN training. As explained in Appendix A.1, standard ONN (Shen et al., 2017) architecture uses singular value decomposition (SVD) to implement matrix-vector multiplication (MVM), and the resulting unitary matrices are implemented with MZI meshes (Clements et al., 2016). For a $N \times N$ weight matrix, this requires $O(N^2)$ MZIs, which is infeasible for practical PINNs. In contrast, our method utilizes tensor-compressed ZO training, therefore we utilize the tensorized ONN (TONN) accelerator (Xiao et al., 2021) as our inference engine. A TONN inference accelerator only implements the photonic TT-cores $\{\mathcal{G}_k\}_{k=1}^{L}$ instead of the matrix $\mathbf{W}$ on an integrated photonics chip, significantly reducing the number of MZIs required for large-scale layer implementation. The target of on-chip ONN training is to find the optimimal MZI phases $\mathbf{\Phi}$ under various variations. We then implement the tensor-compressed BP-free training by updating the MZI phases $\mathbf{\Phi}_k$ in each photonic TT-core $\mathcal{G}_k(\mathbf{\Phi}_k)$, which has a greatly reduced training dimension compared with updating the matrix $\mathbf{W}(\mathbf{\Phi})$ in a conventional ONN. In the following, we give the details of our TONN design and BP-free training implementation.

**Two Tensorized ONN (TONN) Inference Accelerator Designs.** Here we present two designs for the TONN inference engine: the first design TONN-1 integrates the whole tensor-compressed model on a single chip. The architecture is illustrated in Fig. 3. Each photonic TT-core is implemented by several identical photonic tensor cores. The tensor multiplications between the input data and all TT-cores are realized in a single clock cycle by cascading the photonic TT-cores in the space domain and adding parallelism in the wavelength domain (Xiao et al., 2021).

TONN-1 is "memory-free": no intermediate states need to be stored. The second design TONN-2 (c.f. Fig. 4) uses a single wavelength-parallel photonic tensor core (Xiao et al., 2023) with time multiplexing. Compared with TONN-1, TONN-2 exhibits a smaller footprint at the expense of higher latency and additional memory requirements. In each clock cycle, the photonic tensor core with parallel processing in the wavelength domain is updated to multiply with the input tensor. Then, the intermediate output data is stored in the buffer for the next cycle.

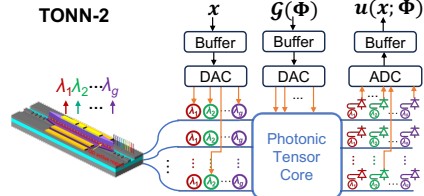

Figure 4: TONN-2 architecture.

**Tensor-compressed BP-free On-chip Training.** BP-free training repeatedly calls the TONN inference engine to evaluate the loss and estimate the gradients, then update the MZI phases. To get the ZO gradient estimation $\hat{\nabla}_{\mathbf{\Phi}}\mathcal{L}(\mathbf{\Phi})$ given by Eq. (6), the digital control system generates Rademacher random perturbations (entries are integers +1 or -1 with equal probability) and re-program the MZIs with the perturbed phase values $\mathbf{\Phi} + \mu\boldsymbol{\xi}$. Here we set $\mu$ as the minimum control resolution of MZI phse tuning. Loss evaluation $\mathcal{L}(\mathbf{\Phi} + \mu\boldsymbol{\xi})$ requires a few inferences with perturbed input data to estimate first- and second-order derivatives by sparse-grid Stein estimator. The digital controller gathers the gradient estimation of $N$ i.i.d. perturbations, and update the MZI phases with $\hat{\nabla}_{\mathbf{\Phi}}\mathcal{L}(\mathbf{\Phi})$.

Figure 3: TONN-1 architecture. PTC: photonic tensor core, DAC: digital-analog converter, ADC: analog-digital converter.

## 5 EXPERIMENTAL RESULTS

To validate our proposed framework, we consider two PDEs: a 1-dim Black-Scholes equation modeling call option price dynamics in financial markets, and a 20-dim Hamilton-Jacobi-Bellman (HJB) equation arising from optimal control of robotics and autonomous systems. The base neural networks are 3-layer MLPs with 128 neurons and `tanh` activation for Black-Scholes, and 512 neurons with `sine` activation for 20-dim HJB. Models are trained for 10,000 iterations using Adam optimizer (learning rate 1e-3), implementing both first-order (FO) and ZO training approaches. FO training uses true gradients computed by BP, while ZO training use RGE gradient estimation. For ZO training, we set query number $N = 1$, smoothing factor $\mu = 0.01$, and use a tensor-wise gradient estimation scheme. (*i.e.,* perturb one tensor and estimate the gradients of that tensor at a time, repeat it sequentially for all tensors, and finally gather all gradients to perform one parameter update step). We evaluate model accuracy on a hold-out set using the relative $\ell_2$ error $\|\hat{u} - u\|^2 / \|u\|^2$ in domain $\Omega$, where $\hat{u}$ is the model prediction and $u$ is the reference solution. The reported results are averaged across 3 different runs. Detailed PDE formulations, experimental settings, and hyperparameter configurations are provided in Appendices A.2 and A.3.

### 5.1 NUMERICAL RESULTS OF SOLVING VARIOUS PDES

We first evaluate the numerical performance of our BP-free PINNs training algorithm. We conduct training in the *weight domain*, where the trainable parameters are the weight matrices $\mathbf{W}$ (tensor cores $\mathcal{G}$ in tensor-compressed training) with tractable gradients to enable FO training as baselines.

**Effectiveness of BP-free Loss Computation:** We consider three methods for computing derivatives in the PINN loss: 1) BP-based method via automatic differentiation (**AD**) as a golden reference, 2) BP-free method via Monte Carlo-based Stein Estimator (**SE**) (He et al., 2023) using 2048 random samples, and 3) our proposed BP-free method via sparse-grid (**SG**). Loss evaluation set-ups are provided in Appendix A.3. We perform FO training on standard PINNs and report the results in Table 1. The BP-free loss computation does not hurt the training performance, and our **SG** method is competitive compared to the original PINN loss

Table 1: Relative $\ell_2$ error of FO training using different loss computation methods.

| Problem | AD | SE | SG (ours) |
|---|---|---|---|
| Black-Scholes | 5.35E-02 | 5.41E-02 | **5.28E-02** |
| 20dim-HJB | 1.99E-03 | 1.52E-03 | **8.16E-04** |

Table 2: Relative $\ell_2$ error achieved using different training methods. The proposed method (TT-compressed ZO training) is underlined.

| Problem | Standard, FO | TT, FO | Standard, ZO | TT, ZO |
|---|---|---|---|---|
| Black-Scholes | 5.28E-02 | **5.97E-02** | 3.91E-01 | **8.30E-02** |
| 20dim-HJB | 8.16E-04 | **2.05E-04** | 6.86E-03 | **1.54E-03** |

evaluation using **AD** while requiring much less forward evaluations than **SE**.

**Evaluation of BP-free PINN Training.** We compare the **FO** training (BP) and **ZO** training (BP-free) in the form of standard (**Std.**) uncompressed and our tensor-compressed (**TT**) formats. We employ the same sparse-grid loss computation for all experiments. Table 2 summarizes the results. **Tensor-compressed training greatly reduces the dimensionality:** For Black-Scholes equation, the dimension of standard training is 17025. The dimension of tensor-compressed training is reduced to 833 (20.44× fewer) by folding the hidden layer as size $4 \times 4 \times 8 \times 8 \times 4 \times 4$ and decomposing with a TT-rank $(1, 2, 2, 1)$. For 20-dim HJB equation, the dimension of standard training is 274433. The

dimension of tensor-compressed training is reduced to 1929 (142.27× fewer) by folding the input layer and the hidden layer as size $1 \times 1 \times 3 \times 7 \times 8 \times 4 \times 4 \times 4$ and $4 \times 4 \times 4 \times 8 \times 8 \times 4 \times 4 \times 4$, respectively. Both the input layer and hidden layer are decomposed with a TT-rank $(1, 2, 2, 2, 1)$.

**TT dimension-reduction does little harm to the accuracy of the PINN model:** The first two columns list the relative $\ell_2$ error achieved after FO training. TT compressed training achieves an error similar to standard training with FC hidden layers. **TT dimension reduction greatly improves the convergence of ZO training:** The last two columns list the relative $\ell_2$ error achieved after ZO training. Fig. 5 shows that standard ZO training fail to converge well due to the high gradient variance which stems from the high dimensionality. By employing TT compressed training to reduce the gradient variance, our ZO training method achieves much better convergence and final accuracy. This showcase that our proposed TT compressed ZO optimization is the key to the success of BP-free training on real-size PINNs. The observations above clearly demonstrates that our method can bypass BP in both loss evaluation and model parameter updates, and still capable of learning a good solution.

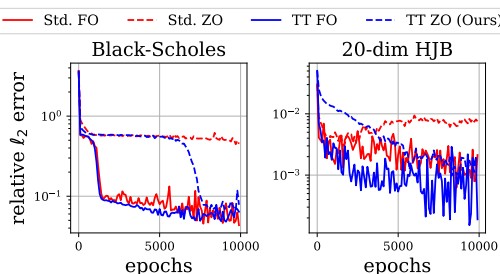

Figure 5: Relative $\ell_2$ error curves of weight domain training for Black Scholes equation (left) and 20dim-HJB (right) equation, respeciely.

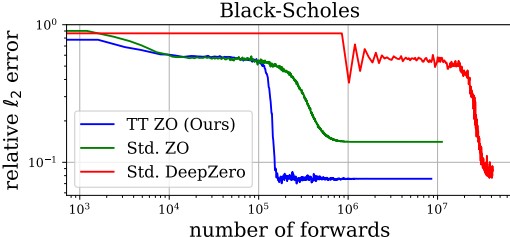

Figure 6: Comparison of ZO training methods.

**Remark.** There is a performance gap between ZO training and FO training, due to the additional variance term of ZO gradient estimation. While this gap cannot be completely eliminated, it may be narrowed by using more forward passes per iteration in the late training stage to achieve a low-variance ZO gradient [*e.g.,* ZO-RGE with a large N, or coordinate-wise gradient estimatior used in `DeepZero` (Chen et al., 2023)]. Overall, our method is the most computation-efficient to train *from scratch*. As shown in Fig. 6, standard ZO training fails to converge well; `DeepZero` may eventually converge to a good solution, however at the cost of over $200\times$ more forward passes.

## 5.2 HARDWARE PERFORMANCE SIMULATION

We further conduct training in the *phase domain* where the trainable parameters are MZI phases $\boldsymbol{\Phi}$ that parameterize weight matrix $\mathbf{W}(\boldsymbol{\Phi})$ (TT-cores $\boldsymbol{\mathcal{G}}(\boldsymbol{\Phi})$ in our proposed method) to simulate the on-chip ONN training. Simulation codes are implemented with an open-source PyTorch-centric ONN library TorchONN. We follow Gu et al. (2021b) to consider the following hardware-restricted objective $\boldsymbol{\Phi}^* = \arg\min_{\boldsymbol{\Phi}} \mathcal{L}(\boldsymbol{W}(\boldsymbol{\Omega}\boldsymbol{\Gamma}\mathcal{Q}(\boldsymbol{\Phi}) + \boldsymbol{\Phi}_b))$, which jointly considers control resolution limit $\mathcal{Q}(\cdot)$, phase-shifter $\gamma$ coefficient drift $\boldsymbol{\Gamma} \sim \mathcal{N}(\gamma, \sigma_\gamma^2)$ caused by fabrication variations, thermal cross-talk between adjacent devices $\boldsymbol{\Omega}$, and phase bias due to manufacturing error $\boldsymbol{\Phi}_b \sim \mathcal{U}(0, 2\pi)$. Detailed set-up is provided in Appendix A.1.3.

**Training Performance.** Table 3 compares our method with existing on-chip BP-free ONN training methods, including `FLOPS` (Gu et al., 2020) and subspace training `L`$^2$`ight` (Gu et al., 2021b). Note that previous methods do not support PINN training. We apply the same sparse-grid loss computation in all methods. We use the same number of ONN forward evaluations per step in different BP-free training methods for fair comparisons. The first two subfigures in Fig. 7 shows the relative $\ell_2$ error curves of different training protocols. `FLOPS` can only handle toy-size neural networks ($20 \sim 30$ neurons per layer, $\sim 1000$ parameters) and fail to converge well on real-size PINNs, thus is not capable of solving realistic PDEs due to the limited scalability. Subspace BP training method `L`$^2$`ight` enables on-chip FO training of ONN, however the trainable parameters are restricted to the diagonal matrix $\boldsymbol{\Sigma}(\boldsymbol{\Phi})$ while orthogonal matrices $\boldsymbol{U}(\boldsymbol{\Phi})$ and $\boldsymbol{V}(\boldsymbol{\Phi})$ are frozen at random initialization due to the intractable gradients. Such restricted learnable space hinders the degree of freedom for training PINNs from scratch. As a result, `L`$^2$`ight` only finds a roughly converged solution with a large relative $\ell_2$ error. Our tensor-compressed BP-free training achieves the lowest relative $\ell_2$ error after on-chip training. We also visualize the learned solution $\hat{u}$ to examine the quality (the

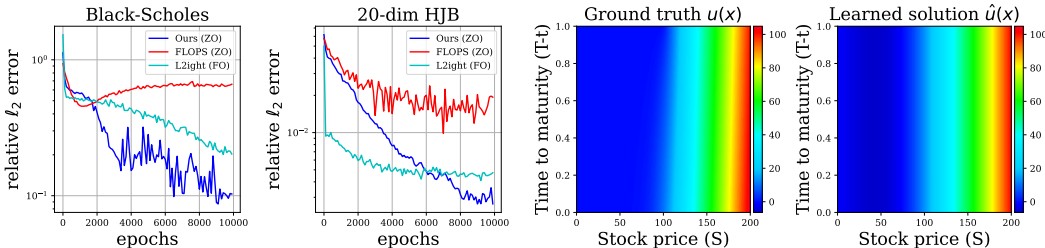

Figure 7: The first two subfigures show the relative $\ell_2$ error of Black-Scholes and 20-dim HJB equations learned by different ONN training methods. The last two subfigures show the ground truth $u(x)$, and the learned solution $\hat{u}(x)$ using our proposed method.

last two subfigures in Fig. 7). Table 3 shows that our method requires much fewer MZIs. The above results show that our method is the most scalable solution to enable real-size PINNs training, capable of solving realistic PDEs on photonic computing hardware. The on-chip phase-domain training results normally show some performance degradation compared with the numerical results of weight-domain training, due to the limited control resolution, device uncertainties, *etc.*.

Table 3: Comparison between different photonic training methods.

|  | Black Scholes | | | 20dim-HJB | | |
|---|---|---|---|---|---|---|
|  | # MZIs | # Trainable MZIs | rel. $\ell_2$ error | # MZIs | # Trainable MZIs | rel. $\ell_2$ error |
| `FLOPS` Gu et al. (2020) | 18,065 | 18,065 | 0.667 | 279,232 | 279,232 | 1.40E-02 |
| `L²ight` Gu et al. (2021b) | 18,065 | 2,561 | 0.203 | 279,232 | 35,841 | 4.09E-03 |
| Ours | **1,685** | 1,685 | **0.103** | **2,057** | 2,057 | **1.57E-03** |

**System Performance.** Table 4 compares the on-chip training system performance to implement a $128 \times 128$ hidden layer for solving Black-Scholes equation. We compare our tensor-compressed ONN (TONN) inference/training accelerator design TONN-1, TONN-2 and the conventional ONN design. It is not practical for a single photonic chip to integrate a matrix as large as $128 \times 128$ due to the huge device footprint and the insurmount-

Table 4: Performance comparison of different methods to implement a $128 \times 128$ hidden layer in solving Black-Scholes equation. The latency means total on-chip training time.

|  | # of MZIs | Footprint (mm$^2$) | Latency (s) |
|---|---|---|---|
| ONN | 16,384 | 3,975.68 | 1.74 |
| TONN-1 | 384 | 102.72 | **1.64** |
| TONN-2 | **64** | **18.72** | 9.80 |

able optical loss due to the quadratic scaling rule. In comparison, our method reduces the number of MZIs by $42.7\times$, which is the key to enable whole-model integration (TONN-1) with a reasonable footprint. The simulation results show that our photonic accelerator achieve ultra-high-speed PINN training (1.64-second training time) to solve the Black-Scholes equation. Detailed breakdown of system performance analysis is provided in Appendix A.4.

## 6 CONCLUSION

This paper has proposed a two-level BP-free training approach to train real-size physics-informed neural networks (PINNs) on optical computing hardware. Specifically, our method integrates a sparse-grid Stein derivative estimator to avoid BP in loss evaluation and a tensor-compressed ZO optimization to avoid BP in model parameter update. The tensor compressed ZO optimization can simultaneously reduce the ZO gradient variance and model parameters, thus scale up ZO training to real-size PINNs with hundreds of neurons per layer. We have further designed the BP-free training on an integrated photonic platform. Our approach has successfully solved a 1-dim Black-Scholes PDE and a 20-dim HJB PDE with smallest relative error compared with existing photonic on-chip training protocols. Future studies of variance reduction can help narrow the performance gap between ZO training and FO training. Our tensor-compressed BP-free training method is not restricted to PINNs. It can be easily extended to solve image and speech problems on photonic and other types of edge platforms where the hardware cost to implement BP is not feasible.

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

## A APPENDIX

link to the anonymous source code repository

### A.1 ONN BASICS

#### A.1.1 MZI-BASED ONN ARCHITECTURE.

We focus on the ONN (Shen et al., 2017) architecture with singular value decomposition (SVD) to implement matrix-vector multiplication (MVM), i.e., $y = Wx = U\Sigma V^* x$. The unitary matrices $U$ and $V^*$ are implemented by MZIs in Clements mesh (Clements et al., 2016). The parametrization of $U$ and $V^*$ is given by $U(\Phi^U) = D^U \prod_{i=k}^{2} \prod_{j=1}^{i-1} R_{ij}(\phi_{ij}^U), V^*(\Phi^V) = D^V \prod_{i=k}^{2} \prod_{j=1}^{i-1} R_{ij}(\phi_{ij}^V)$, where $D$ is a diagonal matrix, and each 2-dimensional rotator $R_{ij}(\phi_{ij})$ can be implemented by a reconfigurable $2 \times 2$ MZI containing one phase shifter ($\phi$) and two 50/50 splitters, which can produce interference of input light signals as follows:

$$\begin{pmatrix} y_1 \\ y_2 \end{pmatrix} = \begin{pmatrix} \cos\phi & \sin\phi \\ -\sin\phi & \cos\phi \end{pmatrix} \begin{pmatrix} x_1 \\ x_2 \end{pmatrix} \tag{14}$$

The diagonal matrix $\Sigma$ is implemented by on-chip attenuators, e.g., single-port MZIs, to perform signal scaling. The parameterization is given by $\Sigma(\Phi^S) = \max(|\Sigma|) \operatorname{diag}(\cdots, \cos\phi_i^S, \cdots)$. We denoted all programmable phases as $\Phi$ and $W$ is parameterized by $W(\Phi) = U(\Phi^U)\Sigma(\Phi^S)V^*(\Phi^V)$.

#### A.1.2 INTRACTABLE GRADIENTS OF MZI PHASES

The analytical gradient w.r.t each MZI phases is given by:

$$\frac{\partial\mathcal{L}}{\partial R_{ij}} = (DR_{n1}R_{n2}R_{n3})^T \nabla_y \mathcal{L} x^T (\cdots R_{32}R_{21}\Sigma V^*)^T \tag{15}$$

$$\frac{\partial\mathcal{L}}{\partial\phi_{ij}} = \operatorname{Tr}\left(\left(\frac{\partial\mathcal{L}}{\partial R_{ij}} \odot \frac{\partial R_{ij}}{\partial\phi_{ij}}\right)(e_i + e_j)(e_i + e_j)^T\right) \tag{16}$$

This analytical gradient is computationally-prohibitive, and requires detecting the whole optical field to read out all intermediate states $x$, which is not practical or scalable on integrated photonics chip.

#### A.1.3 ONN NON-IDEALITY

We follow Gu et al. (2021b) to consider the following ONN non-ideality set-ups in the simulation.

**Limited Phase-tuning Control Resolution.** Given the control resolution limits, we can only achieve discretized MZI phase tuning. We assume the phases $\phi$ is uniformly quantized into 8-bit within $[0, 2\pi]$ for phases in $U(\Phi^U)$, $\Sigma(\Phi^S)$, $V^*(\Phi^V)$.

**Phase-shifter Variation.** We assume the real phase shift $\tilde{\phi} = \frac{\gamma + \Delta\gamma}{\gamma}\phi$, which is proportional to the device-related parameter. We assume $\Delta\gamma \sim \mathcal{N}(0, 0.002^2)$. We formulate this error as a diagonal matrix $\Gamma$ multiplied on the phase shift $\Phi' = \Gamma\Phi$.

**MZI Crosstalk.** The crosstalk effect can be modeled as coupling matrix $\Omega$,

$$\begin{pmatrix} \phi_0^c \\ \phi_1^c \\ \vdots \\ \phi_{N-1}^c \end{pmatrix} = \begin{pmatrix} \omega_{0,0} & \omega_{0,1} & \cdots & \omega_{0,N-1} \\ \omega_{1,0} & \omega_{1,1} & \cdots & \omega_{1,N-1} \\ \vdots & \vdots & \ddots & \vdots \\ \omega_{N-1,0} & \omega_{N-1,1} & \cdots & \omega_{N-1,N-1} \end{pmatrix} \begin{pmatrix} \phi_0^v \\ \phi_1^v \\ \vdots \\ \phi_{N-1}^v \end{pmatrix} \tag{17}$$

$$\text{s.t. } \omega_{i,j} = 1, \quad \forall i = j$$
$$\omega_{i,j} = 0, \quad \forall i \neq j \text{ and } \phi_j \in \mathcal{P}$$
$$0 \leq \omega_{i,j} < 1, \quad \forall i \neq j \text{ and } \phi_j \in \mathcal{A}.$$

The diagonal factor $\omega_{i,j}$, $i = j$ is the self-coupling coefficient, $\omega_{i,j}$, $i \neq j$ is the mutual coupling coefficient. We follow Gu et al. (2021b) to assume the self-coupling coefficient to be 1, and the mutual coupling coefficient is 0.005 for adjacent MZIs.

## A.2 PDE DETAILS

**Black-Scholes Equation.** We examine the Black-Scholes equation for option price dynamics:

$$\partial_t u + \frac{1}{2}\sigma^2 x^2 \partial_{xx} u + rx\partial_x u - ru = 0, \quad x \in [0, 200], \ t \in [0, T],$$
$$u(x, T) = \max(x - K, 0), \quad x \in [0, 200], \tag{18}$$
$$u(0, t) = 0, \quad u(200, t) = 200 - Ke^{-r(T-t)}, \quad t \in [0, T],$$

where $u(x, t)$ is the option price, $x$ is the stock price, $\sigma = 0.2$ is volatility, $r = 0.05$ is risk-free rate, $K = 100$ is strike price, and $T = 1$ is expiration time. The analytical solution is:

$$u(x, t) = xN(d_1) - Ke^{-r(T-t)}N(d_2), \tag{19}$$

with $d_1$ and $d_2$ defined as:

$$d_1 = \frac{\ln(x/K) + (r + \sigma^2/2)(T - t)}{\sigma\sqrt{T - t}},$$
$$d_2 = d_1 - \sigma\sqrt{T - t}, \tag{20}$$

where $N(\cdot)$ is the cumulative distribution function of the standard normal distribution. The base neural network is a 3-layer MLP with 128 neurons and `tanh` activation in each hidden layer. In tensor-train (TT) compressed training, the input layer ($2 \times 128$) and the output layer ($128 \times 1$) are left as-is, while we fold the hidden layer as size $4 \times 4 \times 8 \times 8 \times 4 \times 4$. We preset the TT-ranks as [1,$r$,$r$,1], where $r$ controls the compression ratio.

**20-dim HJB Equation.** We consider the following 20-dim HJB PDE for high-dimensional optimal control:

$$\partial_t u(\boldsymbol{x}, t) + \Delta u(\boldsymbol{x}, t) - 0.05 \|\nabla_{\boldsymbol{x}} u(\boldsymbol{x}, t)\|_2^2 = -2,$$
$$u(\boldsymbol{x}, 1) = \|\boldsymbol{x}\|_1, \quad \boldsymbol{x} \in [0, 1]^{20}, \ t \in [0, 1]. \tag{21}$$

Here $\|\cdot\|_p$ denotes an $\ell_p$ norm. The exact solution is $u(\boldsymbol{x}, t) = \|\boldsymbol{x}\|_1 + 1 - t$. The base network is a 3-layer MLP with 512 neurons and `sine` activation in each hidden layer. For TT compression, we fold the input layer and hidden layers as size $1 \times 1 \times 3 \times 7 \times 8 \times 4 \times 4 \times 4$ and $4 \times 4 \times 4 \times 8 \times 8 \times 4 \times 4 \times 4$, respectively, with TT-ranks [1,$r$,$r$,$r$,1]. The output layer ($512 \times 1$) is left as-is.

## A.3 EXPERIMENTAL SETTINGS

**Loss Evaluation Set-ups.** We compare three methods for computing derivatives in the loss function equation 3: 1) automatic differentiation (**AD**) as a golden reference, 2) Monte Carlo-based Stein Estimator (**SE**) He et al. (2023), and 3) our sparse-grid (**SG**) method. For Black-Scholes, we approximate the solution $u_{\boldsymbol{\theta}}$ using a neural network $f_{\boldsymbol{\theta}}(\boldsymbol{x}, t)$, which can be either the base network or its TT-compressed version. In the **AD** approach, $u_{\boldsymbol{\theta}}(\boldsymbol{x}, t) = f_{\boldsymbol{\theta}}(\boldsymbol{x}, t)$, while for **SE** and **SG**, $u_{\boldsymbol{\theta}}(\boldsymbol{x}, t) = \mathbb{E}_{(\boldsymbol{\delta}_{\boldsymbol{x}}, \delta_t) \sim \mathcal{N}(\boldsymbol{0}, \sigma^2 \boldsymbol{I})} f_{\boldsymbol{\theta}}(\boldsymbol{x} + \boldsymbol{\delta}_{\boldsymbol{x}}, t + \delta_t)$. We set the noise level $\sigma$ to 1e-3 in **SE** and **SG**, using 2048 samples in **SE** and 13 samples in **SG** with a level-3 sparse Gaussian quadrature rule to approximate the expectations equation 7 and equation 8. For HJB, we employ a transformed neural network $f_{\boldsymbol{\theta}}'(\boldsymbol{x}, t) = (1 - t)f_{\boldsymbol{\theta}}(\boldsymbol{x}, t) + \|\boldsymbol{x}\|_1$, where $f_{\boldsymbol{\theta}}(\boldsymbol{x}, t)$ is the base or TT-compressed network. The solution approximation follows the same pattern as in the Black-Scholes case. Here the transformed network is designed to ensure that our approximated solution either exactly satisfies (**AD**) or closely adheres to the terminal condition (**SE**, **SG**), allowing us to focus solely on minimizing the HJB residual during training. We set the noise level $\sigma$ to 0.1 in **SE** and **SG**, using 1024 samples in **SE** and 925 samples in **SG** with a level-3 sparse Gaussian quadrature rule.

**Training Set-ups.** We implemented all methods in PyTorch, utilizing an NVIDIA GTX 2080Ti GPU and an Intel(R) Xeon(R) Gold 5218 CPU @ 2.30GHz.

**Data Sampling.** For Black-Scholes, we uniformly sample 100 random residual points, 10 initial points, and 10 boundary points on each boundary per epoch to evaluate the PDE loss equation 3. For HJB, we select 100 random residual points per epoch. The model architecture for the HJB equation incorporates the terminal condition, eliminating the need for additional terminal loss term.

**ONN Simulation Settings.** We apply the same setups as that in $\mathtt{L^2ight}$ Gu et al. (2021b) to implement uncompressed ONNs in baseline methods $\mathtt{FLOPS}$ Gu et al. (2020) and $\mathtt{L^2ight}$ Gu et al. (2021b). The linear projection in an ONN adopts blocking matrix multiplication, where the $M \times N$ weight matrix is partitioned into $P \times P$ blocks of size $k \times k$. Here $P = \lceil M/k \rceil, Q = \lceil N/k \rceil$. Implementing ONNs with smaller MZI blocks is more practical and robust, and provides enough trainable parameters ($N^2/k$ singular values) for first-order based method $\mathtt{L^2ight}$. We select $k = 8$ for practical consideration.

The weight matrix $\boldsymbol{W}$ is parameterized by MZI phases $\boldsymbol{\Phi}$ as $\boldsymbol{W}(\boldsymbol{\Phi}) = \{\boldsymbol{W}_{pq}(\boldsymbol{\Phi}_{pq})\}_{p=0,q=0}^{p=P-1,q=Q-1}$. Each block $\boldsymbol{W}_{pq}$ is parameterized as $\boldsymbol{W}_{pq}(\boldsymbol{\Phi}_{pq}) = \boldsymbol{U}_{pq}(\boldsymbol{\Phi}_{pq}^U)\boldsymbol{\Sigma}_{pq}(\boldsymbol{\Phi}_{pq}^S)\boldsymbol{V}_{pq}^*(\boldsymbol{\Phi}_{pq}^V)$.

$\mathtt{FLOPs}$ Gu et al. (2020) is a ZO based method. We use zeroth-order gradient estimation to estimate the gradients of all MZI phases (i.e., $\boldsymbol{\Phi}_{pq}^U, \boldsymbol{\Phi}_{pq}^S, \boldsymbol{\Phi}_{pq}^V$)

$\mathtt{L^2ight}$ Gu et al. (2021b) is a subspace FO based method. Due to the intractable gradients for $\boldsymbol{\Phi}_{pq}^U$ and $\boldsymbol{\Phi}_{pq}^V$, only the MZI phase shifters in the diagonal matrix $\boldsymbol{\Phi}_{pq}^S$ are trainable. This restricts the training space (i.e., subspace training).

We follow Gu et al. (2021b) to consider the following hardware-restricted objective $\boldsymbol{\Phi}^* = \arg\min_{\boldsymbol{\Phi}} \mathcal{L}(\boldsymbol{W}(\boldsymbol{\Omega\Gamma}\mathcal{Q}(\boldsymbol{\Phi}) + \boldsymbol{\Phi}_b))$, which jointly considers control resolution limit $\mathcal{Q}(\cdot)$, phase-shifter $\gamma$ coefficient drift $\boldsymbol{\Gamma} \sim \mathcal{N}(\gamma, \sigma_\gamma^2)$ caused by fabrication variations, thermal cross-talk between adjacent devices $\boldsymbol{\Omega}$, and phase bias due to manufacturing error $\boldsymbol{\Phi}_b \sim \mathcal{U}(0, 2\pi)$.

### A.4 System Performance Evaluation

We evaluate the system performance of learning the Black-Scholes equation. The system performance for the accelerators based on ONNs and TONNs are evaluated and compared assuming the III-V-on-Si device platform Liang et al. (2022). The total number of wavelengths used is 8 Xiao et al. (2021). The SVD implementation of the arbitrary matrices is considered in the calculation.

#### A.4.1 Footprint:

Only the footprint of the photonic devices, which occupy the major area of the accelerator, is used for comparison. The photonic footprint includes the areas of hybrid silicon comb laser, microring resonator (MRR) modulator arrays, photonic tensor cores, MRR add-drop filters, photodiodes, and electrical cross-connects.

Table 5: Footprint breakdown. All units are mm$^2$.

|  | Laser | MRR Mod. | Tensor core | Photodetector | Cross-connect | Total |
|---|---|---|---|---|---|---|
| ONN | 25.6 | 1.28 | 3947.52 | 1.28 | / | 3975.68 |
| TONN-1 | 1.6 | 0.8 | 97.92 | 0.8 | 1.6 | 102.72 |
| TONN-2 | 1.6 | 0.4 | 16.32 | 0.4 | / | 18.72 |

#### A.4.2 Latency:

**Latency per Inference.** The latency per inference is calculated by:

$$t_{\text{inference}} = n_{\text{cycle}} * (t_{\text{DAC}} + t_{tuning} + t_{\text{opt}} + t_{\text{ADC}}) \tag{22}$$

where $t_{\text{DAC}}$ is the DAC conversion delay ($\sim$24 ns), $t_{\text{tuning}}$ is the metal-oxide-semiconductor capacitor (MOSCAP) phase shifter tuning delay ($\sim$0.1 ns), $t_{\text{opt}}$ is the propagation latency of optical signal ($\sim$3.20 ns for ONN, $\sim$0.64 ns for TONN-1, and $\sim$0.21 ns for TONN-2), $t_{\text{ADC}}$ is the ADC delay($\sim$24 ns). The TONN-2 uses 6 cycles for one inference, while ONN and TONN-1 only needs

1 cycle. The latency per inference is estimated at 51.30 ns for ONN, 48.74 ns for TONN-1, and 289.86 ns for TONN-2.

**Latency per Epoch.**    The latency per epoch is calculated by:

$$t_{epoch} = (t_{inference} \times N_{point} \times N_{loss} + t_{tuning}) \times N_{grads} + t_{DIG} \tag{23}$$

$t_{\text{DIG}}$ is the digital computation overhead ($\sim$500 ns) for gradient accumulation and phase updates at the end of each epoch. New random perturbation samples could be sampled from environment in parallel with optical inference, so we didn't include this overhead. We use $N_{point} = 130, N_{loss} = 13, N_{grads} = 2$. The latency per epoch is estimated at 0.174 ms for ONN, 0.164 ms for TONN-1, and 0.980 ms for TONN-2.

**Total Training Latency.**    On average our BP-free training finds a good solution after 10000 epochs of update. The total training latency is estimated as 1.74 s for ONN, 1.64 s for TONN-1, and 9.80 s for TONN-2.

Table 6 summarizes the breakdown of training latency.

Table 6: Latency breakdown. The results are based on simulation. ONN-1 and TONN-1 denote space-multiplexing implementation. ONN-2 and TONN-2 denote time-multiplexing implementation.

|  | Latency per inference (ns) | Time per epoch (ms) | Number of epochs | Time to converge (s) | rel. $\ell_2$ error |
|---|---|---|---|---|---|
| ONN-1 | 51.30 | 0.17 | 10,000 | 1.74 | 0.667 |
| ONN-2 | 1545.92 | 5.23 | 10,000 | 52.27 | 0.667 |
| TONN-1 (ours) | 48.74 | 0.16 | 10,000 | 1.64 | **0.103** |
| TONN-2 (ours) | 289.86 | 0.98 | 10,000 | 9.80 | **0.103** |

## A.5    ADDITIONAL EXPERIMENTS ON PDE BENCHMARKS

In addition to the experiments discussed in the main text, we have conducted further evaluations of our method on three additional PDE problems: the one-dimensional Burgers' equation, the two-dimensional Navier-Stokes equation for lid-driven cavity flow, and the two-dimensional Darcy flow problem.

### A.5.1    DEFINITIONS OF PDES

**One-dimensional Burgers' Equation Hao et al. (2023):**

$$\partial_t u + u \partial_x u = \nu \partial_{xx} u, \quad (x, t) \in [-1, 1] \times [0, 1], \tag{24}$$

where the viscosity $\nu = \frac{0.01}{\pi}$. The initial and boundary conditions are:

$$u(x, 0) = -\sin(\pi x), \quad x \in [-1, 1], \tag{25}$$
$$u(-1, t) = u(1, t) = 0, \quad t \in [0, 1]. \tag{26}$$

**Two-dimensional Navier-Stokes Lid-driven Flow Hao et al. (2023):**

$$\mathbf{u} \cdot \nabla \mathbf{u} + \nabla p - \frac{1}{Re} \Delta \mathbf{u} = 0, \quad \mathbf{x} \in [0, 1]^2, \tag{27}$$
$$\nabla \cdot \mathbf{u} = 0, \quad \mathbf{x} \in [0, 1]^2, \tag{28}$$

where $\mathbf{u} = (u, v)$ represents the velocity, $p$ is the pressure and $Re = 100$ is the Reynolds number. The boundary conditions are specified as:

$$\mathbf{u}(\mathbf{x}) = (4x(1 - x), 0), \quad \mathbf{x} \in \Gamma_1, \tag{29}$$
$$\mathbf{u}(\mathbf{x}) = (0, 0), \quad \mathbf{x} \in \Gamma_2, \tag{30}$$
$$p = 0, \quad \mathbf{x} = (0, 0). \tag{31}$$

with the top boundary denoted by $\Gamma_1$ and the left, right, and bottom boundaries by $\Gamma_2$.

**Two-dimensional Darcy Flow Li et al. (2020):**

$$\nabla \cdot (k(\mathbf{x})\nabla u(\mathbf{x})) = f(\mathbf{x}), \quad \mathbf{x} \in \Omega, \tag{32}$$

$$u(\mathbf{x}) = 0, \quad \mathbf{x} \in \partial\Omega, \tag{33}$$

where $k(\mathbf{x})$ is the permeability field, $u(\mathbf{x})$ is the pressure, and $f(\mathbf{x})$ is the forcing function. We define $\Omega = [0, 1]^2$, set $f(\mathbf{x}) = 1$, and use a piecewise constant function for $k(\mathbf{x})$ as shown in Fig. 8.

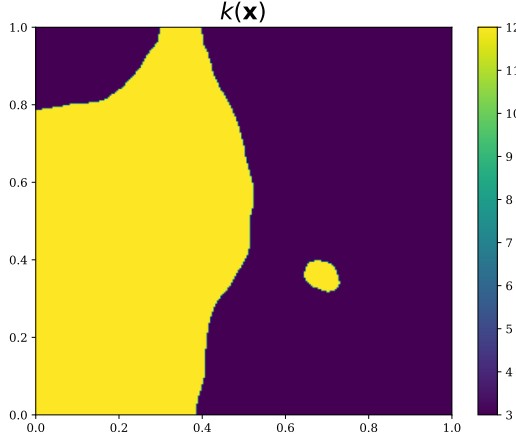

Figure 8: Permeability field in the Darcy flow problem.

### A.5.2 Experimental Results

Our baseline model aligns with the state-of-the-art PINN benchmark from Hao et al. (2023). It consists of a fully connected neural network with five hidden layers, each containing 100 neurons, totaling 30,701 trainable parameters. The dimension of our tensor-compressed training is reduced to 1,241 by folding the weight matrices in hidden layers as size $4 \times 5 \times 5 \times 5 \times 5 \times 4$ and decomposing it with a TT-rank $(1, 2, 2, 1)$. We trained the models for 40,000 iterations on both the Burgers' and Navier-Stokes equations, and for 20,000 iterations on the Darcy flow problem. All other training configurations were kept consistent with our main experimental setups.

The experiment results are provided in Table 7, Table 8, and Table 9. We summarize the findings from our experiments on the three additional PDE benchmarks as follows:

- **BP-free loss computation maintains training performance:** As shown in Table 7, our sparse-grid (SG) method for loss computation is competitive with the original PINN loss evaluation using automatic differentiation (AD), while requiring significantly fewer forward evaluations compared to the Monte Carlo-based Stein Estimator (SE). This indicates that the BP-free loss computation does not compromise training performance.

- **Tensor-Train (TT) dimension reduction improves ZO training convergence:** Table 8 compares first-order (FO) training using backpropagation (BP) and zeroth-order (ZO) training (BP-free) in both standard uncompressed and tensor-compressed (TT) formats. Standard ZO training failed to converge well, whereas our ZO training method with TT dimension reduction achieved much lower relative $\ell_2$ error. This demonstrates that our TT dimension reduction significantly enhances the convergence of ZO training.

- **BP-free training achieves the lowest relative $\ell_2$ error in phase-domain training:** As indicated in Table 9, our method outperforms the ZO method `FLOPS` Gu et al. (2021b), which we attribute to our tensor-train (TT) dimension reduction. Furthermore, our method surpasses the FO method $\text{L}^2\text{ight}$ Gu et al. (2021b); the restricted learnable subspace of $\text{L}^2\text{ight}$ is not capable of training PINNs from scratch. Our BP-free training achieves the lowest relative $\ell_2$ error in phase-domain training.

These results support our claims in the main text. Our method is the most scalable solution to enable real-size PINN training on photonic computing hardware.

**Remark:** Regarding the Navier-Stokes equation, we observed a larger performance gap compared with weight-domain FO training, which serves as the "ideal" upper bound. While the Navier-Stokes PDE is a simple example for traditional non-machine learning PDE solvers, it is a very challenging case for ZO training and photonic computing. Even the FO training requires over 15,000 iterations to converge well (versus 4,000 iterations for the HJB PDE) due to its complicated optimization landscape. All ZO training and photonic training methods failed to achieve good convergence after 40,000 iterations. Among them, our method achieved the best accuracy, with a test relative $\ell_2$ error of $4.82 \times 10^{-1}$ in weight-domain training and $6.99 \times 10^{-1}$ in phase-domain photonic training. Further studies are needed for all tested ZO training and photonic training methods to achieve highly accurate results for the Navier-Stokes PDE. In addition to optimizing the ZO gradient estimation, we may need to consider: (1) optimization frameworks beyond popular SGD/GD, (2) improved PINN architectures, and (3) a deeper understanding of the optimization landscape. ZO training achieved stable convergence for the Black-Scholes and 20-dim HJB equations in the main text and the Darcy flow in additional experiments. The gap between ZO training and FO training is narrowed in the weight-domain training. In phase-domain training simulations, our method also significantly improved over state-of-the-art photonic training methods `FLOPS` Gu et al. (2020) and $\texttt{L}^2\texttt{ight}$ Gu et al. (2021b).

Table 7: Relative $\ell_2$ error of FO training in weight domain using different loss computation methods.

| Problem | AD | SE | SG (ours) |
|---|---|---|---|
| Burgers' | **1.37E-02** | 2.08E-02 | 1.39E-02 |
| Navier-Stokes | 3.79E-02 | 5.34E-02 | **3.66E-02** |
| Darcy flow | 7.25E-02 | 7.39E-02 | **7.07E-02** |

Table 8: Relative $\ell_2$ error of different training methods in weight domain. All experiments use sparse-grid loss computation. The best ZO training results are **bolded**.

| Problem | Standard, FO | TT, FO | Standard, ZO | TT, ZO (ours) |
|---|---|---|---|---|
| Burgers' | 1.39E-02 | 4.82E-02 | 4.47E-01 | **9.50E-02** |
| Navier-Stokes | 3.66E-02 | 6.86E-02 | 5.69E-01 | **4.82E-01** |
| Darcy flow | 7.07E-02 | 7.65E-02 | 2.26E-01 | **8.93E-02** |

Table 9: Relative $\ell_2$ error of phase domain training simulation.

| Problem | `FLOPS` Gu et al. (2020) | $\texttt{L}^2\texttt{ight}$ Gu et al. (2021b) | ours |
|---|---|---|---|
| Burgers' | 4.50E-01 | 5.72E-01 | **2.79E-01** |
| Navier-Stokes | 9.84E-01 | 7.85E-01 | **6.99E-01** |
| Darcy flow | 4.80E-01 | 1.27E-01 | **9.60E-02** |

## A.6 ADDITIONAL EXPERIMENTS OF IMAGE CLASSIFICATION

Our tensor-compressed zeroth-order training is a general back-propagation-free training method that applies to lightweight neural networks other than PINNs. In this section, we extended it to the image classification task on the MNIST dataset. Note that our proposed sparse-grid loss evaluation is designed for PINN training only, so sparse-grid is not used here.

Our baseline model is a two-layer MLP (784×1024, 1024×10) with 814,090 parameters. The dimension of our tensor-compressed training is reduced to 3,962 by folding the input and output layer

as size $7 \times 4 \times 4 \times 7 \times 8 \times 4 \times 4 \times 8$ and $8 \times 4 \times 4 \times 8 \times 1 \times 5 \times 2 \times 1$, respectively. Both the input layer and the output layer are decomposed with a TT-rank $(1, 6, 6, 6, 1)$. Models are trained for 15,000 iterations with a batch size 2,000, using Adam optimizer with an initial learning rate 1e-3 and decayed by 0.8 every 3,000 iterations. In ZO training, we set query number $N = 10$ and smoothing factor $\mu = 0.01$.

Table 10 compares results of weight domain training.

- Our tensor-train (TT) compressed training does not harm the model expressivity, as TT training achieved a similar test accuracy as standard training in first-order (FO) training.
- Our TT compressed training greatly improves the convergence of ZO training and reduces the performance gap between ZO and FO.

Table 11 compares results of phase domain training. Our method outperforms the baseline ZO training method FLOPS Gu et al. (2020). This is attributed to the tensor-train (TT) dimension reduction that reduced gradient variance. Note that the performance gap between phase domain training and weight domain training could be attributed to the low-precision quantization, hardware imperfections, etc., as illustrated in Section 5.2. Our ZO training method did not surpass the FO subspace training method L²ight Gu et al. (2021b). The performance of L²ight versus our method should be considered case by case. L²ight does not have additional gradient errors due to its FO optimization. Meanwhile, its sub-space training can prevent the solver from achieving a good optimal solution. The real performance depends on the trade-off of these two facts. In our PINN experiments, L²ight underperforms our method because the limitation of its sub-space training plays a dominant role. L²ight performs better on the MNIST dataset, probably because the model is more over-parameterized that even subspace training can achieve a good optimal solution.

The results on the MNIST dataset are consistent with our claims in the submission and support our claim that our method can be extended to image problems with higher dimensions.

Table 10: Validation accuracy of weight domain training on MNIST dataset. We report the averaged accuracy and the standard deviation across three runs.

| Method | Standard, FO | TT, FO | Standard, ZO | TT, ZO (ours) |
|---|---|---|---|---|
| Val. Accuracy (%) | 97.83±1.02 | 97.26±0.15 | 83.83±0.44 | 93.21±0.46 |

Table 11: Validation accuracy of phase domain training on MNIST dataset. We report the averaged accuracy and the standard deviation across three runs.

| Method | FLOPS Gu et al. (2020) | L²ight Gu et al. (2021b) | ours |
|---|---|---|---|
| Val. Accuracy (%) | 41.72±5.50 | 95.80±0.48 | 87.91±0.59 |

## A.7 ABLATION STUDIES

### A.7.1 TENSOR-TRAIN (TT) RANKS

To validate our tensor-train (TT) rank choice, we add an ablation study on different TT ranks. The results are provided in Table 12 below. We tested tensor-train compressed training with different TT-ranks on solving 20-dim HJB equations. The model setups are the same as illustrated in Appendix A.2. We fold the input layer and hidden layers as size $1 \times 1 \times 3 \times 7 \times 8 \times 4 \times 4 \times 4$ and $4 \times 4 \times 4 \times 8 \times 8 \times 4 \times 4 \times 4$, respectively, with TT-ranks $[1,r,r,r,1]$. We use automatic differentiation for loss evaluation and first-order (FO) gradient descent to update model parameters. Other training setups are the same as illustrated in Appendix A.3. The results reveal that models with larger TT-ranks have better model expressivity and achieve smaller relative $\ell_2$ error. However, increasing TT-ranks increases the hardware complexity (e.g., number of MZIs) of photonics implementation as it increases the number of parameters. Therefore, we chose a small TT-rank as 2, which provides enough expressivity to solve the PDE equations, while maintaining a small model size.

Table 12: Ablation study on tensor-train (TT) ranks when training the TT compressed model on solving 20-dim HJB equations. We report the average error and the standard deviation across three runs.

| TT-rank | 2 | 4 | 6 | 8 |
|---|---|---|---|---|
| Params | 1,929 | 2,705 | 3,865 | 5,409 |
| rel. $\ell_2$ error | (3.17±1.16)E-04 | (2.45±0.82)E-04 | (4.00±3.69)E-05 | (3.02±3.16)E-05 |

### A.7.2 HIDDEN LAYER WIDTH OF BASELINE MLP MODEL

We also performed an ablation study on the hidden layer width of the baseline MLP model. We trained 3-layer MLPs with different hidden layer widths to solve the 20-dim HJB equation. We use automatic differentiation for loss evaluation and first-order (FO) gradient descent to update model parameters. Other training setups are the same as illustrated in Appendix A.3. The results are shown in Table 13. The MLP model with a smaller hidden layer width leads to larger testing errors. This indicates that a large hidden layer is favored to ensure enough model expressivity. The MLP model used in our submission does not have an overfitting problem.

Table 13: Ablation study on hidden layer size of baseline 3-layer MLP model when learning 20-dim HJB equation. We report the average error and the standard deviation across three runs.

| Hidden layer size | 512 | 256 | 128 | 64 | 32 |
|---|---|---|---|---|---|
| Params | 274,433 | 71,681 | 19,457 | 5,633 | 1,793 |
| rel. $\ell_2$ error | (2.72±0.23)E-03 | (4.31±0.19)E-03 | (7.51±0.36)E-03 | (8.15±0.67)E-03 | (9.25±0.27)E-03 |

### A.8 MORE TABLES OF EXPERIMENTS

In this section, we provide the extended results of Table 1, 2, and 3. Each relative $\ell_2$ error takes the form mean ± std, where mean denotes the averaged result over three independent experiments, and std denotes the corresponding standard deviation.

Table 14: Relative $\ell_2$ error of FO training using different loss computation methods. We report the averaged results and standard deviations across three runs.

| Problem | AD | SE | SG (ours) |
|---|---|---|---|
| Black-Scholes | (5.35±0.13)E-02 | (5.41±0.09)E-02 | **(5.28±0.05)E-02** |
| 20dim-HJB | (1.99±0.15)E-03 | (1.52±0.14)E-03 | **(8.16±1.24)E-04** |

Table 15: Relative $\ell_2$ error achieved using different training methods. We report the averaged results and standard deviations across three runs. The best ZO training results are **bolded**.

| Problem | Standard, FO | TT, FO | Standard, ZO | TT, ZO (ours) |
|---|---|---|---|---|
| Black-Scholes | (5.28±0.05)E-02 | (5.97±0.01)E-02 | (3.91±0.05)E-01 | **(8.30±0.08)E-02** |
| 20dim-HJB | (8.16±1.24)E-04 | (2.05±0.39)E-04 | (6.86±0.27)E-03 | **(1.54±0.35)E-03** |

Table 16: Comparison between different photonic training methods. We report the averaged relative $\ell_2$ error and standard deviations across three runs.

| | Black Scholes | | | 20dim-HJB | | |
|---|---|---|---|---|---|---|
| | # MZIs | # Trainable MZIs | rel. $\ell_2$ error | # MZIs | # Trainable MZIs | rel. $\ell_2$ error |
| `FLOPS` Gu et al. (2020) | 18,065 | 18,065 | 0.663±0.045 | 279,232 | 279,232 | (1.38±0.07)E-02 |
| `L²ight` Gu et al. (2021b) | 18,065 | 2,561 | 0.192±0.381 | 279,232 | 35,841 | (2.95±0.99)E-03 |
| Ours | **1,685** | 1,685 | **0.114±0.095** | **2,057** | 2,057 | **(2.10±0.55)E-03** |

## A.9 MORE FIGURES OF EXPERIMENTS

In this section, we provide the extended results of Fig. 5 and 7. The curves denote averaged relative $\ell_2$ error over three independent experiments and shades denote the corresponding standard deviations.

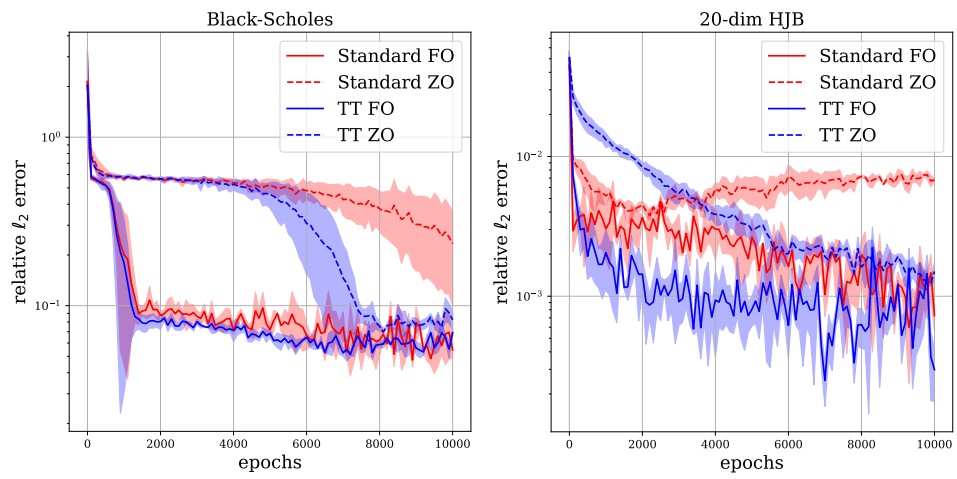

Figure 9: Relative $\ell_2$ error curves of weight domain training for Black-Scholes equation (left) and 20-dim HJB equation (right), respectively. The value at each step is averaged across three runs, and the shade indicates the standard deviation.

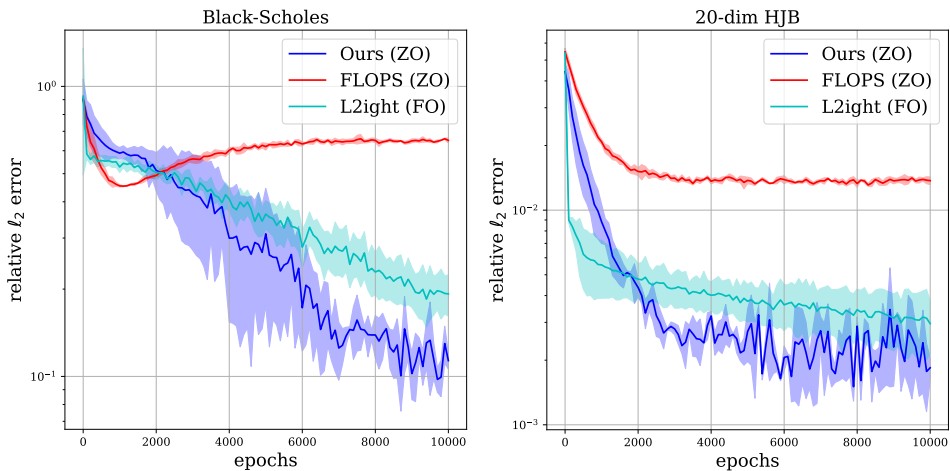

Figure 10: Relative $\ell_2$ error curves of phase domain training for Black-Scholes equation (left) and 20-dim HJB equation (right), respectively. The value at each step is averaged across three runs, and the shade indicates the standard deviation.

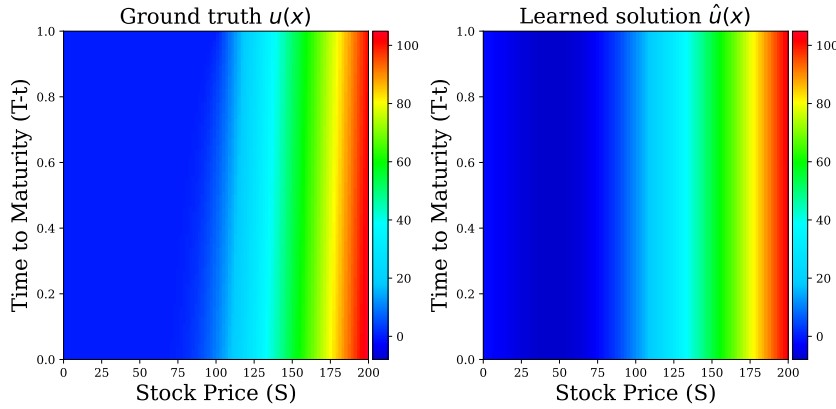

Figure 11: Visualization of Black-Scholes equation in photonic on-chip learning simulation. The left subfigure shows the ground truth $u(x)$, and the right subfigure shows the learned solution $\hat{u}(x)$ using our proposed BP-free PINNs training method.

## A.10 ADDITIONAL DETAILS ON TONN-1 ARCHITECTURE

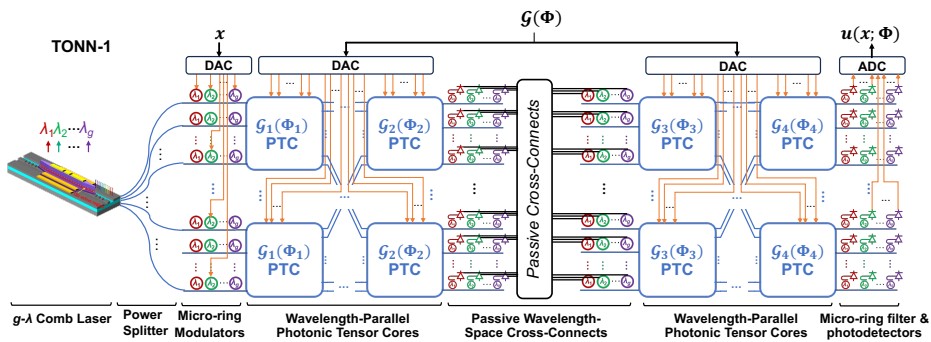

Figure 12: (The same as Figure 3) TONN-1 architecture. PTC: photonic tensor core, DAC: digital-analog converter, ADC: analog-digital converter.

In TONN-1, the input data $\mathbf{x} \in \mathbb{R}^N$, is folded to a d-way tensor $\boldsymbol{\mathcal{X}} \in \mathbb{R}^{N_d \times \cdots \times N_1}$. The indices of the input tensor is then represented by $g$ wavelength division multiplexing (WDM) channels at $N/g$ inputs of the tensor cores, where $g = N_{d/2} \times \ldots \times N_1$. The light source is provided by a $g$-wavelength comb laser and power splitters. The splitted WDM light is modulated by $g$-wavelength optical modulator arrays, then multiplied by each of the photonic tensor core layers, and finally detected by $g$-wavelength WDM microring add-drop filter and detector arrays. The photonic tensor core layer $k$ $(k = d, \ldots, 1, k \neq d/2 + 1)$ consists of $h_k$ number of $R_{k-1}M_k \times N_k R_k$ MZI meshes (tensor cores) and an optical passive cross-connect to switch indices of $M_k$ and $N_{k-1}$. Here, $h_k = M_d \ldots M_{k+1}N_{k1} \ldots N_{d/2+1}$ for $d/2 < k < d$ or $M_{d/2} \ldots M_{k+1}N_{k1} \ldots N_1$ for $k \leq d/2$. For TT-core $d/2 + 1$, the optical passive cross-connect is replaced by a passive wavelength-space cross-connect to switch the indices between the wavelength domain $(N_{d/2}, \ldots, N_1)$ and the space domain $(M_d, \ldots, M_{d/2+1})$.

