# OpenReview forum: "Scalable Back-Propagation-Free Training of Optical Physics-Informed Neural Networks"
_ICLR.cc/2025/Conference — Submitted to ICLR 2025_

### Official Review · Reviewer_qkBf · 2024-10-29

**Soundness:** 3
**Presentation:** 4
**Contribution:** 3
**Rating:** 8
**Confidence:** 3

**Summary:**

This paper mainly discuss the BP-free training scheme of PINNs on photonic devices. The major contributions include 1) sparse grid stein derivative estimator for BP-free loss evaluation. 2) tensor decomposition for better scalability and 3) photonic accelerator design with photonic tensor cores.
Overall, the paper is well written, with a good technique discussion, novelty, and comprehensive experiments. I would recommend proceeding with publication upon addressing some minor comments.

**Strengths:**

1. Two-level BP Free PINN Training with sparse-grid and tensor-compressed variance reduction reduces memory footprint and improves convergence.
2. Hardware design, and emulation with comprehensive performance evaluation.
3. Detailed and clear contents that explain each proposed contribution's underlying idea.

**Weaknesses:**

1. More PDE/ODE case studies might be a plus to demonstrate the generality of the method. e.g. navier-strokes equation. heat spread equations. Also, there could be finer 2D/3D mesh grids in certain applications, and the proposed method scale to these problems is a question mark.
2. I might miss some parts, but It would be better to discuss how the contributions in the paper relate to PINN itself. This is a top-level comment, on why PINN executed on photonic chips. I am trying to understand are the proposed techniques specifically effective on PINNs, or just some general method that can be applicable to any light-weight neural networks. To my understanding, PINN vs general NNs, there would be extra computing of the derivatives described in the governing equations.
3. In 3.2.1 how are the ranks chosen during tensor decomposition?

**Questions:**

Please refer to the weakness section.

---

> ### Author Response · Authors · 2024-11-20
> **Response to Reviewer qkBf (Part I)**
>
> ## Responses to weaknesses:
>
> ***Weakness 1**: More PDE/ODE case studies might be a plus to demonstrate the generality of the method. e.g. navier-strokes equation. heat spread equations. Also, there could be finer 2D/3D mesh grids in certain applications, and the proposed method scale to these problems is a question mark.*
>
> Response: Thank you for your suggestions. We acknowledge the importance of adding more case studies to demonstrate the generality of the method and adding case studies on finer mesh grids to show our method’s scalability in this aspect. We tested our method on 20-dim HJB equation primarily due to its edge applications in autonomous systems. However, **our method can also be applied to solve general PDEs.**
>
> 1. **More PDE/ODE case studies:**
>
> **We have evaluated our methods on another PDE benchmark, one-dimensional Burgers’ equation,** as shown in **Appendix A.5** in our updated manuscript**.** The PDE definitions, baseline model, and training setups are consistent with the state-of-the-art PINN benchmark [R1]. The baseline model is a 5-layer MLP, each layer has a width of 100 neurons. The baseline model has 30701 parameters. The dimension of our tensor-compressed training is reduced to 1241 by folding the weight matrices in hidden layers as size $4\times 5\times 5\times 5\times 5\times 4$ and decomposing it with a TT-rank $(1, 2, 2, 1)$.
>
> The results are provided in **Table R1, R2, R3**.
>
> - **The BP-free loss computation does not hurt the training performance.** Table R1 compares different loss computation methods. Our sparse-grid (**SG**) method is competitive compared to the original PINN loss evaluation using automatic differentiation (**AD**) while requiring much less forward evaluations than Monte Carlo-based Stein Estimator (**SE**).
> - **Our tensor-train (TT) dimension reduction greatly improves the convergence of ZO training**. Table R2 compares the **FO** training (BP) and **ZO** training (BP-free) in the form of **standard** uncompressed and our tensor-compressed (**TT**) formats. Standard ZO training failed to converge well. Our ZO training method achieved a much better final accuracy.
> - **In phase-domain training, our BP-free training achieves the lowest relative $\ell_2$ error.** Our method outperforms ZO method FLOPS [R5]. This is attributed to our tensor-train (TT) dimension reduction**.** Our method also outperforms FO method L2ight [R6]. The restricted learnable subspace of L2ight is not capable of training PINNs from scratch.
>
> **The above results support our claims in the submission.** Our method is the most scalable solution to enable real-size PINNs training on photonic computing hardware. There remains a performance gap compared with weight domain FO training, the “ideal” upper bound. This performance gap is caused by the additional ZO gradient variance, which cannot be completely eliminated but may be further reduced with better variance reduction approaches in the future.
>
> Table R1: Relative $\ell_2$ error of FO training in weight domain when training on baseline 5-layer MLP model using different loss computation methods.
>
> | Problem | AD | SE | SG (ours) |
> | --- | --- | --- | --- |
> | Samples | / | 2048 | 13 |
> | Burgers‘ | 1.37E-02 | 2.08E-02 | 1.39E-02 |
>
> Table R2: Relative $\ell_2$ error of different training method in weight domain.
>
> | Problem | Standard, FO | TT, FO | Standard, ZO | TT, ZO (ours) |
> | --- | --- | --- | --- | --- |
> | Burgers’ | 1.39E-02 | 4.82E-02 | 4.47E-01 | 9.50E-02 |
>
> Table R3: Relative $\ell_2$ error of phase domain training on photonics.
>
> | Problem | FLOPS [R2] | L2ight [R3] | ours |
> | --- | --- | --- | --- |
> | Burgers‘ | 4.50E-01 | 5.72E-01 | 2.79E-01 |
>
> **We also have tried a Navier-Stokes PDE** (2D Navier-Stokes lid-driven flow experiment from SOTA PINN baseline [R2]). Details and results are provided in the following comment (Part II) due to page limit.
>
> [R1] Hao, Zhongkai, et al. "Pinnacle: A comprehensive benchmark of physics-informed neural networks for solving pdes." *arXiv preprint arXiv:2306.08827* (2023).
>
> [R2] Gu, Jiaqi, et al. "FLOPS: Efficient on-chip learning for optical neural networks through stochastic zeroth-order optimization." *2020 57th ACM/IEEE Design Automation Conference (DAC)*. IEEE, 2020.
>
> [R3] Gu, Jiaqi, et al. "L2ight: Enabling on-chip learning for optical neural networks via efficient in-situ subspace optimization." *Advances in Neural Information Processing Systems* 34 (2021): 8649-8661.

---

> ### Author Response · Authors · 2024-11-20
> **Response to Reviewer qkBf (Part II)**
>
> **Response to *Weakness 1* (continue)**
>
> **We also have tried a Navier-Stokes PDE** (2D Navier-Stokes lid-driven flow experiment from SOTA PINN benchmark [R1]). We provide the results in **Table R4, R5, R6**. However, this is a very challenging task: FO training needs over 15000 iterations to converge well (versus 4000 iterations in HJB PDE) due to its complicated optimization landscape. **All ZO training and photonic training fail to achieve good convergence after 40000 iterations. Among them, our method achieves the best accuracy** (4.82E-1 test relative $\ell_2$ error in weight domain training, and 6.99E-01 in phase domain photonic training). We feel that more studies are needed for Navier-Stokes PDE. Besides optimizing the ZO gradient estimation, we may also need to consider (1) optimization framework better than popular SGD/GD, (2) better PINN architecture, (3) deeper understanding of the optimization landscape. We expect that such an intensive study needs significant time beyond the ICLR rebuttal phase.
>
> Table R4: Navier-Stokes results. Relative $\ell_2$ error of FO training in weight domain using different loss computation methods.
>
> | Problem | AD | SE | SG (ours) |
> | --- | --- | --- | --- |
> | Navier-Stokes | 3.79E-02 | 5.34E-02 | 3.66E-02 |
>
> Table R5: Navier-Stokes results. Relative $\ell_2$ error of different training method in weight domain.
>
> | Problem | Standard, FO | TT, FO | Standard, ZO | TT, ZO (ours) |
> | --- | --- | --- | --- | --- |
> | Navier-Stokes | 3.66E-02 | 6.86E-02 | 5.69E-01 | 4.82E-01 |
>
> Table R6: Navier-Stokes results. Relative $\ell_2$ error of phase domain training on photonics.
>
> | Problem | FLOPS [R2] | L2ight [R3] | ours |
> | --- | --- | --- | --- |
> | Navier-Stokes | 9.84E-01 | 7.85E-01 | 6.99E-01 |
>
> 2. **Finer 2D/3D mesh grids:**
>
> **We have added an ablation study focused on our method’s performance under different mesh grid sizes of input samples.** We applied our proposed fully back-propagation-free PINNs training method, that is using sparse-grid loss computation and tensor-train compressed zeroth-order training, to solve Black-Scholes equation. Other experiment setups are the same as our main experiments. As shown in **Table R7**, if a larger input sample mesh grid is allowed, the converged rel. $\ell_2$ error could be slightly reduced.
>
> Table R7: Ablation study on input sample grid size.
>
> | Grid size | $100 \times 100$ | $1000 \times 1000$  | $10000 \times 10000$ |
> | --- | --- | --- | --- |
> | rel. $\ell_2$ error | 8.30E-02 | 7.38E-02 | 7.19E-02 |
>
>
> [R1] Hao, Zhongkai, et al. "Pinnacle: A comprehensive benchmark of physics-informed neural networks for solving pdes." *arXiv preprint arXiv:2306.08827* (2023).
>
> [R2] Gu, Jiaqi, et al. "FLOPS: Efficient on-chip learning for optical neural networks through stochastic zeroth-order optimization." *2020 57th ACM/IEEE Design Automation Conference (DAC)*. IEEE, 2020.
>
> [R3] Gu, Jiaqi, et al. "L2ight: Enabling on-chip learning for optical neural networks via efficient in-situ subspace optimization." *Advances in Neural Information Processing Systems* 34 (2021): 8649-8661.

---

> ### Author Response · Authors · 2024-11-20
> **Response to Reviewer qkBf (Part III)**
>
> ***Weakness 2**: I might miss some parts, but It would be better to discuss how the contributions in the paper relate to PINN itself. This is a top-level comment, on why PINN executed on photonic chips. I am trying to understand are the proposed techniques specifically effective on PINNs, or just some general method that can be applicable to any light-weight neural networks. To my understanding, PINN vs general NNs, there would be extra computing of the derivatives described in the governing equations.*
>
> **Response**: Thank you for raising these insightful questions.
>
> 1. **Contributions in the paper relate to PINN itself:**
>     - Our work directly improves PINNs training by introducing a sparse-grid Stein estimator to efficiently compute high-order derivatives in the PINN loss function without back-propagation. Compared to the Monte Carlo Stein estimator used in PINNs [R4], our method requires significantly fewer function evaluations and achieves higher accuracy. Please refer to our response to **Reviewer Gxiv regarding *Weakness 4*** for a detailed convergence and complexity analysis of both estimators.
> 2. **Why training PINNs on photonic chips:**
>     - Solving PDEs from scratch on edge devices and in real-time is desired in many civil and defense applications. For instance, aerial dynamic analysis of high-speed space vehicles, safety verification and control of robotic and automatic systems.  Due to the extremely tight runtime budget, electronic computing devices fail to meet the requirements. Photonic computing is a promising low-energy and high-speed solution due to the ultra-high operation speed of light. Therefore, **our work seeks to enable real-time PDE solver on edge by realizing PINN training on photonic chips.**
> 3. **Generalizability of our method to other light-weight neural networks:**
>
>     Our tensor-train compressed zeroth-order training method can be generally applied to other applications. **We have applied our tensor-compressed training for image classification task on MNIST dataset,** as shown in **Appendix A.5** in our updated manuscript. Note that our proposed sparse-grid loss evaluation is for PINNs training only, so sparse-grid is not used for image classification.
>
>     The baseline model is a two-layer MLP (784$\times$1024, 1024$\times$10) with 814090 parameters. The dimension of our tensor-compressed training is reduced to 3962 by folding the input and output layer as size $7\times 4\times 4\times 7\times 8\times 4\times 4\times 8$ and $8\times 4\times 4\times 8\times 1\times 5\times 2\times 1$, respectively. Both the input layer and the output layer are decomposed with a TT-rank $(1, 6, 6, 6, 1)$. Models are trained for 15000 iterations with a batch size 2000, using Adam optimizer with initial learning rate 1e-3 and decayed by 0.8 every 3000 iterations. For ZO training, we set query number $N=10$ and smoothing factor $\mu=0.01$.
>
>     The results are shown in Table R8 and R9.
>
>     - Comparing results of weight domain training (**Table R8**):
>         - **Our tensor-train (TT) compressed training does not harm the model expressivity**, as TT training achieved a similar test accuracy as standard training in first-order (FO) training.
>         - **Our TT compressed training greatly improves the convergence of ZO training,** and reduces the performance gap between ZO and FO.
>     - Comparing results of phase domain training (**Table R9**):
>         - **Our method outperforms baseline ZO training method FLOPS**. This is also attributed to the reduced gradient estimation variance. Note that the performance gap between phase domain training and weight domain training could be attributed to the low-precision quantization, hardware imperfections, etc., as illustrated in section 5.2.
>
>     The results on MNIST dataset are consistent with our claims in the submission, and supports our claim that **our method can be extended to other light-weight neural networks.**
>
>     Table R8: Validation accuracy of weight domain training.
>
>     | Method | Standard, FO | TT, FO | Standard, ZO | TT, ZO (ours) |
>     | --- | --- | --- | --- | --- |
>     | Val. Accuracy (%) | 97.83±1.02 | 97.26±0.15 | 83.83±0.44 | 93.21±0.46 |
>
>     Table R9: Validation accuracy of phase domain training.
>
>     | Method | FLOPS [R2] | ours |
>     | --- | --- | --- |
>     | Val. Accuracy (%) | 41.72±5.50 | 87.91±0.59 |
>
> [R2] Gu, Jiaqi, et al. "FLOPS: Efficient on-chip learning for optical neural networks through stochastic zeroth-order optimization." *2020 57th ACM/IEEE Design Automation Conference (DAC)*. IEEE, 2020.
>
> [R3] Gu, Jiaqi, et al. "L2ight: Enabling on-chip learning for optical neural networks via efficient in-situ subspace optimization." *Advances in Neural Information Processing Systems* 34 (2021): 8649-8661.
>
> [R4] He, Di, et al. "Learning physics-informed neural networks without stacked back-propagation." International Conference on Artificial Intelligence and Statistics. PMLR, 2023.

---

> ### Author Response · Authors · 2024-11-20
> **Response to Reviewer qkBf (Part IV)**
>
> ***Weakness 3**: In 3.2.1 how are the ranks chosen during tensor decomposition?*
>
> **Response**: Thank you for raising this question.
>
> 1. In our experiment, we empirically chose the TT-ranks. This is a trade-off between model compression ratio and model expressivity. Alternatively, the TT-ranks can be empirically determined, or adaptively determined by automatic rank determination algorithms [R5, R6].
> 2. To validate our rank choosing, **we have added an ablation study on different TT-ranks, as shown in Appendix A.7.1 in our updated manuscript.** The results are provided in **Table R10** below. We trained tensor-train compressed models with different TT-ranks on solving 20-dim HJB equations. The model setups are the same as illustrated in Appendix A.2, where we fold the input layer and hidden layers as size $1\times 1\times 3\times 7\times 8\times 4\times 4\times 4$ and $4\times 4\times 4\times 8\times 8\times 4\times 4\times 4$, respectively, with TT-ranks [1,$r$,$r$,$r$,1]. We use automatic differentiation for loss evaluation and first-order (FO) gradient descent to update model parameters. Other training setups are the same as illustrated in Appendix A.3. The results reveal that **models with larger TT-ranks have better model expressivity and achieve smaller relative l2 error**. However, **increasing TT-ranks increases the hardware complexity (e.g., number of MZIs) of photonics implementation** as it increases the number of parameters. Therefore, we chose a small TT-rank as 2, which provides enough expressivity to solve the PDE equations, while maintaining a small model size.
>
> Table R10: Ablation study on tensor-train (TT) ranks when training the TT compressed model on solving 20-dim HJB equations.
>
> | TT-rank | 2 | 4 | 6 | 8 |
> | --- | --- | --- | --- | --- |
> | Params | 1,929 | 2,705 | 3,865 | 5,409 |
> | rel. $\ell_2$ error | (3.17±1.16)E-04 | (2.45±0.82)E-04 | (4.00±3.69)E-05 | (3.02±3.16)E-05 |
>
> [R5] Hawkins, Cole, Xing Liu, and Zheng Zhang. "Towards compact neural networks via end-to-end training: A Bayesian tensor approach with automatic rank determination." *SIAM Journal on Mathematics of Data Science* 4.1 (2022): 46-71.
>
> [R6] Yang, Zi, et al. "CoMERA: Computing-and Memory-Efficient Training via Rank-Adaptive Tensor Optimization." *arXiv preprint arXiv:2405.14377* (2024).

---

> ### Comment · Reviewer_qkBf · 2024-11-23
>
> Thanks for the response, I will maintain my score.

---

> > ### Author Response · Authors · 2024-11-24
> > **Thank you for your acknowledgment!**
> >
> > Dear Reviewer qkBf,
> >
> > Thank you for your quick response. We sincerely appreciate your participation in the discussion and your acknowledgment of our work. Your insightful feedback helped a lot to enhance the quality of our work.

---

### Official Review · Reviewer_tnjk · 2024-10-30

**Soundness:** 3
**Presentation:** 3
**Contribution:** 3
**Rating:** 6
**Confidence:** 4

**Summary:**

The authors design a computational training framework to train hardware realizations of all-optical neural networks (ONNs).  As opposed to previous work, which treated the forward pass, the contributions provided in this paper allow for completion of the framework by adding the backward pass via a gradient estimation approach, thus enabling training to occur in hardware. The details of the methodology are designed considering the physical constraints of the optical hardware, namely the large area footprints of MZIs and the difficulty to store memory in the optical domain (which necessitates electro-optical (EO) conversion in other works).

Gradient approximation methods are difficult to scale on larger networks due to dimension-dependent errors. Two novel aspects are thus introduced in this paper to allow this to work:

(1) a sparse-grid Stein derivative estimation method which addresses dimension-dependant gradient estimation error

(2) To further suppress the dimension-dependant gradient estimation error, they suggest a tensor-compressed variance reduction approach to reduce the dimensionality of the problem, thus achieving better convergence

They compare their designs with a standard ONN accelerator. TONN-1 (fully on-chip) reduces both the footprint and latency of the photonic network as compared to previous, standard ONN designs. TONN-2 (where matrices are decomposed into tiles, requiring some storage of intermediate results and thus EO conversion)  further reduces the hardware footprint at the cost of increased latency from re-using the photonic tensor core multiple times and storing intermediate results.

**Strengths:**

The paper is very well written and organized, and information is presented concisely. The results would be useful to other researchers working on software for ONNs. Without being an expert in the field of optical hardware, I find the results novel. Many of my initial questions (for example, the latency calculations) are answered in the appendix.

**Weaknesses:**

Throughout most of the paper, I was also questioning why the focus was on PINNs, and whether there was anything that prevented it from applying more generally. The authors have answered this in the last sentence of the conclusion (although I still think the focus on PINN applications is a bit unnecessary).

The applications demonstrated are on relatively “toy” problems. Epoch vs. loss plots are only shown for one example in each case. Different network sizes are not tested. Overall, the demonstrated results are fairly minimal, only barely enough to show that the method works. More extensive results would be nice to really flesh out the work and show that the method is useful.

**Questions:**

-In Table 1, why does the sparse-grid Stein estimator perform better than automatic differentiation?

---

> ### Author Response · Authors · 2024-11-20
> **Response to Reviewer tnjk (Part I)**
>
> ## Responses to weaknesses:
>
> ***Weakness 1***: *Throughout most of the paper, I was also questioning why the focus was on PINNs, and whether there was anything that prevented it from applying more generally. The authors have answered this in the last sentence of the conclusion (although I still think the focus on PINN applications is a bit unnecessary).*
>
> **Response**: Thank you for raising this insightful question.
>
> We agree that **our tensor-train compressed zeroth-order training method can be generally applied to other applications**. We focus on only PINN in this paper because:
>
> 1. **Our main motivation is to propose a completely back-propagation-free training for PINNs, so as to realize real-time PINNs training (i.e., real-time PDE solver) on photonics computing chips.** Solving PDEs on edge devices and in real-time is desired in many civil and defense applications. For instance, aerial dynamic analysis of high-speed space vehicles, safety verification and control of robotic and automatic systems.  In other words, we need to train PINNs from scratch in real-time. Electronic computing devices fail to meet the requirements.
> 2. Our proposed sparse-grid derivative estimator is only required in PINNs training.
>
> To avoid confusion, we decided to focus only on PINN in this paper. It’s true that our method can also be used to solve other ML problems (e.g., image classification) if the sparse-grid BP-free loss evaluation is not used. We have shown an image classification example on MNIST dataset, as shown in Appendix A.6 in our updated manuscript.
>
> ***Weakness 2***: *The applications demonstrated are on relatively “toy” problems. Epoch vs. loss plots are only shown for one example in each case. Different network sizes are not tested. Overall, the demonstrated results are fairly minimal, only barely enough to show that the method works.*
>
> **Response**:
>
> Thank you for raising this concern.
>
> - **We have added new epoch vs. loss plots in Appendix A.9 in our updated manuscript.** The loss at each step is averaged over 3 independent experiments, and the shades indicate the standard deviation.
> - **We have added ablation studies on different network sizes, as shown in Appendix A.7.** We summarize the results here:
>     - **Table R1** shows the training results of 3-layer MLPs with different hidden layer size to solve the 20-dim HJB equation. We use automatic differentiation for loss evaluation and first-order (FO) gradient descent to update model parameters. Other training setups are the same as illustrated in Appendix A.3. **MLPs with smaller hidden layer size have fewer parameters, but their testing errors are also larger.** A large hidden layer is favored to ensure enough model expressivity.
>     - **Table R2** shows the results of tensor-train compressed training with different TT-ranks on solving 20-dim HJB equations. The model setups are the same as illustrated in Appendix A.2. We fold the input layer and hidden layers as size $1\times 1\times 3\times 7\times 8\times 4\times 4\times 4$ and $4\times 4\times 4\times 8\times 8\times 4\times 4\times 4$, respectively, with TT-ranks [1,$r$,$r$,$r$,1]. We use automatic differentiation for loss evaluation and first-order (FO) gradient descent to update model parameters. Other training setups are the same as illustrated in Appendix A.3. The results reveal that **models with larger TT-ranks ($r$) have better model expressivity and achieve smaller relative $\ell_2$ error.** However, **increasing TT-ranks increases the hardware complexity (e.g., number of MZIs) of photonics implementation** as it increases the number of parameters. Therefore, we chose a small TT-rank as 2, which provides enough expressivity to solve the PDE equations, while maintains a small model size.
>
> Table R1: Ablation study on hidden layer size of baseline 3-layer MLP model when solving 20-dim HJB equation.
>
> | Hidden layer size | 512 | 256 | 128 | 64 | 32 |
> | --- | --- | --- | --- | --- | --- |
> | Params | 274,433 | 71,681 | 19,457 | 5,633 | 1,793 |
> | rel. $\ell_2$ error | (2.72±0.23)E-03 | (4.31±0.19)E-03 | (7.51±0.36)E-03 | (8.15±0.67)E-03 | (9.25±0.27)E-03 |
>
> Table R2: Ablation study on tensor-train (TT) ranks when solving 20-dim HJB equations.
>
> | TT-rank | 2 | 4 | 6 | 8 |
> | --- | --- | --- | --- | --- |
> | Params | 1,929 | 2,705 | 3,865 | 5,409 |
> | rel. $\\ell_2$ error | (3.17±1.16)E-04 | (2.45±0.82)E-04 | (4.00±3.69)E-05 | (3.02±3.16)E-05 |

---

> ### Author Response · Authors · 2024-11-20
> **Response to Reviewer tnjk (Part II)**
>
> ***Weakness 3***: *More extensive results would be nice to really flesh out the work and show that the method is useful.*
>
> **Response:**
>
> Thank you for your suggestions. We acknowledge the importance of extensive results to show that our method is generally useful. We extend our method to another PDE benchmark **one-dimensional Burgers’ equation**, and an **image classification task on MNIST dataset**.
>
> 1. **We have evaluated our methods on another PDE benchmark, one-dimensional Burgers’ equation,** as shown in **Appendix A.5** in our updated manuscript. The PDE definitions, baseline model, and training setups are consistent with the state-of-the-art PINN benchmark [R1]. The baseline model is a 5-layer MLP, each layer has a width of 100 neurons. The baseline model has 30701 parameters. The dimension of our tensor-compressed training is reduced to 1241 by folding the weight matrices in hidden layers as size $4\times 5\times 5\times 5\times 5\times 4$ and decomposing it with a TT-rank $(1, 2, 2, 1)$.
>
> The results are provided in Table R4, R5, R6.
>
> - **The BP-free loss computation does not hurt the training performance.** Table R4 compares different loss computation methods. Our sparse-grid (**SG**) method is competitive compared to the original PINN loss evaluation using automatic differentiation (**AD**) while requiring much less forward evaluations than Monte Carlo-based Stein Estimator (**SE**).
> - **Our tensor-train (TT) dimension reduction greatly improves the convergence of ZO training**. Table R5 compares the **FO** training (BP) and **ZO** training (BP-free) in the form of **standard** uncompressed and our tensor-compressed (**TT**) formats. Standard ZO training failed to converge well. Our ZO training method achieves a much better final accuracy.
> - **In phase-domain training, our BP-free training achieves the lowest relative $\ell_2$ error.** Our method outperforms ZO method FLOPS [R2]. This is attributed to our tensor-train (TT) dimension reduction**.** Our method also outperforms FO method L2ight [R3]. The restricted learnable subspace of L2ight is not capable of training PINNs from scratch.
>
> **The above results support our claims in the submission.** Our method is the most scalable solution to enable real-size PINNs training on photonic computing hardware. Their remains a performance gap compared with weight domain FO training, the “ideal” upper bound. This performance gap is caused by ZO gradient variance, which cannot be completely eliminated but may be further reduced with better variance reduction approaches in the future.
>
> Table R4: Relative $\ell_2$ error of FO training in weight domain using different loss computation methods.
>
> | Problem | AD | SE | SG (ours) |
> | --- | --- | --- | --- |
> | Samples | / | 2048 | 13 |
> | Burgers‘ | 1.37E-02 | 2.08E-02 | 1.39E-02 |
>
> Table R5: Relative $\ell_2$ error of different training method in weight domain.
>
> | Problem | Standard, FO | TT, FO | Standard, ZO | TT, ZO (ours) |
> | --- | --- | --- | --- | --- |
> | Burgers’ | 1.39E-02 | 4.82E-02 | 4.47E-01 | 9.50E-02 |
>
> Table R6: Relative $\ell_2$ error of phase domain training on photonics.
>
> | Problem | FLOPS [R2] | L2ight [R3] | ours |
> | --- | --- | --- | --- |
> | Burgers‘ | 4.50E-01 | 5.72E-01 | 2.79E-01 |
>
> 2. **We have applied our tensor-compressed training for the image classification task on the MNIST dataset,** as shown in **Appendix A.6** in our updated manuscript. Details and results are provided in the following comment (Part III) due to page limit.
>
> [R1] Hao, Zhongkai, et al. "Pinnacle: A comprehensive benchmark of physics-informed neural networks for solving pdes." *arXiv preprint arXiv:2306.08827* (2023).
>
> [R2] Gu, Jiaqi, et al. "FLOPS: Efficient on-chip learning for optical neural networks through stochastic zeroth-order optimization." *2020 57th ACM/IEEE Design Automation Conference (DAC)*. IEEE, 2020.
>
> [R3] Gu, Jiaqi, et al. "L2ight: Enabling on-chip learning for optical neural networks via efficient in-situ subspace optimization." *Advances in Neural Information Processing Systems* 34 (2021): 8649-8661.

---

> ### Author Response · Authors · 2024-11-20
> **Response to Reviewer tnjk (Part III)**
>
> **Response to *Weakness 3* (continue)**
>
> 2. **We have applied our tensor-compressed training for image classification task on MNIST dataset,** as shown in **Appendix A.6** in our updated manuscript. Note that our proposed sparse-grid loss evaluation is for PINNs training only, so sparse-grid is not used for image classification.
>
> The baseline model is a two-layer MLP (784$\times$1024, 1024$\times$10) with 814090 parameters. The dimension of our tensor-compressed training is reduced to 3962 by folding the input and output layer as size $7\times 4\times 4\times 7\times 8\times 4\times 4\times 8$ and $8\times 4\times 4\times 8\times 1\times 5\times 2\times 1$, respectively. Both the input layer and the output layer are decomposed with a TT-rank $(1, 6, 6, 6, 1)$. Models are trained for 15000 iterations with a batch size 2000, Adam optimized with initial learning rate 1e-3 and decayed by 0.8 every 3000 iterations. For ZO training, we set query number $N=10$ and smoothing factor $\mu=0.01$. We have added the details in Appendix A.6 in our updated manuscript.
>
> The results are shown in Table R7 and R8.
>
> - Comparing results of weight domain training (**Table R7**):
>     - **Our tensor-train (TT) compressed training does not harm the model expressivity**, as TT training achieved a similar test accuracy as standard training in first-order (FO) training.
>     - **Our TT compressed training greatly improves the convergence of ZO training,** and reduces the performance gap between ZO and FO.
> - Comparing results of phase domain training (**Table R8**):
>     - Our method outperforms baseline ZO training method FLOPS. This is also attributed to the reduced gradient estimation variance. Note that the performance gap between phase domain training and weight domain training could be attributed to the low-precision quantization, hardware imperfections, etc., as illustrated in section 5.2.
>
> The results on MNIST dataset are consistent with our claims in the submission, and supports our claim that **our method can be extended to image problems with higher dimensions.**
>
> Table R7: Validation accuracy of weight domain training.
>
> | Method | Standard, FO | TT, FO | Standard, ZO | TT, ZO (ours) |
> | --- | --- | --- | --- | --- |
> | Val. Accuracy (%) | 97.83±1.02 | 97.26±0.15 | 83.83±0.44 | 93.21±0.46 |
>
> Table R8: Validation accuracy of phase domain training.
>
> | Method | FLOPS [R2] | ours |
> | --- | --- | --- |
> | Val. Accuracy (%) | 41.72±5.50 | 87.91±0.59 |
>
> We believe that with above extensive results, we can flesh our our work and show that our method is generally useful.
>
>
> ## Responses to questions:
>
> ***Question 1**: why does the sparse-grid Stein estimator perform better than automatic differentiation*
>
> Thank you for raising this insightful question. We use sparse-grid method to calculate the loss, which **smoothen the loss function**. Existing work showed that smoothening the loss function [R4] or loss landscape [R5] could lead to better generalization behavior.
>
> [R2] Gu, Jiaqi, et al. "FLOPS: Efficient on-chip learning for optical neural networks through stochastic zeroth-order optimization." *2020 57th ACM/IEEE Design Automation Conference (DAC)*. IEEE, 2020.
>
> [R3] Gu, Jiaqi, et al. "L2ight: Enabling on-chip learning for optical neural networks via efficient in-situ subspace optimization." *Advances in Neural Information Processing Systems* 34 (2021): 8649-8661.
>
> [R4] Müller, Rafael, Simon Kornblith, and Geoffrey E. Hinton. "When does label smoothing help?." *Advances in neural information processing systems* 32 (2019).
>
> [R5] Rathore, Pratik, et al. "Challenges in training PINNs: A loss landscape perspective." *arXiv preprint arXiv:2402.01868* (2024).

---

> > ### Author Response · Authors · 2024-11-24
> > **Reminder on follow-up discussion (2 day left before rebuttal ends)**
> >
> > Dear Reviewer tnjk,
> >
> > Thank you very much for the dedicated review of our paper. We are fully aware of the commitment and time your review entails. Your efforts are deeply valued by us, and we have worked hard to address all of your comments.
> >
> > With 2 day remaining before the end of the discussion phase, we wish to extend a respectful request for your feedback about our responses. Your insights are of immense importance to us, and we eagerly anticipate your updated evaluation.
> >
> > If you find our responses informative and useful, we would be grateful for your acknowledgement.
> >
> > Meanwhile, if you have any further inquiries or require additional clarifications, please don't hesitate to reach out. We are fully committed to providing additional responses during this discussion phase.
> >
> > Best regards,
> >
> > Authors

---

> > > ### Author Response · Authors · 2024-11-27
> > > **All recent comments are incorporated in the revised manuscript**
> > >
> > > Dear Reviewer tnjk,
> > >
> > > Thanks again for your insightful and valuable review comments.
> > >
> > > Since ICLR has extended the discussion period, we have uploaded a revised manuscript to incorporate all new comments made by the reviewers in the follow-up discussion.
> > >
> > > We fully understand that you might be super busy at this moment. Meanwhile, we would highly appreciate it if you can take a look at our updated manuscript and our technical responses to your questions.
> > >
> > > Thanks, and happy Thanksgiving!
> > >
> > > Best regards,
> > >
> > > The authors.

---

### Official Review · Reviewer_Syoh · 2024-11-02

**Soundness:** 3
**Presentation:** 3
**Contribution:** 3
**Rating:** 6
**Confidence:** 3

**Summary:**

This paper proposes a back-propagation-free (BP-free) method for efficiently training physics-informed neural networks (PINNs) on real-time edge devices. The method focuses on utilizing optical neural networks (ONNs) in photonic computing to solve high-dimensional partial differential equations (PDEs). Traditional PINN training is computationally expensive, especially for real-time training on edge devices. To address this challenge, the authors introduce three main innovations:

1. They utilize a sparse-grid(SG) Stein derivative estimator to calculate derivatives in the loss function without the need for back-propagation. This enables efficient evaluation of the PINN loss function without complex high-order derivative calculations or Automatic differentiation.

2.  They adopt Tensor-Train (TT) decomposition for low-rank tensor compression, and zeroth-order (ZO) optimization to reduce parameter dimensionality and improve scalability and convergence. These allow faster convergence even for high-dimensional PINNs by reducing the computational burden associated with parameter updates.

3. They propose a scalable on-chip photonic PINN accelerator design that utilizes photonic tensor cores to achieve real-time training without back-propagation. This design enhances energy and area efficiency, enabling fast, real-time operation on edge devices compared to conventional GPU-based training.

The paper validates this framework by testing on high-dimensional PDEs such as the Black-Scholes and 20-dim Hamilton-Jacobi-Bellman equations, demonstrating performance improvements through simulations on actual photonic devices.

**Strengths:**

### Strengths

1. The authors identified memory issues that arise during the training of PINNs on traditional photonic chips. To address this, they introduced a BP-free approach instead of the back-propagation method and utilized TT decomposition to reduce memory usage. This enabled real-time, high-dimensional PINN training on photonic chips and improved computational efficiency, accelerating the convergence rate.


2. They proposed two design options that can be applied according to specific needs in photonic computing, enhancing system performance. The on-chip photonic accelerator proposed by the authors efficiently uses existing photonic MAC devices, which occupy a large area, to maintain energy efficiency even in large-scale PINN training. The advantages of this design were experimentally demonstrated compared to conventional ONNs in terms of MZI number, footprint, and latency.

**Weaknesses:**

### Weaknesses

1. Although the authors claimed memory and computational efficiency for their proposed method, details on time per epoch, total convergence time, and number of epochs until convergence were not provided, aside from accuracy in the experiments.

2. Additional experiments on other commonly used PDE datasets, such as Navier-Stokes, Darcy flow, and Burgers' equation, are needed beyond Black-Scholes and 20-dimensional HJB.

3. The authors did not provide an explanation for the criteria used to determine "convergence," which is a key aspect of their experiments. In Figure 5, it appears possible that different convergence criteria could be applied depending on the epoch.

4. The figures presenting the experimental results are too small, and the associated information or explanations are limited.

5. Detailed settings and environment information used in the experiments were not provided (e.g., code, chip information).

**Questions:**

### Questions

1. As mentioned in Weakness 2, are there any experiments conducted on other datasets?

2. As raised in Weakness 3, what criteria did the authors use to determine the point of convergence?

3. Could you provide more information on the efficiency and memory aspects of the experiments? (e.g., memory usage, time per epoch, the time and number of epochs each method took to converge, etc.)

**Details Of Ethics Concerns:**

There is no ethics concerns

---

> ### Author Response · Authors · 2024-11-20
> **Response to Reviewer Syoh (Part I)**
>
> ## Responses to weaknesses:
>
> ***Weakness 1**: Although the authors claimed memory and computational efficiency for their proposed method, details on time per epoch, total convergence time, and number of epochs until convergence were not provided, aside from accuracy in the experiments.*
>
> **Response:**
>
> Thank you for pointing this out. We fully agree that detailed analysis on memory and computation are essential to demonstrate our method’s advantages.
>
> **Memory efficiency:**
>
> - **On GPU simulation, our method eliminates the memory for backward computation graph and intermediate activations**. Total memory consists of only the model parameters if using SGD optimizer, or three times the model parameters if using Adam optimizer. The memory to store the additional loss values are minimal (several scalars). Our current implementation needs to temporarily store the perturbation, however this memory can further be eliminated by applying a pseudo-random number generator to re-generate exactly the same perturbation vector, as described in Ref. [R1].
> - **On photonic chip, we do not need on-chip photonic memory**. We need memory in digital controller to save tensor-compressed weights and estimated gradients only. The memory cost in digital controller is the same as described in the last paragraph (GPU simulation).
>
> **Computation efficiency:**
>
> In our submission, we provided total convergence time in Table 4. Due to the page limitation we provided detailed break down of latency (training time) analysis in Appendix A.4.2. The results are based on simulation and it is infeasible for implementation on a real photonic chip at this time. We summarize the information in **Table R1** for your convenience.
>
> Table R1: Computation efficiency comparison between ONN and tensorized ONN implementations. **The results are based on simulation.** ONN-1 and TONN-1 denote space-multiplexing implementation. ONN-2 and TONN-2 denote time-multiplexing implementation.
>
> |  | Latency per Inference (ns) | Time per epoch (ms) | Number of epochs | Time to converge (s) | rel. $\ell$_2 error |
> | --- | --- | --- | --- | --- | --- |
> | ONN-1 (infeasible) | 51.30 | 0.17 | 10,000 | 1.74 | 0.667 |
> | ONN-2 | 1545.92 | 5.23 | 10,000 | 52.27 | 0.667 |
> | TONN-1 (ours) | 48.74 | 0.16 | 10,000 | 1.64 | **0.103** |
> | TONN-2 (ours) | 289.86 | 0.98 | 10,000 | 9.80 | **0.103** |
>
> [R1] Malladi, Sadhika, et al. "Fine-tuning language models with just forward passes." Advances in Neural Information Processing Systems 36 (2023): 53038-53075.

---

> ### Author Response · Authors · 2024-11-20
> **Response to Reviewer Syoh (Part II)**
>
> ***Weakness 2**: Additional experiments on other commonly used PDE datasets, such as Navier-Stokes, Darcy flow, and Burgers' equation, are needed beyond Black-Scholes and 20-dimensional HJB.*
>
> **Response**:
>
> Thank you for pointing out this aspect. We acknowledge the importance of evaluating our method on more benchmarks. We tested our method on 20-dim HJB equation primarily due to its edge applications in real-time safety-critical control of autonomous systems. However, **our method can also be applied to solve general PDEs.**
>
> **We have evaluated our methods on another PDE benchmark, one-dimensional Burgers’ equation,** as shown in **Appendix A.5** in our updated manuscript. ****The PDE definitions, baseline model, and training setups are consistent with the state-of-the-art PINN benchmark [R2]. The baseline model is a 5-layer MLP, each layer has a width of 100 neurons. The baseline model has 30701 parameters. The dimension of our tensor-compressed training is reduced to 1241 by folding the weight matrices in hidden layers as size $4\times 5\times 5\times 5\times 5\times 4$ and decomposing it with a TT-rank $(1, 2, 2, 1)$.
>
> The results are provided in Table R2, R3, R4.
>
> - **The BP-free loss computation does not hurt the training performance. Table R2** compares different loss computation methods. Our sparse-grid (**SG**) method is competitive compared to the original PINN loss evaluation using automatic differentiation (**AD**) while requiring much less forward evaluations than Monte Carlo-based Stein Estimator (**SE**).
> - **Our tensor-train (TT) dimension reduction greatly improves the convergence of ZO training**. **Table R3** compares the **FO** training (BP) and **ZO** training (BP-free) in the form of **standard** uncompressed and our tensor-compressed (**TT**) formats. Standard ZO training failed to converge well. Our ZO training method achieves much better convergence and final accuracy.
> - **In phase-domain training, our BP-free training achieves the lowest relative $\ell_2$ error.** As shown in **Table R4**, our method outperforms ZO method FLOPS [R3]. This is attributed to our tensor-train (TT) dimension reduction**.** Our method also outperforms FO method L2ight [R4]. The restricted learnable subspace of L2ight is not capable of training PINNs from scratch.
>
> **The above results support our claims in the submission.** Our method is the most scalable solution to enable real-size PINNs training on photonic computing hardware. Their remains a performance gap compared with weight domain FO training, the “ideal” upper bound. This performance gap cannot be completely eliminated, but may be further reduced with better variance reduction approaches in the future.
>
> Table R2: Relative $\ell_2$ error of FO training in weight domain using different loss computation methods.
>
> | Problem | AD | SE | SG (ours) |
> | --- | --- | --- | --- |
> | Samples | / | 2048 | 13 |
> | Burgers‘ | 1.37E-02 | 2.08E-02 | 1.39E-02 |
>
> Table R3: Relative $\ell_2$ error of different training method in weight domain.
>
> | Problem | Standard, FO | TT, FO | Standard, ZO | TT, ZO (ours) |
> | --- | --- | --- | --- | --- |
> | Burgers’ | 1.39E-02 | 4.82E-02 | 4.47E-01 | 9.50E-02 |
>
> Table R4: Relative $\ell_2$ error of phase domain training  on photonics.
>
> | Problem | FLOPS [R3] | L2ight [R4] | ours |
> | --- | --- | --- | --- |
> | Burgers‘ | 4.50E-01 | 5.72E-01 | 2.79E-01 |
>
> **We also have tried a Navier-Stokes PDE** (2D Navier-Stokes lid-driven flow experiment from SOTA PINN baseline [R2]). Details and results are provided in the following comment (Part III) due to page limit.
>
> [R2] Hao, Zhongkai, et al. "Pinnacle: A comprehensive benchmark of physics-informed neural networks for solving pdes." arXiv preprint arXiv:2306.08827 (2023).
>
> [R3] Gu, Jiaqi, et al. "FLOPS: Efficient on-chip learning for optical neural networks through stochastic zeroth-order optimization." *2020 57th ACM/IEEE Design Automation Conference (DAC)*. IEEE, 2020.
>
> [R4] Gu, Jiaqi, et al. "L2ight: Enabling on-chip learning for optical neural networks via efficient in-situ subspace optimization." *Advances in Neural Information Processing Systems* 34 (2021): 8649-8661.

---

> ### Author Response · Authors · 2024-11-20
> **Response to Reviewer Syoh (Part III)**
>
> ***Response to Weakness 2 (continue)***
>
> **We also have tried a Navier-Stokes PDE** (2D Navier-Stokes lid-driven flow experiment from SOTA PINN baseline [R2]). We provide the results in **Table R5, R6, R7**. However, this is a very challenging task: FO training needs over 15000 iterations to converge well (versus 4000 iterations in HJB PDE) due to its complicated optimization landscape. **All ZO training and photonic training fail to achieve good convergence after 40000 iterations. Among them, our method achieves the best accuracy** (4.82E-1 test relative $\ell_2$ error in weight domain training, and 6.99E-01 in phase domain photonic training). We feel that more studies are needed for Navier-Stokes PDE. Besides optimizing the ZO gradient estimation, we may also need to consider (1) an optimization framework better than popular SGD/GD, (2) better PINN architecture, (3) a deeper understanding of the optimization landscape. We expect that such an intensive study needs significant time beyond the ICLR rebuttal phase.
>
> Table R5: Navier-Stokes results. Relative $\ell_2$ error of FO training in weight domain using different loss computation methods.
>
> | Problem | AD | SE | SG (ours) |
> | --- | --- | --- | --- |
> | Navier-Stokes | 3.79E-02 | 5.34E-02 | 3.66E-02 |
>
> Table R6: Navier-Stokes results. Relative $\ell_2$ error of different training methods in weight domain.
>
> | Problem | Standard, FO | TT, FO | Standard, ZO | TT, ZO (ours) |
> | --- | --- | --- | --- | --- |
> | Navier-Stokes | 3.66E-02 | 6.86E-02 | 5.69E-01 | 4.82E-01 |
>
> Table R7: Navier-Stokes results. Relative $\ell_2$ error of phase domain training on photonics.
>
> | Problem | FLOPS [R3] | L2ight [R4] | TONN (ours) |
> | --- | --- | --- | --- |
> | Navier-Stokes | 9.84E-01 | 7.85E-01 | 6.99E-01 |
>
> ***Weakness 3**: The authors did not provide an explanation for the criteria used to determine "convergence," which is a key aspect of their experiments. In Figure 5, it appears possible that different convergence criteria could be applied depending on the epoch.*
>
> **Response:**
>
> We train all experiments for a total of 10,000 iterations or epochs. This number of training rounds was empirically found to be sufficient for the model to achieve a good global approximation (e.g., test relative l2 error below 1%). Our criteria to determine convergence is the same as the one adopted by SOTA PINN benchmark [R2]. Meanwhile, it is true that zeroth-order (ZO) training needs more iterations to converge than first-order (FO) training, since the convergence rate of ZO-SGD is dimension-dependent, while the convergence rate of SGD is dimension-independent, as shown in [R5]. However, our tensor-compressed ZO training method requires fewer iterations to achieve similar accuracy than other ZO methods, as shown in Figure 6 in our submission.
>
> ***Weakness 4**: The figures presenting the experimental results are too small, and the associated information or explanations are limited.*
>
> **Response:**
>
> Thank you for pointing out this aspect. **We have added larger figures for better readability in Appendix A.9** with extended explanations of each figure in our updated manuscript.
>
> [R2] Hao, Zhongkai, et al. "Pinnacle: A comprehensive benchmark of physics-informed neural networks for solving pdes." *arXiv preprint arXiv:2306.08827* (2023).
>
> [R3] Gu, Jiaqi, et al. "FLOPS: Efficient on-chip learning for optical neural networks through stochastic zeroth-order optimization." *2020 57th ACM/IEEE Design Automation Conference (DAC)*. IEEE, 2020.
>
> [R4] Gu, Jiaqi, et al. "L2ight: Enabling on-chip learning for optical neural networks via efficient in-situ subspace optimization." *Advances in Neural Information Processing Systems* 34 (2021): 8649-8661.
>
> [R5] Liu, Sijia, et al. "A primer on zeroth-order optimization in signal processing and machine learning: Principals, recent advances, and applications." *IEEE Signal Processing Magazine* 37.5 (2020): 43-54.

---

> ### Author Response · Authors · 2024-11-20
> **Response to Reviewer Syoh (Part IV)**
>
> ***Weakness 5**: Detailed settings and environment information used in the experiments were not provided (e.g., code, chip information)*
>
> **Response:**
>
> Thank you for mentioning this aspect. We acknowledge the importance of providing these information for reproductivity.
>
> Code: We have attached an anonymous repository of our source code at the beginning of the Appendix. We will open source our code implementation if the paper fortunately gets accepted.
>
> Chip information: We design the TONN inference accelerator on III–V-on-silicon MOSCAP platform [R6]. Such platform can heterogeneously integrate all the required devices in TONN, including quantum dot (QD) comb lasers, MOSCAP microring modulators, MOSCAP phase shifters (MZIs), and QD avalanche photodetectors (APDs) at wafer scale. The details can be found in Ref. [R6] and [R7].
>
>
> ## Responses to questions:
>
> ***Question 1**: As mentioned in Weakness 2, are there any experiments conducted on other datasets?*
>
> **Response**: Thank you for raising this question. We conducted experiments on Burgers’ equation and Navier-Stokes equation as suggested. More information is provided in our response to ***Weakness 2***.
>
> ***Question 2**:  As raised in Weakness 3, what criteria did the authors use to determine the point of convergence?*
>
> **Response**: Thank you for raising this question. We train the same number of iterations (epochs) for different methods. Our tensor-compressed ZO training method uses fewer iterations than other ZO methods to achieve a similar accuracy.  More information is provided in our response to ***Weakness 3***.
>
> ***Question 3**: Could you provide more information on the efficiency and memory aspects of the experiments? (e.g., memory usage, time per epoch, the time and number of epochs each method took to converge, etc.)*
>
> **Response**: Thank you for raising this question. We have provided detailed training efficiency in the Appendix A.4.2. We also summarized the memory and computational analysis in our response to ***Weakness 1***.
>
> [R6] Liang, D., Srinivasan, S., Kurczveil, G., Tossoun, B., Cheung, S., Yuan, Y., ... & Beausoleil, R. G. (2022). An energy-efficient and bandwidth-scalable DWDM heterogeneous silicon photonics integration platform. IEEE Journal of Selected Topics in Quantum Electronics, 28(6: High Density Integr. Multipurpose Photon. Circ.), 1-19.
>
> [R7] Xiao, X., On, M. B., Van Vaerenbergh, T., Liang, D., Beausoleil, R. G., & Yoo, S. J. (2021). Large-scale and energy-efficient tensorized optical neural networks on III–V-on-silicon MOSCAP platform. Apl Photonics, 6(12).

---

> > ### Comment · Reviewer_Syoh · 2024-11-23
> > **Additional question**
> >
> > I have reviewed the authors' responses. Thank you for the thoughtful and detailed replies. I have an additional question.
> >
> > In the authors' response Part II, it was mentioned that FO performance serves as the ideal upper bound. However, in Part III, it was observed that TT, ZO(proposed method) performance outperformed other settings. Further explanation on this discrepancy would be appreciated.

---

> ### Author Response · Authors · 2024-11-23
> **response to ``additional question" by the reviewer**
>
> Dear Reviewer Syoh,
>
> Thanks for your additional question. Our proposed method outperforms **some of (but not all) other methods** in Part III.
>
> ---In Part III, our TT-ZO method outperforms standard **ZO** (as shown in the 4th column of Table R6) and other **photonic training methods** (including the ZO method FLOPS, and sub-space FO method L2ight, as shown in Table R7). Here L2ight is a special FO method: since it uses sub-space training and only updates the diagonal elements, it has a higher error than our TT-ZO method implemented on photonics (TONN).
>
> --As shown in Table R6 (specifically Columns 2,3,5), **our method still has a higher error than the FO methods in both standard and TT settings**. Specifically, our TT-ZO has an error in the order of E-01, whereas the two FO methods have errors in the order of E-02.
>
> As a result, **the result in Part III is consistent with our statement in Part II**.
>
> I hope that above explanation has clarified the potential misunderstanding. Thanks again for your follow-up question!

---

> > ### Comment · Reviewer_Syoh · 2024-11-24
> > **To author**
> >
> > Thank you for all the kind responses from the authors.
> > I now understand the parts I had misunderstood, and I appreciate the clarifications provided.
> > All my questions have been resolved.
> > I will raise my score accordingly.

---

> > > ### Author Response · Authors · 2024-11-25
> > > **Thanks for your aknowledgemtn**
> > >
> > > Dear Reviewer Syoh,
> > >
> > > We are happy to know that all your questions have been addressed. Thanks a lot for raising your evaluation score!
> > >
> > > Your review comments have greatly enhanced the quality of our work. We sincerely appreciate your participation in the discussion and your recognition of the improved quality of our paper during the rebuttal phase.
> > >
> > > Sincerely,
> > >
> > > The authors

---

### Official Review · Reviewer_Gxiv · 2024-11-02

**Soundness:** 4
**Presentation:** 3
**Contribution:** 3
**Rating:** 6
**Confidence:** 4

**Summary:**

In this work, the authors propose a back-propagation-free framework along with the appropriate design for physics-informed training, targeting the real-time training of real-size Physics-Informed Neural Networks (PINNs) deployed on photonic chips. The proposed method allows one to reduce the required MZIs without sacrificing performance, thus reducing the footprints of the applied components, potentially enabling the scalability of optical neural networks in higher-dimensional architectures and tasks. More specifically, the proposed training framework includes a sparse grid Stein derivative estimation that allows to obviate backpropagation as well as a dimension-reduced zeroth-order optimization via tensor-train decomposition. This, according to the authors, improves the convergence of optimization and scalability, enabling end-to-end training on the chip. In turn, the paper introduces two photonic accelerator designs, one implementing the whole model on a single chip and the other using a single photonic tensor core with time multiplexing. The authors validate the convergence of the proposed training framework in two tasks, presenting numerical experiments that demonstrate the competitive performance that the proposed method achieves in contrast to the backpropagation baseline. Finally, the paper presents some evidence on hardware implementation, including footprint size and training latency time, as acquired from simulation experiments.

**Strengths:**

The paper deals with a highly relevant and challenging task, such as the training and scalability of optical neural networks, proposing a numerical training framework that is accompanied by a photonic accelerator design. The proposed tensor compressed zeroth-order optimization approach shows that not only it allows us to reduce the required MZIs but it is also experimentally demonstrated that benefits the convergence of the model.

With the integration of Strain derivative estimation and the proposed compressed tensor zeroth order optimization that reduces the zeroth order gradient variance, the paper reports significant improvements in contrast to existing on-chip backpropagation-free optical neural networks training lowering simultaneously the required MZIs.

**Weaknesses:**

Although, as the paper presents an optical inference engine it lacks critical details regarding the overall architecture and computational performance. For example, the authors mention in the supplementary material that the phase $\phi$ is uniformly quantized to 8 bit in the simulation but they do not comment on the rate at which such bit resolution can be achieved. Even state-of-the-art optical devices facing low SNRs, especially at high computational rates, significantly affecting the effective bit resolution.

The interconnection between the Digital Control System and TONN is not discussed in a sufficiently detailed way. Do additional devices (such as photodiodes) are needed and do they introduce any further noise to the overall setup?

Synchronization of the Digital Control System and TONN introduces significant additional overhead, increasing the latency of the proposed architecture ($\approx500$ ns /epoch). This is further exaggerated because of the tensor compression which factorizes the weights, significantly increasing the calls on the Digital Control System. This potentially hinders the high inference speed in higher-dimensional tasks. Introducing a complexity analysis with reference to the iterations per epoch and calls to the TONN inference engine will provide further insights regarding the scalability of the proposed method. In my opinion, since the paper targets the scalability of back-propagation-free training, the authors should further discuss the unavoidable overhead that is introduced by tensor compression and further analyze the reported latency $t_{DIG}$ in A.4.2.

Although the sparse-grid stain derivative estimator significantly reduces the number of nodes needed in contrast to the samples needed in Monte Carlo estimation, the paper does not provide any convergence and cost complexity analysis for the proposed method. Taking into account the significant fabrication losses that intrinsically exist in optical devices, a convergence analysis on the ideal optimization case would strengthen the scalability claim of the authors. Without theoretical guarantees or experimental evaluation, the claim that “Our results can be easily extended to solve image and speech problems on photonic and other types of edge platform” sounds stronger than what the paper can support, considering the fabrication imperfections and the fact that it presents only simulation results in 20-dimensional task, which is much smaller than MNIST benchmark, for example.

**Questions:**

In what computational rate do the applied components of the proposed TONN support such a high effective bit resolution?

---

> ### Author Response · Authors · 2024-11-20
> **Response to Reviewer Gxiv (Part I)**
>
> ## Responses to weaknesses:
>
> ***Weakness 1**: Although, as the paper presents an optical inference engine it lacks critical details regarding the overall architecture and computational performance. For example, the authors mention in the supplementary material that the phase  is uniformly quantized to 8 bit in the simulation but they do not comment on the rate at which such bit resolution can be achieved. Even state-of-the-art optical devices facing low SNRs, especially at high computational rates, significantly affecting the effective bit resolution.*
>
> **Response:**
>
> We thank the reviewer’s comments on the bit accuracy of the phase and the SNRs. In this paper, we assume a weight stationary (WS) scheme, where the weight matrices are programmed into the phase shifters in the MZI mesh, and the input vectors are encoded in the high-speed (10 GHz) optical signals. In each training iteration, the same weights (phases) are multiplied with batched (e.g., 1000) input data. As a result, the update rate of the phase shifters is ~ 10 GHz/1000 = 10 MHz. In a system-level study of MZI-mesh-based photonic AI accelerators, a 12-bit DAC is enough to support the 8-bit accuracy of the weights [R1]. Considering that a 12-bit DAC with a 10 MHz sampling rate is very mature [R2], assuming 8-bit weights (phases) in our setting is reasonable.
>
> The minimum optical SNR at the output of the MZI mesh is SNR = $2^{b_{out}}$, where $b_{out}$ is the required bit accuracy of the output of the matrix multiplication. The optical SNR can be improved by increasing the input laser power, reducing the optical insertion loss, increasing the optical gain, and increasing the sensitivity of the photodiodes. For instance, the platform in Ref. [R3] can provide lasers with high wall-plug efficiency, optical modulators, and MZIs with low insertion loss, on-chip optical gain, and quantum dot avalanche photodiodes with low sensitivity. Furthermore, the tensor decomposition in our work reduces the number of cascaded stages of MZIs, significantly reducing the insertion loss induced by cascaded MZIs.
>
> ***Weakness 2**: The interconnection between the Digital Control System and TONN is not discussed in a sufficiently detailed way. Do additional devices (such as photodiodes) are needed and do they introduce any further noise to the overall setup?*
>
> **Response:**
>
> Thank you for raising this concern. The digital control system can be implemented via electronic-photonic co-integration that contains an FPGA or ASIC for controlling and digital calculations required by BP-free training, digital electronic memory (e.g., DRAM) for weight and data storage and buffering, and ADC/DACs for converting the digital data to the tuning voltages of the modulators and phase shifters. As a result, **no additional optical devices are required other than the TONN inference accelerator we introduced in the paper**. The noise induced by the digital control system is decided by the bit accuracy of the ADCs and DACs.
>
> [R1] Demirkiran, C., Eris, F., Wang, G., Elmhurst, J., Moore, N., Harris, N. C., ... & Bunandar, D. (2023). An electro-photonic system for accelerating deep neural networks. ACM Journal on Emerging Technologies in Computing Systems, 19(4), 1-31.
>
> [R2] https://www.ti.com/lit/ds/symlink/ads802.pdf?ts=1731643492549
>
> [R3] Liang, D., Srinivasan, S., Kurczveil, G., Tossoun, B., Cheung, S., Yuan, Y., ... & Beausoleil, R. G. (2022). An energy-efficient and bandwidth-scalable DWDM heterogeneous silicon photonics integration platform. IEEE Journal of Selected Topics in Quantum Electronics, 28(6: High Density Integr. Multipurpose Photon. Circ.), 1-19.

---

> ### Author Response · Authors · 2024-11-20
> **Response to Reviewer Gxiv (Part II)**
>
> ***Weakness 3***: *Synchronization of the Digital Control System and TONN introduces significant additional overhead, increasing the latency of the proposed architecture (≈500 ns /epoch). This is further exaggerated because of the tensor compression which factorizes the weights, significantly increasing the calls on the Digital Control System. This potentially hinders the high inference speed in higher-dimensional tasks. Introducing a complexity analysis with reference to the iterations per epoch and calls to the TONN inference engine will provide further insights regarding the scalability of the proposed method. In my opinion, since the paper targets the scalability of back-propagation-free training, the authors should further discuss the unavoidable overhead that is introduced by tensor compression and further analyze the reported latency in A.4.2.*
>
> **Response:**
>
> We thank the reviewer for pointing out the synchronization overhead. We noticed that you mentioned two aspects: 1) overhead for tensor-train compression and 2) synchronization between the digital system and TONN.
>
> 1. Regarding overhead for tensor-train compression. There is **no latency overhead for low-rank tensor compression/factorization** in our implementation. Instead of factorizing a full-size weight matrix, we initialize the weight matrix in a TT format with a random initial guess. In the training process, we directly update the TT cores without any factorization to minimize the loss function. The reported latency analysis in Appendix A.4.2 has considered all aspects based on our implementation.
>
> 2. Regarding synchronization between the digital system and TONN. In the WS scheme, weight buffers are used, which means that the weights for the next set of matrix multiplication are loaded into the weight buffer, while the MZI mesh performs matrix multiplication with the current weight values. **The latency is limited by the tuning mechanism of the phase shifters.** In our case, the tuning mechanism is the III-V-on-silicon metal-oxide-semiconductor capacitor (MOSCAP) [R3], which has a modulation speed of tens of GHz. This is why we added 0.1 ns latency during the calculation, as shown in Appendix A.4.2.
>
> [R3] Liang, D., Srinivasan, S., Kurczveil, G., Tossoun, B., Cheung, S., Yuan, Y., ... & Beausoleil, R. G. (2022). An energy-efficient and bandwidth-scalable DWDM heterogeneous silicon photonics integration platform. IEEE Journal of Selected Topics in Quantum Electronics, 28(6: High Density Integr. Multipurpose Photon. Circ.), 1-19.

---

> ### Author Response · Authors · 2024-11-20
> **Response to Reviewer Gxiv (Part III)**
>
> ***Weakness 4**: Although the sparse-grid stain derivative estimator significantly reduces the number of nodes needed in contrast to the samples needed in Monte Carlo estimation, the paper does not provide any convergence and cost complexity analysis for the proposed method. Taking into account the significant fabrication losses that intrinsically exist in optical devices, a convergence analysis on the ideal optimization case would strengthen the scalability claim of the authors.*
>
> **Response:**
>
> Thank you for bringing up this important point regarding the convergence and cost complexity analysis of our proposed method. We agree that such an analysis is crucial for demonstrating the scalability and practical applicability of our approach.
>
> 1. **Ablation studies on convergence behavior and computational complexity in PINNs training:**
>
> We have conducted an ablation study comparing the convergence behavior and computational complexity of the Monte Carlo-based Stein estimator and our proposed sparse-grid Stein derivative estimator in solving the Black-Scholes equation. The results are shown in Table R1 and R2 below.
>
> - Reducing the number of samples of Monte Carlo-based Stein estimator from 2048 to 64 harms the training convergence.
> - Our proposed sparse-grid Stein derivative estimator uses much fewer samples than Monte Carlo-based Stein estimator, and achieves smaller test error on convergence.
>
> Table R1: Ablation study on number of samples of Monte Carlo-based Stein Estimator on solving Black-Scholes equation.
>
> | Number of Samples | 64 | 512 | 2048 |
> | --- | --- | --- | --- |
> | rel. $\ell_2$ error | 1.96E-01 | 5.89E-02 | 5.41E-02 |
>
> Table R2: Ablation study on number of samples of our proposed sparse-grid Stein Estimator on solving Black-Scholes equation.
>
> | Level | 2 | 3 | 4 |
> | --- | --- | --- | --- |
> | Number of Samples | 5 | 13 | 29 |
> | rel. $\ell_2$ error | 6.41E-02 | 5.28E-02 | 5.20E-02 |
>
> 2. **Convergence analysis for synthetic functions**
>
> Additionally, to provide a clear illustration of the convergence properties and computational costs, we performed a convergence analysis for evaluating the Laplacian of a synthetic function. Specifically, we considered the function: $u(x, y) = \mathbb{E}_{\delta \sim \mathcal{N}(\mathbf{0}, \sigma^2 \mathbf{I})} [f(x + \delta_x, y + \delta_y)],$ where $f(x, y) = e^{-\frac{\sigma^2}{2}} e^{-x} \sin(y).$ It can be derived that $u(x, y) = e^{-x} \sin(y)$ and thus $\Delta u = 0.$
>
> We set the noise level $\sigma = 0.1$ and approximated the Laplacian of $u$ on a uniform $100 \times 100$ grid over the domain $[0,1]^2.$ We compared the $\ell_2$ errors and computational costs of the sparse-grid Stein estimator at different accuracy levels and the Monte Carlo Stein estimator with varying numbers of samples. The results are summarized in the following tables:
>
> Table R3: Monte Carlo Stein Estimator for evaluating the Laplacian
>
> | Number of Samples | $\ell_2$ Error | Number of function queries |
> | --- | --- | --- |
> | 1,024 | 10.7437 | 4096 |
> | 2,048 | 7.6796 | 8192 |
> | 4,096 | 5.4245 | 16384 |
> | 8,192 | 3.8732 | 32768 |
> | 16,384 | 2.7016 | 65536 |
>
> Table R4: Sparse-Grid Stein Estimator for evaluating the Laplacian
>
> | Accuracy Level | Number of samples | $\ell_2$ Error | Number of function queries |
> | --- | --- | --- | --- |
> | 3 | 13 | 0.1142 | 52 |
> | 4 | 29 | 2.8217e-07 | 116 |
> | 5 | 53 | 4.0797e-08 | 212 |
> | 6 | 89 | 8.4356e-13 | 356 |
> | 7 | 137 | 8.7094e-13 | 548 |
>
> **These results demonstrate that the sparse-grid Stein estimator achieves significantly lower errors with substantially fewer computational resources compared to the Monte Carlo estimator.** For example, at accuracy level 4, the sparse-grid method achieves an $\ell_2$ error of  $2.82 \times 10^{-7}$  using only 29 samples, whereas the Monte Carlo method with 16,384 samples still has an error of  2.7016.

---

> ### Author Response · Authors · 2024-11-20
> **Response to Reviewer Gxiv (Part IV)**
>
> ***Weakness 5**: Without theoretical guarantees or experimental evaluation, the claim that “Our results can be easily extended to solve image and speech problems on photonic and other types of edge platform” sounds stronger than what the paper can support, considering the fabrication imperfections and the fact that it presents only simulation results in 20-dimensional task, which is much smaller than MNIST benchmark, for example.*
>
> **Response:**
>
> Thank you for sharing this concern. We acknowledge the importance of experimental evaluation to verify our claim. **We have applied our tensor-compressed training for image classification task on MNIST dataset,** as shown in **Appendix A.6** in our updated manuscript. Note that our proposed sparse-grid loss evaluation is designed for PINNs training only, so sparse-grid is not used for image classification.
>
> The baseline model is a two-layer MLP (784$\times$1024, 1024$\times$10) with 814,090 parameters. The dimension of our tensor-compressed training is reduced to 3,962 by folding the input and output layer as size $7\times 4\times 4\times 7\times 8\times 4\times 4\times 8$ and $8\times 4\times 4\times 8\times 1\times 5\times 2\times 1$, respectively. Both the input layer and the output layer are decomposed with a TT-rank $(1, 6, 6, 6, 1)$. Models are trained for 15000 iterations with a batch size 2000, Adam optimized with initial learning rate 1e-3 and decayed by 0.8 every 3000 iterations. For ZO training, we set query number $N=10$ and smoothing factor $\mu=0.01$.
>
> The results are shown in Table R5 and R6.
>
> - Comparing results of weight domain training (**Table R5**):
>     - **Our tensor-train (TT) compressed training does not harm the model expressivity**, as TT training achieved a similar test accuracy as standard training in first-order (FO) training.
>     - **Our TT compressed training greatly improves the convergence of ZO training,** and reduces the performance gap between ZO and FO.
> - Comparing results of phase domain training (**Table R6**):
>     - **Our method outperforms baseline ZO training method FLOPS**. This is also attributed to the reduced gradient estimation variance. Note that the performance gap between phase domain training and weight domain training could be attributed to the low-precision quantization, hardware imperfections, etc., as illustrated in Section 5.2.
>
> The results on the MNIST dataset are consistent with our claims in the submission, and supports our claim that **our method can be extended to image problems with higher dimensions.**
>
> Table R5: Validation accuracy of weight domain training.
>
> | Method | Standard, FO | TT, FO | Standard, ZO | TT, ZO (ours) |
> | --- | --- | --- | --- | --- |
> | Val. Accuracy (%) | 97.83±1.02 | 97.26±0.15 | 83.83±0.44 | 93.21±0.46 |
>
> Table R6: Validation accuracy of phase domain training.
>
> | Method | FLOPS [R4] | TONN (ours) |
> | --- | --- | --- |
> | Val. Accuracy (%) | 41.72±5.50 | 87.91±0.59 |
>
> ## Responses to questions:
>
> ***Question 1**: In what computational rate do the applied components of the proposed TONN support such a high effective bit resolution?*
>
> **Response:**
>
> Thank you for raising this question. We assume a **10 GHz computational rate** in the paper, which means 10 GHz modulation and detection of the analog data, as well as 10 MHz weight modulation in a weight stationary (WS) setting. This requires a 10 Gsps sample rate for the ADC and DAC. For example, the ADC12DJ5200SE ADC from Texas Instruments can support singled-ended input RF-sampling 12-bit ADC with single-channel 10.4 GSPS [R5]. High-speed and high-bit-accuracy ADCs are relatively power-consuming. On the algorithm side, quantization can be used to relieve the bit accuracy requirements. On the hardware side, the more advanced node size of the semiconductor chips can further reduce the power consumption of the ADCs/DACs.
>
> [R4] Gu, Jiaqi, et al. "FLOPS: Efficient on-chip learning for optical neural networks through stochastic zeroth-order optimization." *2020 57th ACM/IEEE Design Automation Conference (DAC)*. IEEE, 2020.
>
> [R5] https://www.ti.com/product/ADC12DJ5200SE

---

> > ### Comment · Reviewer_Gxiv · 2024-11-23
> >
> > I would like to thank the authors for answering my comments. I don't have any follow-up questions
> >
> > I will update the soundness from "good" to "excellent" and presentation from "fair" to "good".  I will maintain my score on contribution and overall rate.

---

> > > ### Author Response · Authors · 2024-11-24
> > > **Thank you for your very positive feedback**
> > >
> > > Dear Reviewer Gxiv,
> > >
> > > We are happy to know that all your questions have been addressed. Big thanks to your participation in the discussion and your recognition of the enhanced soundness and presentation of our paper following the discussion process. Your insightful feedback helped a lot to enhance the quality of our work.

---

### Official Review · Reviewer_kjo5 · 2024-11-04

**Soundness:** 2
**Presentation:** 3
**Contribution:** 2
**Rating:** 3
**Confidence:** 5

**Summary:**

In this paper, the authors aim to address the on-chip training problem for a photonic accelerator specific to PINN by using a BP-free method. The authors propose two-level methods, including a sparse grid Stein derivative estimator that uses samples to estimate gradient in a zeroth-order way, followed by a way to reduce the dimensionality of the weight matrix to improve ZO performance. The paper tries to approach the accuracy of the traditional BP-based method on a toy neural network but still has 2x worse error with their method.

**Strengths:**

Strength:
* A complete evaluation from algorithm to chip, and consider some realistic constraints, e.g., resolution limits.

**Weaknesses:**

Weakness
* First, the papers' main methods are both not new but borrow from other domains without many new contributions/modifications. For example, the low-rank compression with tensor-train is not new. Also the sparse-grid method is also a typical method in the PDE domain.
* Second, some claims are not accurate. For example, a photonic accelerator will not build a huge chip directly for a 128x128 array. Instead, it also blokifies the large weight matrix into a small one and implements on-chip. Therefore, the claim in Sec.4 is not right. Especially regarding implementing the NxN matrix, it is infeasible for partial PINNs.
* Third, the implementation of baselines might be wrong, as your understanding of vanilla ONN may not be right. Could you provide more details about how to train an ONN with ZO-based methods like FLOPS and L2ight? What is the photonic tensor core size?
* Fourth, the used model, 3-layer MLP, is a toy example. The model itself is not optimized for PDE tasks; it may be overfitting or have many pruning opportunities. This can be observed from Tabl2. Applying compression leads to much better accuracy on harder task, 20dim-HJB, meaning the model parameters are highly redundant.
* Fifth, the ZO performance is 5x worse than using the FO method (Table. 2) on the harder task, weakening the claims of the paper.

**Questions:**

Questions
* Could you try a harder task and use a more optimized, less-overparametrized PINN model for PDE?

---

> ### Author Response · Authors · 2024-11-20
> **Response to Reviewer kjo5 (Part I)**
>
> ## Responses to weaknesses:
>
> ***Weakness 1**: First, the papers' main methods are both not new but borrow from other domains without many new contributions/modifications. For example, the low-rank compression with tensor-train is not new. Also the sparse-grid method is also a typical method in the PDE domain.*
>
> **Response**: Thanks you for your concerns on our work’s contributions.
>
> - We completely agree that sparse-grid method and tensor-train decomposition methods have already been reported in the math community. Both of them are widely used in high-dimensional function approximation [R1] and uncertainty quantification [R2] of PDE-described problems. However, neither of them have been used in BP-free training.
> - Our algorithmic contributions are two fold: (1). we apply sparse-grid to achieve BP-free loss evaluation in PINN, (2). we apply tensor-train to perform variance-reduction in zeroth-order (ZO) training. These two combined allows us to train a real-size PINN on photonic hardware for the first time, advancing the state-of-the-art in photonic neural network training.
> - With full respect to the reviewer, we would like to remark that many novel ideas in engineering communities (including machine learning) come from the application of existing theories developed in the math community. This kind of research is regarded as novel and often produce high-impact research results. For instance, Neural ODE [R3] uses the well-known ODE model to replace ResNet and makes the model infinitely deep; the recent MAMBA [R4] applies state-space model in dynamic system to solve long-sequence modeling problems and achieved state-of-the-art performance in NLP.
>
> [R1] Griebel, Michael. *Sparse grids and related approximation schemes for higher dimensional problems*. SFB 611, 2005.
>
> [R2] Bigoni, Daniele, Allan P. Engsig-Karup, and Youssef M. Marzouk. "Spectral tensor-train decomposition." SIAM Journal on Scientific Computing 38.4 (2016): A2405-A2439.
>
> [R3] Chen, Ricky TQ, et al. "Neural ordinary differential equations." *Advances in neural information processing systems* 31 (2018).
>
> [R4] Gu, Albert, and Tri Dao. "Mamba: Linear-time sequence modeling with selective state spaces." *arXiv preprint arXiv:2312.00752* (2023).

---

> ### Author Response · Authors · 2024-11-20
> **Response to Reviewer kjo5 (Part II)**
>
> ***Weakness 2**: Second, some claims are not accurate. For example, a photonic accelerator will not build a huge chip directly for a 128x128 array. Instead, it also blokifies the large weight matrix into a small one and implements on-chip. Therefore, the claim in Sec.4 is not right. Especially regarding implementing the NxN matrix, it is infeasible for partial PINNs.*
>
> **Response**: Thank you for raising this concern. We have double checked our implementation, and we confirm that our implementation is correct. Here we provide more details of comparing our proposed zeroth-order tensorized ONN (TONN) training with prior zeroth-order ONN training in both space-multiplexing and time-multiplexing implementations. We consider implementing a $128\times 128$ weight matrix as an example. TONN folds the weight matrix as size $4\times 4\times 8\times 8\times 4\times 4$ and decomposes with a TT-rank $(1,2,2,1)$.
>
> - We first discuss the **space-multiplexing implementation**, which implements the whole network on one chip.
>     - Hardware cost: To implement a $N\times N$ matrix on ONN, $O(N^2)$ MZIs are required no matter the large ONN is implemented with a single large MZI mesh or multiple smaller MZI meshes. For a single MZI mesh implementation, the number of MZIs is $(\frac{N(N-1)}{2} + N + \frac{N(N-1)}{2})=N^2$. To implement with multiple smaller MZI meshes, saying implementing a $N\times N$ matrix by $\frac{N}{k}\times \frac{N}{k}$ blocks with size of with $k\times k$, the number of MZIs is $\frac{N}{k}\times \frac{N}{k}\times (\frac{k(k-1)}{2} + k + \frac{k(k-1)}{2})=N^2$.
>     - Due to the large MZI cost and footprint, space-multiplexing implementation of ONN is not realistic for large weight matrices. In comparison, space-multiplexing implementation of our method is feasible. Detail comparison are provided in Table R1.
> - We then discuss **time-multiplexing implementation**, which only implements one $k\times k$ photonic tensor core (PTC) and computes the large weight matrix by repeatedly re-programme the same PTC.
>     - Hardware cost: We implement both ONN and our method (TONN) with a $8\times 8$ photonic tensor core with 8 wavelengths for each input.
>     - Computation efficiency: ONN requires $(128/8 * 128/8) / 8 = 32$ cycles, while TONN only requires $128/8/8 \times 3 = 6$ cycles.
>     - Impact on zeroth-order training efficiency: ONN with time-multiplexing reduces the MZI cost for implementation, but does not reduce the number of training parameters and ZO gradient variance. As a result, training ONN with zeroth-order method failed to converge. Our TONN reduces both the MZI cost and the number of training parameters, which serves as a highly effective variance-reduction approach to greatly improve the convergence of zeroth-order training.
>
> Table R1: Comparison between ONN and TONN in space-multiplexing implementation when solving Black-Scholes equation.
>
> |  | \# MZIs | Footprint (mm$^2$) | Cycles per inference | Time per inference (ns) | Total training time (s) | rel. $\ell$_2 error |
> | --- | --- | --- | --- | --- | --- | --- |
> | ONN-1 | 16,384 | 3,975.68 (infeasible) | 1 | 51.3 | 1.74 | 0.667 |
> | TONN-1 | 384 | 102.72 (feasible) | 1 | 48.74 | 1.64 | 0.103 |
>
> Table R2: Comparison between ONN and TONN in time-multiplexing implementation when solving Black-Scholes equation.
>
> |  | \# MZIs | Footprint (mm$^2$) | Cycles per inference | Time per inference (ns) | Total training time (s) | rel. $\ell$_2 error |
> | --- | --- | --- | --- | --- | --- | --- |
> | ONN-2 | 64 | 18.72 | 32 | 1,545.92 | 52.27 | 0.667 |
> | TONN-2 | 64 | 18.72 | 6 | 289.86 | 9.8 | 0.103 |

---

> ### Author Response · Authors · 2024-11-20
> **Response to Reviewer kjo5 (Part III)**
>
> ***Weakness 3**: Third, the implementation of baselines might be wrong, as your understanding of vanilla ONN may not be right. Could you provide more details about how to train an ONN with ZO-based methods like FLOPS and L2ight? What is the photonic tensor core size?*
>
> **Response**: Thanks a lot for the comments!
>
> We implement the baseline methods, FLOPS [R5] and L2ight [R6], by adopting the same ONN settings as that in Ref. [R6]. The photonic tensor core size is 8 × 8. The M × N weight matrix $\boldsymbol{W}$ is partitioned into P × Q blocks $\\{\boldsymbol{W}\_{pq}\\}\_{p=0,q=0}^{p=P-1,q=Q-1}$ of size 8 × 8 and mapped to photonic tensor cores. Each block $\boldsymbol{W}\_{pq}$ is parameterized as $\boldsymbol{W}\_{p q}\left(\boldsymbol{\Phi}\_{p q}\right)=\boldsymbol{U}\_{p q}\left(\boldsymbol{\Phi}\_{p q}^U\right) \boldsymbol{\Sigma}\_{p q}\left(\boldsymbol{\Phi}\_{p q}^S\right) \boldsymbol{V}\_{p q}^*\left(\boldsymbol{\Phi}\_{p q}^V\right)$. We believe that this is a correct understanding and implementation of vanilla ONN.
>
> FLOPs [R5] is a ZO based method. We use zeroth-order gradient estimation to estimate the gradients of all MZI phases (i.e., $\boldsymbol{\Phi}\_{pq}^U, \boldsymbol{\Phi}\_{pq}^S, \boldsymbol{\Phi}\_{pq}^V)$
>
> L2ight [R6] is a subspace FO based method. Due to the intractable gradients for $\boldsymbol{\Phi}\_{pq}^U$ and $\boldsymbol{\Phi}\_{pq}^V$, only the MZI phase shifters in the diagonal matrix $\boldsymbol{\Phi}\_{pq}^S$ are trainable. This restricts the training space (i.e., subspace training), leading to lower model accuracy.  We noticed that the trainable MZIs in L2ight were counted incorrectly in our original submission. We have updated the data in the revised manuscript, but the training results (e.g., training convergence curves, accuracy) and conclusion remain the same.
>
> We have added detailed ONN implementation settings for baseline methods in Appendix A.3.
>
> ***Weakness 4***: Fourth, the used model, 3-layer MLP, is a toy example. The model itself is not optimized for PDE tasks; it may be overfitting or have many pruning opportunities. This can be observed from Tabl2. Applying compression leads to much better accuracy on harder task, 20dim-HJB, meaning the model parameters are highly redundant.
>
> **Response**: Thank you for pointing out this aspect.
>
> - Overfitting: **We have added an ablation study on the MLP hidden layer size,** as shown in **Appendix A.7.2** in our updated manuscript. We trained 3-layer MLPs with different hidden layer sizes to solve the 20-dim HJB equation. We use automatic differentiation for loss evaluation and first-order (FO) gradient descent to update model parameters. Other training setups are the same as illustrated in Appendix A.3. The results are shown in Table R3 below. Smaller-size MLP models lead to larger testing errors. This indicates that **the MLP model used in our submission does not have an overfitting problem**. A large hidden layer is favored to ensure enough model expressivity.
> - Pruning opportunities: We agree that it is possible to reduce the number of trainable parameters by pruning the original MLP model, so as to improve ZO training convergence. Meanwhile, **pruning cannot reduce the hardware cost (e.g., MZIs) on photonic platforms**, while our tensor-train compressed method reduces the trainable parameters, improve ZO training convergence, and reduce the photonic hardware cost at the same time.
>
> Table R3: Ablation study on hidden layer size of baseline 3-layer MLP model when learning 20-dim HJB equation.
>
> | Hidden layer size | 512 | 256 | 128 | 64 | 32 |
> | --- | --- | --- | --- | --- | --- |
> | Params | 274,433 | 71,681 | 19,457 | 5,633 | 1,793 |
> | rel. $\ell_2$ error | (2.72±0.23)E-03 | (4.31±0.19)E-03 | (7.51±0.36)E-03 | (8.15±0.67)E-03 | (9.25±0.27)E-03 |
>
> [R5] Gu, Jiaqi, et al. "FLOPS: Efficient on-chip learning for optical neural networks through stochastic zeroth-order optimization." *2020 57th ACM/IEEE Design Automation Conference (DAC)*. IEEE, 2020.
>
> [R6] Gu, Jiaqi, et al. "L2ight: Enabling on-chip learning for optical neural networks via efficient in-situ subspace optimization." *Advances in Neural Information Processing Systems* 34 (2021): 8649-8661.

---

> ### Author Response · Authors · 2024-11-20
> **Response to Reviewer kjo5 (Part IV)**
>
> ***Weakness 5**: Fifth, the ZO performance is 5x worse than using the FO method (Table. 2) on the harder task, weakening the claims of the paper.*
>
> **Response**: Thank you for pointing out this aspect. In principle, ZO method can hardly achieve better accuracy than FO trainining unless very low-precision quantization is used, since the former only use  approximate gradients and induce extra gradient variance and bias. This has been well observed in almost all ZO training papers. The advantage of ZO method versus FO method includes (1) lower memory cost and (2) easier implementation on edge hardware (especially analog platforms such as photonic circuits). **Our goal is not to beat FO training in terms of accuracy by ZO methods. Instead, we aim to (1) greatly improve the scalability and accuracy of end-to-end ZO training method, (2) develop the first end-to-end photonic training accelerator for real-size PINN**.
>
> In addition, we would like to highlight the following:
>
> - In phase-domain training, i.e., on-chip training on ONN, **our ZO training outperforms state-of-the-art (SOTA) FO [R6] training methods**, as shown in Figure 7 and Table 3. This is because the gradients of most MZI phases on ONN are intractable. Such restricted subspace learning cannot provide adequate degree of freedom for training PINNs from scratch.
> - As described above, it’s unfair to compare ZO methods with FO methods. SOTA ZO training method [R7] also has a performance gap compared with FO training. We conducted FO training in our submission because we wanted to use the FO result as a reference or a performance “upper bound” for evaluating ZO training results on ONN. **In practice, real-size full-model BP-based training is infeasible based on current photonic computing technology.**
> - It’s more fair to compare different ZO methods. In Table 2, **our tensor-train compressed ZO training method outperforms standard ZO training by over 5 times**. Our method also outperforms all existing ZO photonic training methods in terms of both accuracy and hardware cost.
>
> [R6] Gu, Jiaqi, et al. "L2ight: Enabling on-chip learning for optical neural networks via efficient in-situ subspace optimization." *Advances in Neural Information Processing Systems* 34 (2021): 8649-8661.
>
> [R7] Chen, Aochuan, et al. "Deepzero: Scaling up zeroth-order optimization for deep model training." *arXiv preprint arXiv:2310.02025* (2023).

---

> ### Author Response · Authors · 2024-11-20
> **Response to Reviewer kjo5 (Part V)**
>
> ***Question 1**: Could you try a harder task and use a more optimized, less-overparametrized PINN model for PDE?*
>
> **Response:**
>
> Thank you for raising this question. We have conducted several experiments to address your question.
>
> 1. **Harder tasks:**
>
> **We have evaluated our methods on another PDE benchmark, one-dimensional Burgers’ equation,** as shown in **Appendix A.5** in our updated manuscript. The PDE definitions, baseline model, and training setups are consistent with the state-of-the-art PINN benchmark [R8]. The baseline model is a 5-layer MLP, each layer has a width of 100 neurons. The baseline model has 30,701 parameters. The dimension of our tensor-compressed training is reduced to 1,241 by folding the weight matrices in hidden layers as size $4\times 5\times 5\times 5\times 5\times 4$ and decomposing it with a TT-rank $(1, 2, 2, 1)$.
>
> The results are provided in Table R4, R5, R6.
>
> - **The BP-free loss computation does not hurt the training performance. Table R4** compares different loss computation methods. Our sparse-grid (**SG**) method is competitive compared to the original PINN loss evaluation using automatic differentiation (**AD**) while requiring much less forward evaluations than Monte Carlo-based Stein Estimator (**SE**).
> - **Our tensor-train (TT) dimension reduction greatly improves the convergence of ZO training**. **Table R5** compares the **FO** training (BP) and **ZO** training (BP-free) in the form of **standard** uncompressed and our tensor-compressed (**TT**) formats. Standard ZO training failed to converge well. Our ZO training method achieves a much better final accuracy due to the gradient variance reduction caused by TT.
> - **In phase-domain training, our BP-free training achieves the lowest relative $\ell_2$ error.** As shown in **Table R6**, our method outperforms ZO method FLOPS [R5]. This is attributed to our tensor-train (TT) variance reduction. Our method also outperforms FO method L2ight [R6]. The restricted learnable subspace of L2ight is not capable of training PINNs from scratch.
>
> **The above results support our claims in the submission.** Our method has achieved the best scalability and enable accurate real-size PINNs training on photonic computing hardware. There remains a performance gap compared with weight-domain FO training, the “ideal” upper bound. This performance gap cannot be completely removed due to the additional gradient variance in ZO training, but it may be further reduced in the future with better and novel variance reduction approaches.
>
> Table R4: Relative $\ell_2$ error of FO training in weight domain using different loss computation methods.
>
> | Problem | AD | SE | SG (ours) |
> | --- | --- | --- | --- |
> | Samples | / | 2048 | 13 |
> | Burgers‘ | 1.37E-02 | 2.08E-02 | 1.39E-02 |
>
> Table R5: Relative $\ell_2$ error of different training method in weight domain.
>
> | Problem | Standard, FO | TT, FO | Standard, ZO | TT, ZO (ours) |
> | --- | --- | --- | --- | --- |
> | Burgers’ | 1.39E-02 | 4.82E-02 | 4.47E-01 | 9.50E-02 |
>
> Table R6: Relative $\ell_2$ error of phase domain training on photonics.
>
> | Problem | FLOPS [R5] | L2ight [R6] | ours |
> | --- | --- | --- | --- |
> | Burgers‘ | 4.50E-01 | 5.72E-01 | 2.79E-01 |
> 2. **A more optimized PINN model**:
>
> MLPs are most widely used network architecture for PINNs and are capable of solving over 20 PDE benchmarks, see Ref. [R8]. It is proper to use MLP architecture as the baseline PINN model. Meanwhile, our response to Weakness 4 shows that reducing the MLP size can lead to increased error. Therefore, we feel that there is not much space to further optimize the PINN model.
>
> 3. **Comparisons with a less-overparameterized PINN model**:
>
> We have conducted experiments to compare this aspect. The results are provided in the following comment (Part VI) due to the page limit.
>
> [R5] Gu, Jiaqi, et al. "FLOPS: Efficient on-chip learning for optical neural networks through stochastic zeroth-order optimization." *2020 57th ACM/IEEE Design Automation Conference (DAC)*. IEEE, 2020.
>
> [R6] Gu, Jiaqi, et al. "L2ight: Enabling on-chip learning for optical neural networks via efficient in-situ subspace optimization." *Advances in Neural Information Processing Systems* 34 (2021): 8649-8661.
>
> [R8] Hao, Zhongkai, et al. "Pinnacle: A comprehensive benchmark of physics-informed neural networks for solving pdes." arXiv preprint arXiv:2306.08827 (2023).

---

> ### Author Response · Authors · 2024-11-21
> **Response to Reviewer kjo5 (Part VI)**
>
> 3. **Comparisons with a less-overparameterized PINN model**:
>
> We have conducted experiments to compare a smaller 3-layer MLP (hidden size 32, 1793 parameters) and our proposed tensor-train (TT) compressed model (1929 parameters) on solving 20-dim HJB equation. We chose this setting because this MLP has similar model size with our TT-compressed PINN. The results are provided in Table R7. **In both FO and ZO training, the converged relative $\ell_2$ errors of the small MLP model are worse than that of our proposed TT compressed model by over one order-of-magnitude**.
>
> We have conducted an ablation study on MLP model size to show that a large hidden layer is favored to ensure enough model expressivity. Please refer to our response to ***Weakness 4***.
>
> Table R7: Comparison between a less-overparameterized PINN model and our tensor-train (TT) compressed model. We report the averaged results and standard deviation of three independent experiments.
>
> |  | MLP (hidden size 32) | TT (ours) |
> | --- | --- | --- |
> | Parameters | 1793 | 1929 |
> | FO training rel. $\ell_2$ error | (9.25±0.27)E-03 | (2.05±0.39)E-04 |
> | ZO training rel. $\ell_2$ error | (1.48±0.33)E-02 | (1.54±0.35)E-03 |

---

> > ### Author Response · Authors · 2024-11-24
> > **Reminder on follow-up discussion (2 day left before rebuttal ends)**
> >
> > Dear Reviewer kjo5,
> >
> > Thank you very much for the dedicated review of our paper. We are fully aware of the commitment and time your review entails. Your efforts are deeply valued by us, and we have worked hard to address all of your comments.
> >
> > With 2 day remaining before the end of the discussion phase, we wish to extend a respectful request for your feedback about our responses. Your insights are of immense importance to us, and we eagerly anticipate your updated evaluation.
> >
> > If you find our responses informative and useful, we would be grateful for your acknowledgement.
> >
> > Meanwhile, if you have any further inquiries or require additional clarifications, please don't hesitate to reach out. We are fully committed to providing additional responses during this discussion phase.
> >
> > Best regards,
> >
> > Authors

---

> ### Comment · Reviewer_kjo5 · 2024-11-25
> **Thank you for your responce**
>
> I appreciate the authors' detailed response a lot. However,
> * **My concern about novelty still remains**.  As the authors admit, both sparse grid and tensor decomposition are borrowed. However, I didn't see a significant modification. For example, low-rank compression with a tensor train is a typical method in ML compression, and the authors directly introduce this to shrink # trainable parameters to stabilize ZO performance. So, I still feel this paper lacks of sufficient core contributions.
> * **Comparison with FLOPS and L2ight.** I saw the authors try on the MNIST dataset and show FLOPS yields very bad performance in Table 11. I wonder whether you have tried L2ight, as the L2ight paper's performance is quite good. Moreover, did you enable the mapping phase in L2ight when you run L2ight? As I saw in Burgers's task, L2ight is even worse than FLOPS.
> * **I understand ZO is hard to compete with FO, but your demonstration is still on simple models (2-layer MLP) and some simple PDE tasks.** For example, on the Navier-Stokes PDE task (which is still not too hard...), your method only shows slight improvement over L2ight, still making me concerned about your method on hard tasks...

---

> ### Author Response · Authors · 2024-11-25
> **Response to reviewer kjo5 for follow-up questions (Part I)**
>
> Dear reviewer kjo5,
>
> Thanks very much for your additional follow-up questions. Here are our detailed responses.
>
> 1. ***My concern about novelty still remains**. As the authors admit, both sparse grid and tensor decomposition are borrowed. However, I didn't see a significant modification. For example, low-rank compression with a tensor train is a typical method in ML compression, and the authors directly introduce this to shrink # trainable parameters to stabilize ZO performance. So, I still feel this paper lacks of sufficient core contributions.*
>
> Thanks for letting us know your remaining concern. We fully understand and respect the reviewer’s different opinions. Meanwhile, we still want to humbly point out the following important facts, which will help evaluating our contributions more comprehensively and fairly.
>
> - **The role and application of sparse grid in our work is different from those in the applied math community**. Instead of using sparse grid for functional approximation and uncertainty quantification [R1] as done in applied math, we use sparse grid to **avoid the back propagation in PINN loss function computation**. **This overcomes a major challenge that prevents photonic training of PINN, which has not been reported before**.
> - While tensor-train decomposition has been used in ML for model compression, **our motivation and goal here is different: we aim to reduce the gradient variance and improve the convergence of ZO training**. Traditional TT compression does not necessarily improve the training convergence, but our ZO-TT method can greatly improve the training convergence in all tested cases due to the huge variance reduction of ZO gradients. The parameter reduction and hardware complexity reduction is a by-product.  It is worth noting that many recent ZO training methods (including some high-impact ones) also borrowed ideas from existing methods to improve the ZO training performance and advance the state-of-the-art. For example, MeZO [R2] used vanilla ZO gradient estimation for LLM fine tuning. DeepZero [R3] borrowed the ideas from sparse training to improve the ZO training efficiency. Ref. [R4] borrowed the idea of existing SVRG to improve the ZO fine-tuning of LLM.
> - Finally, it is worth noting that the **TT variance reduction is only one of the three contributions in our work**. The other two contributions are: (1) sparse-grid BP-free PINN loss evaluation, (2) design and evaluation of the completely BP-free photonic training accerator. **These three contributions combined enable us to train a real-size PINN on photonics**.
>
> 2. ***Comparison with FLOPS and L2ight.** I saw the authors try on the MNIST dataset and show FLOPS yields very bad performance in Table 11. I wonder whether you have tried L2ight, as the L2ight paper's performance is quite good. Moreover, did you enable the mapping phase in L2ight when you run L2ight? As I saw in Burgers's task, L2ight is even worse than FLOPS.*
>     - Thanks for pointing out this issue. Yes, we enabled the mapping phase in L2ight. If we disable the phase mapping, that is to only update the bias terms, L2ight can only achieve 8.79E-01 relative $\ell_2$ error in Burgers’ task.
>     - Motivated by your question, we just tried L2ight on the MNIST dataset. L2ight achieved 95.6% validation accuracy. The performance of L2ight versus our method should be considered case by case. L2ight does not have additional gradient error due to its FO optimization. Meanwhile, its sub-space training can prevent the solver to achieve a good optimal solution. The real performance depends on the trade-off of these two facts. In our PINN experiments, L2ight underperforms our method because the limitation of its sub-space training plays a dominant role. L2ight performs better on the MNIST dataset, probably because the model is more over-parameterized that even a subspace training can achieve a good optimal solution.
>     - Meanwhile, we would like to mention that both FLOPS and L2ight cannot directly train PINNs. As a result, we have improved them by using our sparse-grid BP-free PINN loss computation in order to handle PINN examples.
>
>     We hope that the above details have addressed your questions about the comparision with FLOPS and L2ight.
>
> [R1] Nobile, Fabio, Raúl Tempone, and Clayton G. Webster. "A sparse grid stochastic collocation method for partial differential equations with random input data." *SIAM Journal on Numerical Analysis* 46.5 (2008): 2309-2345.
>
> [R2] Malladi, Sadhika, et al. "Fine-tuning language models with just forward passes." *Advances in Neural Information Processing Systems* 36 (2023): 53038-53075.
>
> [R3] Chen, Aochuan, et al. "Deepzero: Scaling up zeroth-order optimization for deep model training." * ICLR 2024.
>
> [R4]. Gautam, Tanmay, et al. "Variance-reduced Zeroth-Order Methods for Fine-Tuning Language Models." *International Conference on Machine Learning, 2024*

---

> ### Author Response · Authors · 2024-11-25
> **Response to reviewer kjo5 for follow-up questions (Part II)**
>
> 3. *I understand ZO is hard to compete with FO, but your demonstration is still on simple models (2-layer MLP) and some simple PDE tasks. For example, on the Navier-Stokes PDE task (which is still not too hard...), your method only shows slight improvement over L2ight, still making me concerned about your method on hard tasks…*
>
>       Thanks a lot for sharing your thoughts on benchmarks. These PDE benchmarks are actually not simple/toy examples as they may seem.
>
> - **MLP Models**: MLP is the most widely used architecture in PINN. In both Burgers’ and Navier-Stokes PDE task, we used the 5-layer MLP model as baselines, the same model architecture used in the state-of-the-art PINNs benchmark [R5].
> - **Regarding the difficulty of PDE tasks**: unlike image classification where the problem complexity is normally decided by a neural network size and architecture, **the complexity of PINN training is often dominated by the PDE operator and initial/boundary conditions**.
>     - **Actually the 20-dim HJB PDE is a challenging example**. The PDE dimensionality is 20, which is significantly higher than most PDE dimensions (e.g., 2 or 3) used in science and engineering. Further, this is a hyperbolic PDE, which is much more difficult to solve than elliptic PDEs. In fact, even solving such a high-dim HJB PDE using a first-order method is a challenging task, and its FO training is still under heavy investigation by many recent papers in the math community (see [R8] and [R9] for instance) . This is also why safety-critical control of robotic and autonomous systems is so challenging from the perspective of math foundation.  While our chosen benchmark may not represent the most challenging case of HJB PDE,  the zeroth-order training plus hardware constraints (e.g., memory, device footprints and device noises) make it super challenging for photonic computing;
>     - **Navier-Stokes PDE is hard for ZO and photonic training**: indeed this is not a challenging task if we solve it with traditional discretizaiton-based approaches such as finite difference of finite-volume methods. However, **the complicated optimization landscape in the PINN formulation makes it much more challenging for both FO and ZO training**. Specifically, the FO training takes 15,000 iterations to converge, while FO training of the HJB PDE needs 4000 iterations. Due to the additional ZO gradient errors and photonic device noises, all ZO methods and photonics-aware training used in our paper fail to converge to a good solution, as mentioend in our previous response. Among these ZO and photonics-aware solvers, our method achieves the best performance;
>     - We also want to point out that **none of these PDE benchmarks can be solved directly via existing photonic training accelerators**.  They were solved by FLOPS [R6] and L2ight [R7] in our work (producing higher errors than our proposed method), because we have modified FLOP and L2ight by using our proposed sparse-grid BP-free loss evaluation. Without using our sparse-grid BP-free PINN loss evaluation, FLOPS and L2ight cannot handle these benchmarks.
>
> - **Slight improvement over L2ight**:
>     - We elaborate on this concern in the following Part III due to the page limitation here
>
> [R5] Hao, Zhongkai, et al. "Pinnacle: A comprehensive benchmark of physics-informed neural networks for solving pdes." *arXiv preprint arXiv:2306.08827* (2023).
>
> [R6] Gu, Jiaqi, et al. "FLOPS: Efficient on-chip learning for optical neural networks through stochastic zeroth-order optimization." *2020 57th ACM/IEEE Design Automation Conference (DAC)*. IEEE, 2020.
>
> [R7] Gu, Jiaqi, et al. "L2ight: Enabling on-chip learning for optical neural networks via efficient in-situ subspace optimization." *Advances in Neural Information Processing Systems* 34 (2021): 8649-8661.
>
> [R8] Nakamura-Zimmerer, Tenavi, Qi Gong, and Wei Kang. "Adaptive deep learning for high-dimensional Hamilton--Jacobi--Bellman equations." *SIAM Journal on Scientific Computing* 43.2 (2021): A1221-A1247.
>
> [R9] Zhou, Mo, Jiequn Han, and Jianfeng Lu. "Actor-critic method for high dimensional static Hamilton--Jacobi--Bellman partial differential equations based on neural networks." *SIAM Journal on Scientific Computing* 43.6 (2021): A4043-A4066.

---

> ### Author Response · Authors · 2024-11-25
> **Response to reviewer kjo5 for follow-up questions (Part III)**
>
> - **Slight improvement over L2ight**:
>     - As explained above, L2ight itself cannot train PINNs. We combined our proposed sparse-grid loss computation and L2ight’s subspace FO training to demonstrate L2ight’s performance on PINN training.
>     - In addition, we would like to mention again that all ZO and photonic training methods do not converge well for the Naiver-Stokes equation due to the complicated optimization landscape. As a result, **the slight improvement over L2ight should be regarded as an intermediate result without good convergence (for all methods) rather than a final conclusion**. In order to have a more comprehensive evaluation between our method and L2ight on new PDE benchmarks, we present **a 2D Darcy flow problem**, which is a well-known benchmark in the study of neural operators [R10, R11]. The detailed results are shown in **Tables R1-R3**. As indicated in Table R3, **our method achieves a 1.31X smaller error than L2ight when both methods converged**.
>     - Moreover, **L2ight method requires a more complicated photonic hardware design**. L2ight requires 1) a reciprocal photonic tensor core to perform back-propagation, 2) additional detectors to read out intermediate results and 3) additional memory to store the intermediate results. In comparison, our method’s photonic accelerator part (TONN in Figure 2) is exactly the same as the inference (forward) accelerator design. We do not need reciprocal optical computing, extra detectors, or extra memory.
>
> Overall, our method is the first solution to enable real-size training PINNs on photonic computing hardware, achieves the best accuracy in photonics simulation, and requires easier photonic hardware design. These indicate the benefits of our proposed method.
>
> With the above details, we hope that we have addressed your additional questions and that our submission can be considered more favorably. However, please feel free to reach out if you have any additional questions.
>
> Table R1: Darcy Flow example. Relative $\ell_2$ error of FO training in weight domain using different loss computation methods.
>
> | Problem | AD | SE | SG (ours) |
> | --- | --- | --- | --- |
> | Darcy Flow | 7.25E-02 | 7.39E-02 | 7.07E-02 |
>
> Table R2: Darcy Flow example. Relative $\ell_2$ error of different training method in weight domain.
>
> | Problem | Standard, FO | TT, FO | Standard, ZO | TT, ZO (ours) |
> | --- | --- | --- | --- | --- |
> | Darcy Flow | 7.07E-02 | 7.65E-02 | 2.26E-01 | 8.93E-02 |
>
> Table R3: Darcy Flow example. Relative $\ell_2$ error of phase domain training on photonics.
>
> | Problem | FLOPS | L2ight | ours |
> | --- | --- | --- | --- |
> | Darcy Flow | 4.80E-01 | 1.27E-01 | 9.60E-02 |
>
> [R10] Li, Zongyi, et al. "Fourier neural operator for parametric partial differential equations." *arXiv preprint arXiv:2010.08895* (2020).
>
> [R11] Li, Zongyi, et al. "Physics-informed neural operator for learning partial differential equations." *ACM/JMS Journal of Data Science* 1.3 (2024): 1-27.

---

> ### Author Response · Authors · 2024-11-27
> **your follow-up comments are incoorperated into our revised manuscript.**
>
> Dear Reviewer kjo5,
>
> Thanks again for your follow-up questions.
>
> Since ICLR has extended the discussion period, we have uploaded a revised manuscript to incorporate your and other reviewers' new comments in the follow-up discussion.
>
> We would highly appreciate it if you are available to take a look at our updated manuscript as well as our technical responses to your additional questions.
>
> Thanks a lot for your attention!
>
> Best regards,
>
> The authors.

---

> > ### Comment · Reviewer_kjo5 · 2024-11-28
> > **Thanks for authors' detailed comments**
> >
> > Thanks for the authors' detailed rebuttal. However, I am still inclined to keep my score as
> > * **I still think the paper is a good application paper by combining/applying previous techniques, but it lacks sufficient ML core contributions** The three contributions the author re-raised are (1) sparse-grid BP-free PINN loss evaluation, (2) TT variance reduction (3) design and evaluation of the completely BP-free photonic training accelerator. **First, the (3) is not novel, as there are already some tensorized ONNs (TONN) [1].** Second, as I mentioned before, both sparse-grid and TT variance are borrowed with the main aim of helping the ZO optimization, either reducing #queries or reducing variances. It is okay to borrow ideas to enhance performance if they are natural choices and enhance performance a lot. However,  it fails to show exciting improvement on some tasks, MINIST and Navier-Stokes PDE, making me think it still needs more concrete contributions to unlock more solid results.
> > * **Methods's effectiveness needs justification on reasonably hard tasks.** As PINN is a widely investigated area, the authors can try more different tasks, from easy to hard, to fairly justify the effectiveness and contribution, for example, from [2] (it seems some harder tasks exist here with much worse l2 loss with FO compared to those you use)
> >
> > Minors:
> > * I have a question for your claim, "Actually the 20-dim HJB PDE is a challenging example." I am curious why it is a challenging example as I saw your paper shows that l2 loss is in the e-4 level with just 3-layer MLPs.
> >
> > **References**
> >
> > [1] Xiao, X., On, M. B., Van Vaerenbergh, T., Liang, D., Beausoleil, R. G., & Yoo, S. J. (2021). Large-scale and energy-efficient tensorized optical neural networks on III–V-on-silicon MOSCAP platform. Apl Photonics, 6(12).
> >
> > [2] Hao, Zhongkai, et al. "Pinnacle: A comprehensive benchmark of physics-informed neural networks for solving pdes." arXiv preprint arXiv:2306.08827 (2023).

---

> ### Author Response · Authors · 2024-11-28
> **Thanks for reviewer's additional comments, Part 1**
>
> Dear Reviewer kjo5.
>
> Thanks a lot for your additional comments during the thanksgiving holiday.
>
> **Reviewer's Comment 1**: I still think the paper is a good application paper by combining/applying previous techniques, but it lacks sufficient ML core contributions The three contributions the author re-raised are (1) sparse-grid BP-free PINN loss evaluation, (2) TT variance reduction (3) design and evaluation of the completely BP-free photonic training accelerator. First, the (3) is not novel, as there are already some tensorized ONNs (TONN) [1]. Second, as I mentioned before, both sparse-grid and TT variance are borrowed with the main aim of helping the ZO optimization, either reducing #queries or reducing variances. It is okay to borrow ideas to enhance performance if they are natural choices and enhance performance a lot. However, it fails to show exciting improvement on some tasks, MINIST and Navier-Stokes PDE, making me think it still needs more concrete contributions to unlock more solid results.
>
> **Response**： With full respect to your different opinions, we hope that you can understand our disagreement on this issue. Our work addresed some core fundamental challenges of training on the edge hardware (scalability, convergence and BP-free trainining with extremely limited on-chip memory).
>
> --Regarding **contribution (3)**: the reference (Xiao 2021) you pointed out is an **inference accelerator** (which is explained well in our paper). As stated in our paper, our contribution is to propose a BP-free photonic PINN **training accelerator** that can easily reuse existing inference accelerator with minor modification to do training. The inference accelerator of (Xiao 2021) is used only as one block in our training accelerator for performing forward pass. **This has been made clear in Lines 095-096 in our paper submission** when we claimed the paper contribution in the introduction section: "Our design reuses a tensorized ONN inference accelerator, and just add a digital control system to implement on-chip BP-free training."
>
> --Regarding **Contribution (1) and (2)**: **Neither sparse-grid BP-free loss evaluation nor TT-based ZO gradient variance reduction have been reported before**. They are the key algorithm contributions that enables BP-free and scalable training of PINN for the first time. Meanwhile, **these two algorithm-level contributions can be applied to many other platforms** such as FPGA, edge GPU/CPU, probabilistic computers and distributed training platforms, so these contributions are generic enough to make broad impacts rather than being an application-focused work that only benefits a specific hardware platform.
>
> --Regarding **“fails to show exciting improvement on some tasks”**: our framework has already dramatically improved the scalability and ease of implementation for end-to-end photonic training accelerators. Our benchmarks are already significantly more challenging in terms of problem sizes and optimizaiton landscape than the benchmarks used in most (if not all) of the photonic training accelerators. This is also the first time to demonstrate the photonic training of PINNs. More importantly, we have shown the training of real-size PINNs rather than toy-size PINNs, which is super challenging for photonic AI platform due to its poor scalability and lack of photonic memory.
>
> It is also worth noting that our paper is under review in the track of **``infrastructure, software libraries, hardware, systems"**. This track, based on our understanding, expects a better balance between algorithm and hardware innovations, rather than pure theoretical innovations.
>
>
> **Reviewer's Comment 2**: Methods's effectiveness needs justification on reasonably hard tasks. As PINN is a widely investigated area, the authors can try more different tasks, from easy to hard, to fairly justify the effectiveness and contribution, for example, from [2] (it seems some harder tasks exist here with much worse l2 loss with FO compared to those you use)
>
> **Response**: Thanks a lot for this comment. We would like to remark that that our paper scope is about **photonic end-to-end training**, which has fundamental challenges in memory and scalability. This prevents handling arbitrarily large benchmarks as we do on GPU/CPU. The benchmarks we used in this paper is already harder than those used in recent photonic AI papers which often has only dozens of neurons in total for end-to-end training. Our work is also the first to handle the PINN loss via a BP-free method, enabling completely BP-free training of PINN on edge devices. Based on the reviewer's suggestions, **we have already added three different PDE benchmarks in the rebuttal phase**: Burgers’ equation, Naiver-Stokes PDE and Darcy flow PDE. They are not simple benchmarks. Instead, they are widely used benchmarks in the math community to challenge PINN models and first-order training algorithms.

---

> ### Author Response · Authors · 2024-11-28
> **Thanks for reviewer's additional comments, Part 2**
>
> **Minors comment**: I have a question for your claim, "Actually the 20-dim HJB PDE is a challenging example." I am curious why it is a challenging example as I saw your paper shows that l2 loss is in the e-4 level with just 3-layer MLPs.
>
> **Response**: HJB PDE is mainly used for safety-critical control of autonous systems, and much higher accuracy is required. An $\ell_2$ loss of e-4 does not mean that this is a perfect solution: this can be just an acceptable accuracy in practical engineering applications. If we look at Fig. 5 in our paper, our training method **reduced the loss by about 2 orders of magniidue compared with the starting loss, for both the HJB PDE and the Black-Scholes PDE**. This indicates that the HJB PDE training result is not better than other benchmarks. Meanwhile, the training curves of the HJB-PDE are much more oscillatory than the Black-Scholes PDE, indicating its **more complicated optimization landscape**.
>
> **Regarding reviewer's statement " However, I am still inclined to keep my score as."**:
>
> **Response**:  We appreciate all your technical comments in the review and rebuttal phases. Based on your comments, **we have provided the following new contents** in the rebuttal phase to address all actionable technical comments:
>
> --(1) Additional comparison of our method with existing photonic training in both space-multiplexing and time-multiplexing settings. The analysis show the advantages of our mehtod in both settings;
>
> --(2) More details to explain our (correct) implementation of FLOPS  and L2ight;
>
> --(3) Ablation study under different MLP model sizes to show that the PINN for the 20-dim HJB PDE is not over-parameterized;
>
> --(4) Theoretical explanation about the unavoidable performance gap between ZO and FO training (which widely exist in almost all ZO training papers);
>
> --(5) Three additional PDE experiments: Burgers’ equation, Naiver-Stokes PDE and Darcy flow PDE. Our methods show better performance than other tested ZO and photonic training methods on all these additional PDE benchmarks;
>
> --(6) Explanation about the results of FLOPS and L2ight versus our method on the MNIST example;
>
> --(7) Explanation regarding the less significant benefit of our method over other methods on the Naiver-Stokes PDE due to the training challenge, and further interpretations about the small erorr of the 20-dim HJB PDE.
>
> We fully understand and respect that the reviewer has different opinions regarding the paper's overall contributions. This a normal case in our diverse academic world: different people have different opinions. This is exactly one of the major forces that push science and technology forward.
>
> We are already very grateful that the reviewer's comments have helped us improved the paper quality significantly. While we do hope that the reviewer can raise his/her evaluation score to reflect our huge work [see Points (1) -(7) above] that have addressed massive technical comments/confusions/misunderstandings, we are also completely fine if the the reviewer decides to maintain his/her score.
>
> Lastly, thanks again for your follow-up discussion during the holiday!
>
> Best regards,
>
> The authors.

---

### Official Review · Reviewer_zjiy · 2024-11-05

**Soundness:** 3
**Presentation:** 4
**Contribution:** 3
**Rating:** 6
**Confidence:** 4

**Summary:**

The paper introduces a BP-free training framework for physics-informed neural networks on photonic hardware. The authors propose a sparse-grid Stein estimator to replace BP in loss evaluation; a tensor-compressed zeroth-order (ZO) optimization method to improve scalability, and scalable photonic accelerator design aimed at real-time PINN training. Through simulations on low- and high-dimensional PDE benchmarks, the paper demonstrates the efficacy of these methods in reducing training time and chip area requirements.

**Strengths:**

Strengths:

1.	The proposed method largely reduce the dimensionality in zeroth-order ONN training, thus can train realistic PINN model using sampling-based ZO optimizer.

2.	Compared to prior methods, it shows faster convergence and better error.

3.	The training process considers hardware nonideality, e.g., quantization, noises.

**Weaknesses:**

Weaknesses:

1.	Besides training from scratch, if there is a pretrained digital PINN model, how does the mapping efficiency of the proposed method compared to prior methods. This is also an actual deployment use case.

2.	What is the detailed photonic accelerator setting, e.g., core size, etc. Noticed the training parameters in Table 3 are different, especially L2ight has very few parameters. What could be this reason? Is it possible to keep a similar parameter size by modifying the model settings and comparing the training algorithm itself? Now the effect of #params and training methods are mixed.

3.	Any experiments on different TT-ranks? How does that impact the model expressivity compared to dense NNs? What is the trade-off between trainability and model expressivity?

**Questions:**

1.	Besides training from scratch, if there is a pretrained digital PINN model, how does the mapping efficiency of the proposed method compared to prior methods. This is also an actual deployment use case.

2.	What is the detailed photonic accelerator setting, e.g., core size, etc. Noticed the training parameters in Table 3 are different, especially L2ight has very few parameters. Is it related to the overly large sub-matrix size? How about using 8x8 matrix blocks so the diagonal still has enough parameters to train?

3.	Any experiments on different TT-ranks? How does that impact the model expressivity compared to dense NNs? What is the trade-off between trainability and model expressivity?

4.     Is there any hardware implementation difficulty for the assumed tensorized ONN TONN with complicated signal crossings?

---

> ### Author Response · Authors · 2024-11-20
> **Response to Reviewer zjiy (Part I)**
>
> ## Responses to weaknesses:
>
> ***Weakness 1**: Besides training from scratch, if there is a pretrained digital PINN model, how does the mapping efficiency of the proposed method compared to prior methods. This is also an actual deployment use case.*
>
> **Response**: Thanks a lot for the comments! We consider a more challenging end-to-end training task. **The mapping efficiency of training from scratch and fine-tuning is the same.** We just need to assign different values to photonic devices.
>
> - Training from scratch: we first initialize the weight matrix in a tensor-train (TT) cores format with a random initial guess, and map the digital TT core values to photonic TT cores values (e.g., MZI phase shifter values).
> - Pretraining then mapping: In the pre-training process, we first initialize the weight matrix in a tensor-train (TT) cores format with a random initial guess, and then directly updates the TT cores without any factorization to minimize the loss function. After pre-training, we map the digital TT core values to photonic TT cores values (e.g., MZI phase shifter values). There is **no extra latency overhead for low-rank tensor factorization**.
>
> Compared with prior work such FLOPS [R1] and L2ight [R2], our method also have better mapping efficiency in the fine-tuning setting, since our method requires much less photonic devices due to the tensor-compressed implementation.
>
> ***Weakness 2**: What is the detailed photonic accelerator setting, e.g., core size, etc. Noticed the training parameters in Table 3 are different, especially L2ight has very few parameters. What could be this reason? Is it possible to keep a similar parameter size by modifying the model settings and comparing the training algorithm itself? Now the effect of #params and training methods are mixed.*
>
> **Response**: Thank you for your insightful suggestions!
>
> 1. **Detailed photonic accelerator settings**: For baseline methods FLOPS [R1] and L2ight [R2], we adopt the same ONN settings as that in Ref. [R2]. The linear projection in an ONN adopts blocking matrix multiplication, where **a M × N weight matrix is implemented with P × Q blocks, each block is a 8 × 8 MZI mesh.** We have added detailed ONN implementation settings of baseline methods in the Appendix A.3.
> 2. **Very few trainable parameters in L2ight**: Our baseline experiments for L2ight method were correctly set-up by applying smaller-size (8 × 8) matrix blocks to implement large weight matrices. **This is consistent with your suggestions to ensure enough trainable parameters on the diagonal matrix.** We realize that the number of trainable parameters for L2ight in Table 3 was counted incorrectly while our training setup was correct. The numbers of trainable MZIs in L2ight **have been updated to 2,561 for Black Scholes and 35,841 for 20-dim HJB** respectively. Meanwhile, we would like to remark that this change will not change the conclusion, because: (1) our training experiment setup was correct, and the loss curve and convergence behavior will not change, (2) our method still have better accuracy and scalability than L2ight, (3) our method can handle PINN whereas previous methods cannot. We genuinely thank you for pointing this out.
>
> [R1] Gu, Jiaqi, et al. "FLOPS: Efficient on-chip learning for optical neural networks through stochastic zeroth-order optimization." *2020 57th ACM/IEEE Design Automation Conference (DAC)*. IEEE, 2020.
>
> [R2] Gu, Jiaqi, et al. "L2ight: Enabling on-chip learning for optical neural networks via efficient in-situ subspace optimization." *Advances in Neural Information Processing Systems* 34 (2021): 8649-8661.

---

> > ### Comment · Reviewer_zjiy · 2024-11-23
> > **Thanks for the Responses**
> >
> > All my questions are addressed. I will raise my score.

---

> > > ### Author Response · Authors · 2024-11-24
> > > **Thank you for the very positive feedback!**
> > >
> > > Dear Reviewer zjiy,
> > >
> > > We are happy to know that all your questions have been addressed. Your review feedback greatly enhanced the quality of our work. We sincerely appreciate your participation in the discussion and your recognition of the improved quality of our paper following the discussion process.

---

> ### Author Response · Authors · 2024-11-20
> **Response to Reviewer zjiy (Part II)**
>
> ***Weakness 3**: Any experiments on different TT-ranks? How does that impact the model expressivity compared to dense NNs? What is the trade-off between trainability and model expressivity?*
>
> **Response**: Thank you for mentioning this aspect.
>
> 1. It is important to have proper TT-ranks. This is a trade-off between model compression ratio and model expressivity. The TT-ranks can be empirically determined, or adaptively determined by automatic rank determination algorithms [R3, R4]. For such rank-adaptive training, we only need to add a regularization terms to the loss function. This only requires minor modifications in the digital control system and not change to the photonic hardware.
> 2. To validate our rank choice, **we have added an ablation study on different TT-ranks, as shown in Appendix A.7.1 in our updated manuscript.** The results are shown in **Tab. R1** below. We tested tensor-train compressed training with different TT-ranks on solving 20-dim HJB equations. The model setups are the same as illustrated in Appendix A.2. We fold the input layer and hidden layers as size $1\times 1\times 3\times 7\times 8\times 4\times 4\times 4$ and $4\times 4\times 4\times 8\times 8\times 4\times 4\times 4$, respectively, with TT-ranks [1,$r$,$r$,$r$,1]. We use automatic differentiation for loss evaluation and first-order (FO) gradient descent to update model parameters. Other training setups are the same as illustrated in Appendix A.3.  The results reveal that **models with larger TT-ranks have better model expressivity and achieve smaller relative $\ell_2$ error.** However, **increasing TT-ranks increases the hardware complexity (e.g., number of MZIs) of photonics implementation** as it increases the number of parameters. Therefore, we chose a small TT-rank as 2, which provides enough expressivity to solve the PDE equations, while maintains a small model size.
>
> Table R1: Ablation study on tensor-train (TT) ranks when training the TT compressed model on solving 20-dim HJB equations.
>
> | TT-rank | 2 | 4 | 6 | 8 |
> | --- | --- | --- | --- | --- |
> | Params | 1,929 | 2,705 | 3,865 | 5,409 |
> | rel. $\\ell_2$ error | (3.17±1.16)E-04 | (2.45±0.82)E-04 | (4.00±3.69)E-05 | (3.02±3.16)E-05 |
>
> ## Responses to questions:
>
> ***Question 1-3***: Please refer to our response to ***Weakness 1-3***
>
> ***Question 4**: Is there any hardware implementation difficulty for the assumed tensorized ONN TONN with complicated signal crossings?*
>
> **Response**: Thank you for raising this concern!
>
> The cross-connects in TONN will induce a lot of waveguide crossings. However, unlike electronic wire crossings, optical waveguide crossings can have extremely low losses with proper design. For instance, the single-layer silicon multimode interference (MMI) based waveguide crossing has an insertion loss of 0.017 dB/crossing [R5]. The optical insertion loss for multi-layer silicon waveguides can be as low as 0.0003 dB/crossing [R6].
>
> [R3] Hawkins, Cole, Xing Liu, and Zheng Zhang. "Towards compact neural networks via end-to-end training: A Bayesian tensor approach with automatic rank determination." *SIAM Journal on Mathematics of Data Science* 4.1 (2022): 46-71.
>
> [R4] Yang, Zi, et al. "CoMERA: Computing-and Memory-Efficient Training via Rank-Adaptive Tensor Optimization." *arXiv preprint arXiv:2405.14377* (2024).
>
> [R5] Ma, Y., Zhang, Y., Yang, S., Novack, A., Ding, R., Lim, A. E. J., ... & Hochberg, M. (2013). Ultralow loss single layer submicron silicon waveguide crossing for SOI optical interconnect. Optics express, 21(24), 29374-29382.
>
> [R6] Chiles, J., Buckley, S., Nader, N., Nam, S. W., Mirin, R. P., & Shainline, J. M. (2017). Multi-planar amorphous silicon photonics with compact interplanar couplers, cross talk mitigation, and low crossing loss. APL Photonics, 2(11).

---

### Meta-Review · Area_Chair_d9CY · 2024-12-19

**Metareview:**

This paper proposes a method to train PINNs without the backpropagation. Only with the forward pass, this paper approximates the gradient and train PINNs. Moreover, their training methods are based on special photonic accelerators. They focused on reducing the complexity differently from other zeroth-order training methods. In order to show the efficacy of their method, they conducted experiments with various PDEs and one simple image classification task.

Although interesting, this work also has the following limitations: 1) vanilla PINNs are outdated and many other enhanced PINNs for solving (parameterized) PDEs have been developed. This training method needs to be tested with them. 2) It is vague to connect the photonic accelerator and the PINN. Although they showed experiments with image classification, those tasks are considered too old in the deep learning field.

I think the authors need to show the following things further: 1) they need to use more advanced PINNs, e.g., P2INN [Cho et al., ICML 2024 Oral], 2) in the current writing, they abruptly jump from the PINN to the photonic computing in Introduction. I think this transition is not smooth and do not see any specialized reason of using the photonic accelerator for PINNs (but not for Transformers).

**Additional Comments On Reviewer Discussion:**

The authors showed such faithful responses to the reviewers and made good rebuttal. Most reviewers are also satisfied and one reviewer gave 3, which I think too low.

All in all, I feel ambivalent to this paper. I can understand their contribution, but I think their goal is too narrow in my opinion (unless they show a strong connection between the PINN and the photonic computing).

---

### Decision · Program_Chairs · 2025-01-22

Reject